# Towards Explaining the Regularization Effect of Initial Large Learning Rate in Training Neural Networks

**Yuanzhi Li**
Machine Learning Department
Carnegie Mellon University
yuanzhil@andrew.cmu.edu

**Colin Wei**
Computer Science Department
Stanford University
colinwei@stanford.edu

**Tengyu Ma**
Computer Science Department
Stanford University
tengyuma@stanford.edu

## Abstract

Stochastic gradient descent with a large initial learning rate is widely used for training modern neural net architectures. Although a small initial learning rate allows for faster training and better test performance initially, the large learning rate achieves better generalization soon after the learning rate is annealed. Towards explaining this phenomenon, we devise a setting in which we can prove that a two layer network trained with large initial learning rate and annealing provably generalizes better than the same network trained with a small learning rate from the start. The key insight in our analysis is that the order of learning different types of patterns is crucial: because the small learning rate model first memorizes easy-to-generalize, hard-to-fit patterns, it generalizes worse on hard-to-generalize, easier-to-fit patterns than its large learning rate counterpart. This concept translates to a larger-scale setting: we demonstrate that one can add a small patch to CIFAR-10 images that is immediately memorizable by a model with small initial learning rate, but ignored by the model with large learning rate until after annealing. Our experiments show that this causes the small learning rate model's accuracy on unmodified images to suffer, as it relies too much on the patch early on.

## 1 Introduction

It is a commonly accepted fact that a large initial learning rate is required to successfully train a deep network even though it slows down optimization of the train loss. Modern state-of-the-art architectures typically start with a large learning rate and anneal it at a point when the model's fit to the training data plateaus [25, 32, 17, 42]. Meanwhile, models trained using only small learning rates have been found to generalize poorly despite enjoying faster optimization of the training loss.

A number of papers have proposed explanations for this phenomenon, such as sharpness of the local minima [22, 20, 24], the time it takes to move from initialization [18, 40], and the scale of SGD noise [38]. However, we still have a limited understanding of a surprising and striking part of the large learning rate phenomenon: from looking at the section of the accuracy curve before annealing, it would appear that a small learning rate model should outperform the large learning rate model in both training and test error. Concretely, in Fig. 1, the model trained with small learning rate outperforms

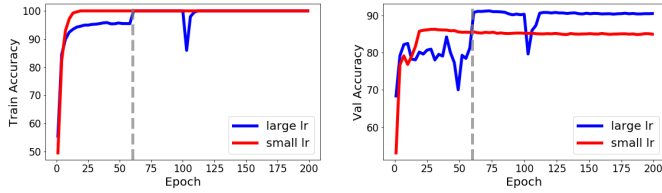

Figure 1: CIFAR-10 accuracy vs. epoch for WideResNet with weight decay, no data augmentation, and initial lr of 0.1 vs. 0.01. Gray represents the annealing time. **Left:** Train. **Right:** Validation.

the large learning rate until epoch 60 when the learning rate is first annealed. Only after annealing does the large learning rate visibly outperform the small learning rate in terms of generalization.

In this paper, we propose to theoretically explain this phenomenon via the concept of *learning order* of the model, i.e., the rates at which it learns different types of examples. This is not a typical concept in the generalization literature — learning order is a training-time property of the model, but most analyses only consider post-training properties such as the classifier's complexity [8], or the algorithm's output stability [9]. We will construct a simple distribution for which the learning order of a two-layer network trained under large and small initial learning rates determines its generalization.

Informally, consider a distribution over training examples consisting of two types of patterns ("pattern" refers to a grouping of features). The first type consists of a set of easy-to-generalize (i.e., discrete) patterns of low cardinality that is difficult to fit using a low-complexity classifier, but easily learnable via complex classifiers such as neural networks. The second type of pattern will be learnable by a low-complexity classifier, but are inherently noisy so it is difficult for the classifier to generalize. In our case, the second type of pattern requires *more samples* to correctly learn than the first type. Suppose we have the following split of examples in our dataset:

$$20\% \text{ containing only easy-to-generalize and hard-to-fit patterns}$$
$$20\% \text{ containing only hard-to-generalize and easy-to-fit patterns} \tag{1.1}$$
$$60\% \text{ containing both pattern types}$$

The following informal theorems characterize the learning order and generalization of the large and small initial learning rate models. They are a dramatic simplification of our Theorems 3.4 and 3.5 meant only to highlight the intuitions behind our results.

**Theorem 1.1** (Informal, large initial LR + anneal). *There is a dataset with size $N$ of the form* (1.1) *such that with a large initial learning rate and noisy gradient updates, a two layer network will:*

*1) initially only learn hard-to-generalize, easy-to-fit patterns from the $0.8N$ examples containing such patterns.*

*2) learn easy-to-generalize, hard-to-fit patterns only after the learning rate is annealed.*

*Thus, the model learns hard-to-generalize, easily fit patterns with an effective sample size of $0.8N$ and still learns all easy-to-generalize, hard to fit patterns correctly with $0.2N$ samples.*

**Theorem 1.2** (Informal, small initial LR). *In the same setting as above, with small initial learning rate the network will:*

*1) quickly learn all easy-to-generalize, hard-to-fit patterns.*

*2) ignore hard-to-generalize, easily fit patterns from the $0.6N$ examples containing both pattern types, and only learn them from the $0.2N$ examples containing only hard-to-generalize patterns.*

*Thus, the model learns hard-to-generalize, easily fit patterns with a smaller effective sample size of $0.2N$ and will perform relatively worse on these patterns at test time.*

Together, these two theorems can justify the phenomenon observed in Figure 1 as follows: in a real-world network, the large learning rate model first learns hard-to-generalize, easier-to-fit patterns and is unable to memorize easy-to-generalize, hard-to-fit patterns, leading to a plateau in accuracy. Once the learning rate is annealed, it is able to fit these patterns, explaining the sudden spike in both train and test accuracy. On the other hand, because of the low amount of SGD noise present in easy-to-generalize, hard-to-fit patterns, the small learning rate model quickly overfits to them before fully learning the hard-to-generalize patterns, resulting in poor test error on the latter type of pattern.

Both intuitively and in our analysis, the non-convexity of neural nets is crucial for the learning-order effect to occur. Strongly convex problems have a unique minimum, so what happens during training does not affect the final result. On the other hand, we show the non-convexity causes the learning order to highly influence the characteristics of the solutions found by the algorithm.

In Section F.1, we propose a mitigation strategy inspired by our analysis. In the same setting as Theorems 1.1 and 1.2, we consider training a model with small initial learning rate while adding noise before the activations which gets reduced by some constant factor at some particular epoch in training. We show that this algorithm provides the same theoretical guarantees as the large initial learning rate, and we empirically demonstrate the effectiveness of this strategy in Section 6. In Section 6 we also empirically validate Theorems 1.1 and 1.2 by adding an artificial memorizable patch to CIFAR-10 images, in a manner inspired by (1.1).

## 1.1 Related Work

The question of training with larger batch sizes is closely tied with learning rate, and many papers have empirically studied large batch/small LR phenomena [22, 18, 35, 34, 11, 41, 16, 38], particularly focusing on vision tasks using SGD as the optimizer.[1] Keskar et al. [22] argue that training with a large batch size or small learning rate results in sharp local minima. Hoffer et al. [18] propose training the network for longer and with larger learning rate as a way to train with a larger batch size. Wen et al. [38] propose adding Fisher noise to simulate the regularization effect of small batch size.

Adaptive gradient methods are a popular method for deep learning [14, 43, 37, 23, 29] that adaptively choose different step sizes for different parameters. One motivation for these methods is reducing the need to tune learning rates [43, 29]. However, these methods have been observed to hurt generalization performance [21, 10], and modern architectures often achieve the best results via SGD and hand-tuned learning rates [17, 42]. Wilson et al. [39] construct a toy example for which ADAM [23] generalizes provably worse than SGD. Additionally, there are several alternative learning rate schedules proposed for SGD, such as warm-restarts [28] and [33]. Ge et al. [15] analyze the exponentially decaying learning rate and show that its final iterate achieves optimal error in stochastic optimization settings, but they only analyze convex settings.

There are also several recent works on implicit regularization of gradient descent that establish convergence to some idealized solution under particular choices of learning rate [27, 36, 1, 7, 26]. In contrast to our analysis, the generalization guarantees from these works would depend only on the complexity of the final output and not on the order of learning.

Other recent papers have also studied the order in which deep networks learn certain types of examples. Mangalam and Prabhu [30] and Nakkiran et al. [31] experimentally demonstrate that deep networks may first fit examples learnable by "simpler" classifiers. For our construction, we prove that the neural net with large learning rate follows this behavior, initially learning a classifier on linearly separable examples and learning the remaining examples after annealing. However, the phenomenon that we analyze is also more nuanced: with a small learning rate, we prove that the model first learns a complex classifier on low-noise examples which are not linearly separable.

Finally, our proof techniques and intuitions are related to recent literature on global convergence of gradient descent for over-parametrized networks [6, 12, 13, 1, 5, 7, 4, 26, 2]. These works show that gradient descent learns a *fixed* kernel related to the initialization under sufficient over-parameterization. In our analysis, the underlying kernel is *changing* over time. The amount of noise due to SGD governs the space of possible learned kernels, and as a result, regularizes the order of learning.

## 2 Setup and Notations

**Data distribution.** We formally introduce our data distribution, which contains examples supported on two types of components: a $\mathcal{P}$ component meant to model hard-to-generalize, easier-to-fit patterns, and a $\mathcal{Q}$ component meant to model easy-to-generalize, hard-to-fit patterns (see the discussion in our introduction). Formally, we assume that the label $y$ has a uniform distribution over $\{-1, 1\}$, and the

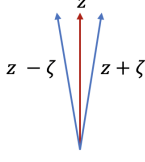

Figure 2: A visualization of the vectors $z$, $z - \zeta$, and $z + \zeta$ used to define the distribution $\mathcal{Q}$ in 2 dimensions. $z \pm \zeta$ will have label $-1$ and $z$ has label $+1$. Note that the norm of $\zeta$ is much smaller than the norm of $z$.

data $x$ is generated as

$$\text{Conditioned on the label } y \tag{2.1}$$
$$\text{with probability } p_0, \quad x_1 \sim \mathcal{P}_y, \text{ and } x_2 = 0 \tag{2.2}$$
$$\text{with probability } q_0, \quad x_1 = 0, \text{ and } x_2 \sim \mathcal{Q}_y \tag{2.3}$$
$$\text{with probability } 1 - p_0 - q_0, \quad x_1 \sim \mathcal{P}_y, \text{ and } x_2 \sim \mathcal{Q}_y \tag{2.4}$$

where $\mathcal{P}_{-1}, \mathcal{P}_1$ are assumed to be two half Gaussian distributions with a margin $\gamma_0$ between them:

$$x_1 \sim \mathcal{P}_1 \Leftrightarrow x_1 = \gamma_0 w^\star + z | \langle w^\star, z \rangle \geq 0, \text{ where } z \sim \mathcal{N}(0, I_{d \times d}/d)$$
$$x_1 \sim \mathcal{P}_{-1} \Leftrightarrow x_1 = -\gamma_0 w^\star + z | \langle w^\star, z \rangle \leq 0, \text{ where } z \sim \mathcal{N}(0, I_{d \times d}/d)$$

Therefore, we see that when $x_1$ is present, the linear classifier $\text{sign}(w^{\star \top} x_1)$ can classify the example correctly with a margin of $\gamma_0$. To simplify the notation, we assume that $\gamma_0 = 1/\sqrt{d}$ and $w^\star \in \mathbb{R}^d$ has a unit $\ell_2$ norm. Intuitively, $\mathcal{P}$ is linearly separable, thus learnable by *low complexity* (e.g. linear) classifiers. However, because of the dimensionality, $\mathcal{P}$ has high noise and requires a relatively large sample complexity to learn. The distribution $\mathcal{Q}_{-1}$ and $\mathcal{Q}_1$ are supported only on three distinct directions $z - \zeta, z$ and $z + \zeta$ with some random scaling $\alpha$, and are thus low-noise and memorizable. Concretely, $z - \zeta$ and $z + \zeta$ have negative labels and $z$ has positive labels.

$$x_2 \sim \mathcal{Q}_1 \Leftrightarrow x_2 = \alpha z \text{ with } \alpha \sim [0, 1] \text{ uniformly}$$
$$x_2 \sim \mathcal{Q}_{-1} \Leftrightarrow x_2 = \alpha(z + b\zeta) \text{ with } \alpha \sim [0, 1], b \sim \{-1, 1\} \text{ uniformly} \tag{2.5}$$

Here for simplicity, we take $z$ to be a unit vector in $\mathbb{R}^d$. We assume $\zeta \in \mathbb{R}^d$ has norm $\|\zeta\|_2 = r$ and $\langle z, \zeta \rangle = 0$. We will assume $r \ll 1$ so that $z + \zeta, z, z - \zeta$ are fairly close to each other. We depict $z - \zeta, z, z + \zeta$ in Figure 2. We choose this type of $\mathcal{Q}$ to be the easy-to-generalize, hard-to-fit pattern. Note that $z$ is not linearly separable from $z + \zeta, z - \zeta$, so non-linearity is necessary to learn $\mathcal{Q}$. On the other hand, it is also easy for *high-complexity* models such as neural networks to memorize $\mathcal{Q}$ with relatively small sample complexity.

**Memorizing $\mathcal{Q}$ with a two-layer net.** It is easy for a two-layer relu network to memorize the labels of $x_2$ using two neurons with weights $w, v$ such that $\langle w, z \rangle < 0$, $\langle w, z - \zeta \rangle > 0$ an $\langle v, z \rangle < 0$, $\langle v, z + \zeta \rangle > 0$. In particular, we can verify that $-\langle w, x_2 \rangle_+ - \langle v, x_2 \rangle_+$ will output a negative value for $x_2 \in \{z - \zeta, z + \zeta\}$ and a zero value for $x_2 = z$. Thus choosing a small enough $\rho > 0$, the classifier $-\langle w, x_2 \rangle_+ - \langle v, x_2 \rangle_+ + \rho$ gives the correct sign for the label $y$.

We assume that we have a training dataset with $N$ examples $\{(x^{(1)}, y^{(1)}), \cdots, (x^{(N)}, y^{(N)})\}$ drawn i.i.d from the distribution described above. We use $p$ and $q$ to denote the empirical fraction of data points that are drawn from equation (2.2) and (2.3).

**Two-layer neural network model.** We will use a two-layer neural network with relu activation to learn the data distribution described above. The first layer weights are denoted by $U \in \mathbb{R}^{m \times 2d}$ and the second layer weight is denoted by $u \in \mathbb{R}^m$. With relu activation, the output of the neural network is $u^\top (\mathbb{1}(Ux) \odot Ux)$ where $\odot$ denotes the element-wise dot product of two vectors and $\mathbb{1}(z)$ is the binary vector that contains $\mathbf{1}(z_i \geq 0)$ as entries. It turns out that we will often be concerned with the object that disentangles the two occurrences of $U$ in the formula $u^\top (\mathbb{1}(Ux) \odot Ux)$. We define the following notation to facilitate the reference to such an object. Let

$$N_A(u, U; x) \triangleq w^\top (\mathbb{1}(Ax) \odot Ux) \tag{2.6}$$

That is, $N_A(w, W; x)$ denotes the function where we compute the activation pattern $\mathbb{1}(Ax)$ by the matrix $A$ instead of $U$. When $u$ is clear from the context, with slight abuse of notation, we write $N_A(U; x) \triangleq u^\top (\mathbb{1}(Ax) \odot Ux)$. In this notation, our model is defined as $f(u, U; x) = N_U(u, U; x)$. We consider several different structures regarding the weight matrices $U$. The simplest version

which we consider in the main body of this paper is that $U$ can be decomposed into two $U = \begin{bmatrix} W \\ V \end{bmatrix}$ where $W$ only operates on the first $d$ coordinates (that is, the last $d$ columns of $W$ are zero), and $V$ only operates on the last $d$ coordinates (those coordinates of $x_2$.) Note that $W$ operates on the $\mathcal{P}$ component of examples, and $V$ operates on the $\mathcal{Q}$ component of examples. In this case, the model can be decomposed into

$$f(u, U; x) = N_U(u, U; x) = N_W(w, W; x) + N_V(v, V; x) = N_W(w, W; x_1) + N_V(v, V; x_2)$$

Here we slightly abuse the notation to use $W$ to denote both a matrix of $2d$ columns with last $d$ columns being zero, or a matrix of $d$ columns. We also extend our theorem to other $U$ such as a two layer convolution network in Section F.

**Training objective.** Let $\ell(f; (x, y))$ be the loss of the example $(x, y)$ under model $f$. Throughout the paper we use the logistic loss $\ell(f; (x, y)) = -\log \frac{1}{1+e^{-yf(x)}}$. We use the standard training loss function $\widehat{L}$ defined as: $\widehat{L}(u, U) = \frac{1}{N} \sum_{i \in [N]} \ell\left(f(u, U; \cdot); (x^{(i)}, y^{(i)})\right)$ and let $\widehat{L}_{\mathcal{S}}(u, U)$ denote the average over some subset $\mathcal{S}$ of examples instead of the entire dataset.

We consider a regularized training objective $\widehat{L}_\lambda(u, U) = \widehat{L}(u, U) + \frac{\lambda}{2}\|U\|_F^2$. For the simplicity of derivation, the second layer weight vector $u$ is random initialized and fixed throughout this paper. Thus with slight abuse of notation the training objective can be written as $\widehat{L}_\lambda(U) = \widehat{L}(u, U) + \frac{\lambda}{2}\|U\|_F^2$.

**Notations.** Here we collect additional notations that will be useful throughout our proofs. The symbol $\oplus$ will refer to the symmetric difference of two sets or two binary vectors. The symbol $\backslash$ refers to the set difference. Let us define $\mathcal{M}_1$ to be the set of all $i \in [N]$ such that $x_1^{(i)} \neq 0$, let $\bar{\mathcal{M}}_1 = [N]\backslash\mathcal{M}_1$. Let $\mathcal{M}_2$ to be the set of all $i \in [N]$ such that $x_2^{(i)} \neq 0$, let $\bar{\mathcal{M}}_2 = [N]\backslash\mathcal{M}_2$. We define $q = \frac{|\mathcal{M}_1|}{N}$ and $p = \frac{|\mathcal{M}_2|}{N}$ to be the empirical fraction of data containing patterns only from $\mathcal{Q}$ and $\mathcal{P}$, respectively. We will sometimes use $\widehat{\mathbb{E}}$ to denote an empirical expectation over the training samples. For a vector or matrix $v$, we use $\text{supp}(v)$ to denote the set of indices of the non-zero entries of $v$. For $U \in \mathbb{R}^{m \times d}$ and $R \subset [m]$, let $U^R$ be the restriction of $U$ to the subset of rows indexed by $R$. We use $[U]_i$ to denote the $i$-th row of $U$ as a row vector in $\mathbb{R}^{1 \times d}$. Let the symbol $\odot$ denote the element-wise product between two vectors or matrices. The notation $I_{n \times n}$ will denote the $n \times n$ identity matrix, and $\mathbf{1}$ the all 1's vector where dimension will be clear from context. We define "with high probability" to mean with probability at least $1 - e^{-C \log^2(d)}$ for a sufficiently large constant $C$. $\tilde{O}, \tilde{\Omega}$ will be used to hide polylog factors of $d$.

## 3 Main Results

The training algorithm that we consider is stochastic gradient descent with spherical Gaussian noise. We remark that we analyze this algorithm as a simplification of the minibatch SGD noise encountered when training real-world networks. There are a number of works theoretically characterizing this particular noise distribution [19, 18, 38], and we leave analysis of this setting to future work.

We initialize $U_0$ to have i.i.d. entries from a Gaussian distribution with variance $\tau_0^2$, and at each iteration of gradient descent we add spherical Gaussian noise with coordinate-wise variance $\tau_\xi^2$ to the gradient updates. That is, the learning algorithm for the model is

$$U_0 \sim \mathcal{N}(0, \tau_0^2 I_{m \times m} \otimes I_{d \times d})$$
$$U_{t+1} = U_t - \gamma_t \nabla_U(\widehat{L}_\lambda(u, U_t) + \xi_t) = (1 - \gamma_t\lambda)U_t - \gamma_t(\nabla_U \widehat{L}(u, U_t) + \xi_t) \quad (3.1)$$
$$\text{where } \xi_t \sim \mathcal{N}(0, \tau_\xi^2 I_{m \times m} \otimes I_{d \times d}) \quad (3.2)$$

where $\gamma_t$ denotes the learning rate at time $t$. We will analyze two algorithms:

> **Algorithm 1** (L-S): The learning rate is $\eta_1$ for $t_0$ iterations until the training loss drops below the threshold $\varepsilon_1 + q \log 2$. Then we anneal the learning rate to $\gamma_t = \eta_2$ (which is assumed to be much smaller than $\eta_1$) and run until the training loss drops to $\varepsilon_2$.

> **Algorithm 2** (S): We used a fixed learning rate of $\eta_2$ and stop at training loss $\varepsilon_2' \leq \varepsilon_2$.

For the convenience of the analysis, we make the following assumption that we choose $\tau_0$ in a way such that the contribution of the noises in the system stabilize at the initialization:[2]

**Assumption 3.1.** *After fixing $\lambda$ and $\tau_\xi$, we choose initialization $\tau_0$ and large learning rate $\eta_1$ so that*

$$(1 - \eta_1\lambda)^2\tau_0^2 + \eta_1^2\tau_\xi^2 = \tau_0^2 \qquad (3.3)$$

*As a technical assumption for our proofs, we will also require $\eta_1 \lesssim \varepsilon_1$.*

We also require sufficient over-parametrization.

**Assumption 3.2** (Over-parameterization). *We assume throughout the paper that $\tau_0 = 1/\text{poly}\left(\frac{d}{\varepsilon}\right)$ and $m \geq \text{poly}\left(\frac{d}{\varepsilon\tau_0}\right)$ where* poly *is a sufficiently large constant degree polynomial. We note that we can choose $\tau_0$ arbitrarily small, so long as it is fixed before we choose $m$.*

As we will see soon, the precise relation between $N, d$ implies that the level of over-parameterization is polynomial in $N, \epsilon$, which fits with the conditions assumed in prior works, such as [26, 13].

**Assumption 3.3.** *Throughout this paper, we assume the following dependencies between the parameters. We assume that $N, d \to \infty$ with a relationship $\frac{N}{d} = \frac{1}{\kappa^2}$ where $\kappa \in (0, 1)$ is a small value.[3] We set $r = d^{-3/4}$, $p_0 = \kappa^2/2$, and $q_0 = \Theta(1)$. The regularizer will be chosen to be $\lambda = d^{-5/4}$. All of these choices of hyper-parameters can be relaxed, but for simplicity of exposition we only work this setting.*

We note that under our assumptions, for sufficiently large $N$, $p \approx p_0$ and $q \approx q_0$ up to constant multiplicative factors. Thus we will mostly work with $p$ and $q$ (the empirical fractions) in the rest of the paper. We also note that our parameter choice satisfies $(rd)^{-1}, d\lambda, \lambda/r \leq \kappa^{O(1)}$ and $\lambda \leq r^2/(\kappa^2 q^3 p^2)$, which are a few conditions that we frequently use in the technical part of the paper.

Now we present our main theorems regarding the generalization of models trained with the L-S and S algorithms. The final generalization error of the model trained with the L-S algorithm will end up a factor $O(\kappa) = O(p^{1/2})$ smaller than the generalization error of the model trained with S algorithm.

**Theorem 3.4** (Analysis of Algorithm L-S). *Under Assumption 3.1, 3.2, and 3.3, there exists a universal constant $0 < c < 1/16$ such that Algorithm 1 (L-S) with annealing at loss $\varepsilon_1 + q\log 2$ for $\varepsilon_1 \in \left(d^{-c}, \kappa^2 p^2 q^3\right)$ and stopping criterion $\varepsilon_2 = \sqrt{\varepsilon_1/q}$ satisfies the following:*

1. *It anneals the learning rate within $\widetilde{O}\left(\frac{d}{\eta_1\varepsilon_1}\right)$ iterations.*

2. *It stops at at most $t = \widetilde{O}\left(\frac{d}{\eta_1\varepsilon_1} + \frac{1}{\eta_2 r\varepsilon_1^3}\right)$. With probability at least 0.99, the solution $U_t$ has test (classification) error and test loss **at most** $O\left(p\kappa\log\frac{1}{\varepsilon_1}\right)$.*

Roughly, the learning order and generalization of the L-S model is as follows: before annealing the learning rate, the model only learns an effective classifier for $\mathcal{P}$ on the $\approx (1-q)N$ samples in $\mathcal{M}_1$ as the large learning rate creates too much noise to effectively learn $\mathcal{Q}$ (Lemma 4.1 and Lemma 4.2). After the learning rate is annealed, the model memorizes $\mathcal{Q}$ and correctly classifies examples with only a $\mathcal{Q}$ component during test time (formally shown in Lemmas 4.3 and 4.4). For examples with only $\mathcal{P}$ component, the generalization error is (ignoring log factors and other technicalities) $p\sqrt{\frac{d}{N}} = O(p\kappa)$ via standard Rademacher complexity. The full analysis of the L-S algorithm is clarified in Section 4.

**Theorem 3.5** (Lower bound for Algorithm S). *Let $\varepsilon_2$ be chosen in Theorem 3.4. Under Assumption 3.1, 3.2 and 3.3, there exists a universal constant $c > 0$ such that w.h.p, Algorithm 2 with any $\eta_2 \leq \eta_1 d^{-c}$ and any stopping criterion $\varepsilon_2' \in (d^{-c}, \varepsilon_2]$, achieves training loss $\varepsilon_2'$ in at most $\widetilde{O}\left(\frac{d}{\eta_2\varepsilon_2'}\right)$ iterations, and both the test error and the test loss of the obtained solution are **at least** $\Omega(p)$.*

We explain this lower bound as follows: the S algorithm will quickly memorize the $\mathcal{Q}$ component which is low noise and ignore the $\mathcal{P}$ component for the $\approx 1 - p - q$ examples with both $\mathcal{P}$ and $\mathcal{Q}$ components (shown in Lemma 5.2). Thus, it only learns $\mathcal{P}$ on $\approx pN$ examples. It obtains a small margin on these examples and therefore misclassifies a constant fraction of $\mathcal{P}$-only examples at test time. This results in the lower bound of $\Omega(p)$. We formalize the analysis in Section 5.

**Decoupling the Iterates.** It will be fruitful for our analysis to separately consider the gradient signal and Gaussian noise components of the weight matrix $U_t$. We will decompose the weight matrix $U_t$ as follows: $U_t = \overline{U}_t + \widetilde{U}_t$. In this formula, $\overline{U}_t$ denotes the signals from all the gradient updates accumulated over time, and $\widetilde{U}_t$ refers to the noise accumulated over time:

$$
\begin{aligned}
\overline{U}_t &= -\sum_{s=1}^{t} \gamma_{s-1} \left( \prod_{i=s}^{t-1}(1 - \gamma_i \lambda) \right) \nabla \widehat{L}(U_{s-1}) \\
\widetilde{U}_t &= \left( \prod_{i=0}^{t-1}(1 - \gamma_i \lambda) \right) U_0 - \sum_{s=1}^{t} \gamma_{s-1} \left( \prod_{i=s}^{t-1}(1 - \gamma_i \lambda) \right) \xi_{s-1}
\end{aligned}
\tag{3.4}
$$

Note that when the learning rate $\gamma_t$ is always $\eta$, the formula simplifies to $\overline{U}_t = \sum_{s=1}^{t} \eta(1 - \eta\lambda)^{t-s}\nabla\widehat{L}(U_{s-1})$ and $\widetilde{U}_t = (1 - \eta\lambda)^t U_0 + \sum_{s=1}^{t} \eta(1 - \eta\lambda)^{t-s}\xi_{s-1}$. The decoupling and our particular choice of initialization satisfies that the noise updates in the system stabilize at initialization, so the marginal distribution of $\widetilde{U}_t$ is always the same as the initialization. Another nice aspect of the signal-noise decomposition is as follows: we use tools from [6] to show that if the signal term $\overline{U}$ is small, then using only the noise component $\widetilde{U}$ to compute the activations roughly preserves the output of the network. This facilitates our analysis of the network dynamics. See Section A.1 for full details.

**Decomposition of Network Outputs.** For convenience, we will explicitly decompose the model prediction at each time into two components, each of which operates on one pattern: we have $N_{U_t}(u, U_t; x) = g_t(x) + r_t(x)$,

$$
\text{where } g_t(x) = g_t(x_2) \triangleq N_{V_t}(v, V_t; x) = N_{V_t}(v, V_t; x_2)
\tag{3.5}
$$

$$
r_t(x) = r_t(x_1) \triangleq N_{W_t}(w, W_t; x) = N_{W_t}(w, W_t; x_1)
\tag{3.6}
$$

In other words, the network $g_t$ acts on the $\mathcal{Q}$ component of examples, and the network $r_t$ acts on the $\mathcal{P}$ component of examples.

## 4 Characterization of Algorithm 1 (L-S)

We characterize the behavior of algorithm L-S with large initial learning rate. We provide proof sketches in Section B.1 with full proofs in Section D.

**Phase I: initial learning rate $\eta_1$.** The following lemma bounds the rate of convergence to the point where the loss gets annealed. It also bounds the total gradient signal accumulated by this point.

**Lemma 4.1.** *In the setting of Theorem 3.4, at some time step $t_0 \leq \widetilde{O}\left(\frac{d}{\eta_1 \varepsilon_1}\right)$, the training loss $\widehat{L}(U_{t_0})$ becomes smaller than $q\log 2 + \epsilon_1$. Moreover, we have $\|\overline{U}_{t_0}\|_F^2 = O\left(d\log^2 \frac{1}{\varepsilon_1}\right)$.*

Our proof of Lemma 4.1 views the SGD dynamics as optimization with respect to the neural tangent kernel induced by the activation patterns where the kernel is rapidly changing due to the noise terms $\xi$. This is in contrast to the standard NTK regime, where the activation patterns are assumed to be stable [13, 26]. Our analysis extends the NTK techniques to deal with a sequence of changing kernels which share a common optimal classifier (see Section B.1 and Theorem B.2 for additional details).

The next lemma says that with large initial learning rate, the function $g_t$ does not learn anything meaningful for the $\mathcal{Q}$ component before the $\frac{1}{\eta_1 \lambda}$-timestep. Note that by our choice of parameters $1/\lambda \gg d$ and Lemma 4.1, we anneal at the time step $\widetilde{O}\left(\frac{d}{\eta_1 \varepsilon_1}\right) \leq \frac{1}{\eta_1 \lambda}$. Therefore, the function has not learned anything meaningful about the memorizable pattern on distribution $\mathcal{Q}$ before we anneal.

**Lemma 4.2.** *In the setting of Theorem 3.4, w.h.p., for every $t \leq \frac{1}{\eta_1 \lambda}$,*

$$
|g_t(z + \zeta) + g_t(z - \zeta) - 2g_t(z)| \leq \widetilde{O}\left(\frac{r^2}{\lambda}\right) = \widetilde{O}(d^{-1/4})
\tag{4.1}
$$

**Phase II: after annealing the learning rate to $\eta_2$.** After iteration $t_0$, we decrease the learning rate to $\eta_2$. The following lemma bounds how fast the loss converges after annealing.

**Lemma 4.3.** *In the setting of Theorem 3.4, there exists $t = \widetilde{O}\left(\frac{1}{\varepsilon_1^3 \eta_2 r}\right)$, such that after $t_0 + t$ iterations, we have that*

$$\widehat{L}(U_t) = O\left(\sqrt{\varepsilon_1/q}\right)$$

*Moreover, $\|\overline{U}_{t_0+t} - \overline{U}_{t_0}\|_F^2 \leq \widetilde{O}\left(\frac{1}{\varepsilon_1^2 r}\right) \leq O(d)$.*

The following lemma bounds the training loss on the example subsets $\mathcal{M}_1, \bar{\mathcal{M}}_1$.

**Lemma 4.4.** *In the setting of Lemma 4.3 using the same $t = \widetilde{O}\left(\frac{1}{\varepsilon_1^3 \eta_2 r}\right)$, the average training losses on the subsets $\mathcal{M}_1$ and $\bar{\mathcal{M}}_1$ are both good in the sense that*

$$\widehat{L}_{\mathcal{M}_1}(r_{t_0+t}) = O(\sqrt{\varepsilon_1/q}) \text{ and } \widehat{L}_{\bar{\mathcal{M}}_1}(g_{t_0+t}) = O(\sqrt{\varepsilon_1/q^3}) \tag{4.2}$$

Intuitively, low training loss of $g_{t_0+t}$ on $\bar{\mathcal{M}}_1$ immediately implies good generalization on examples containing patterns from $\mathcal{Q}$. Meanwhile, the classifier for $\mathcal{P}$, $r_{t_0+t}$, has low loss on $(1-q)N$ examples. Then the test error bound follows from standard Rademacher complexity tools applied to these $(1-q)N$ examples.

## 5 Characterization of Algorithm 2 (S)

We present our small learning rate lemmas, with proofs sketches in Section B.2 and full proofs in Section E.

**Training loss convergence.** The below lemma shows that the algorithm will converge to small training error too quickly. In particular, the norm of $W_t$ is not large enough to produce a large margin solution for those $x$ such that $x_2 = 0$.

**Lemma 5.1.** *In the setting of Theorem 3.5, there exists a time $t' = \tilde{O}\left(\frac{1}{\eta_2 \varepsilon_2'^3 r}\right)$ such that $\widehat{L}_{\mathcal{M}_2}(U_{t'}) \leq \varepsilon_2'$. Moreover, there exists $t$ with $t = \tilde{O}\left(\frac{1}{\eta_2 \varepsilon_2'^3 r} + \frac{Np}{\eta_2 \varepsilon_2'}\right)$ such that $\widehat{L}(U_t) \leq \varepsilon_2'$ after $t$ iterations. Moreover, we have that $\|\overline{U}_t\|_F^2 \leq \tilde{O}\left(\frac{1}{\varepsilon_2'^2 r} + Np\right)$.*

**Lower bound on the generalization error.** The following important lemma states that our classifier for $\mathcal{P}$ does not learn much from the examples in $\mathcal{M}_2$. Intuitively, under a small learning rate, the classifier will already learn so quickly from the $\mathcal{Q}$ component of these examples that it will not learn from the $\mathcal{P}$ component of examples in $\mathcal{M}_1 \cap \mathcal{M}_2$. We make this precise by showing that the magnitude of the gradients on $\mathcal{M}_2$ is small.

**Lemma 5.2.** *In the setting of theorem 3.5, let*

$$\overline{W}_t^{(2)} = \frac{1}{N}\eta_2 \sum_{s \leq t}(1 - \eta_2\lambda)^{t-s} \sum_{i \in \mathcal{M}_2} \nabla_W \widehat{L}_{\{i\}}(U_s) \tag{5.1}$$

*be the (accumulated) gradient of the weight $W$, restricted to the subset $\mathcal{M}_2$. Then, for every $t = O\left(d/\eta_2 \varepsilon_2'\right)$, we have: $\left\|\overline{W}_t^{(2)}\right\|_F \leq \tilde{O}\left(d^{15/32}/\varepsilon_2'^2\right)$. For notation simplicity, we will define $\varepsilon_3 = d^{-1/32}\frac{1}{\varepsilon_2'^2}$. Then, $\left\|\overline{W}_t^{(2)}\right\|_F \leq \tilde{O}\left(\sqrt{d}\varepsilon_3\right)$.*

The above lemma implies that $W$ does not learn much from examples in $\mathcal{M}_2$, and therefore must overfit to the $pN$ examples in $\bar{\mathcal{M}}_2$. As $pN \leq d/2$ by our choice of parameters, we will not have enough samples to learn the $d$-dimensional distribution $\mathcal{P}$. The following lemma formalizes the intuition that the margin will be poor on samples from $\mathcal{P}$.

**Lemma 5.3.** *There exists $\alpha \in \mathbb{R}^d$ such that $\alpha \in span\{x_1^{(i)}\}_{i \in \bar{\mathcal{M}}_2}$ and $\|\alpha\|_2 = \tilde{\Omega}(\sqrt{Np})$ such that w.h.p. over a randomly chosen $x_1$, we have that*

$$r_t(x_1) - r_t(-x_1) = 2\langle \alpha, x_1 \rangle \pm \tilde{O}(\varepsilon_3) \tag{5.2}$$

As the margin is poor, the predictions will be heavily influenced by noise. We use this intuition to prove the classification lower bound for Theorem 3.5.

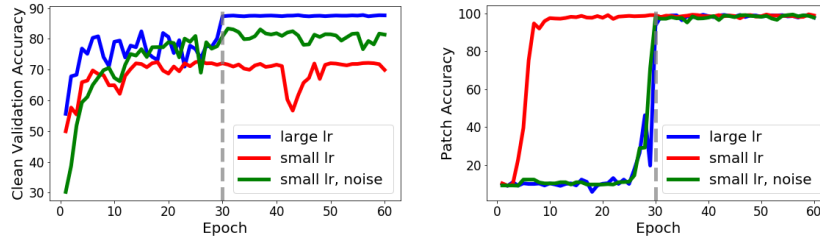

Figure 3: Accuracy vs. epoch on patch-augmented CIFAR-10. The gray line indicates annealing of activation noise and learning rate. **Left:** Clean validation set. **Right:** Images containing only the patch.

## 6 Experiments

Our theory suggests that adding noise to the network could be an effective strategy to regularize a small learning rate in practice. We test this empirically by adding small Gaussian noise during training *before* every activation layer in a WideResNet16 [42] architecture, as our analysis highlights pre-activation noise as a key regularization mechanism of SGD. The noise level is annealed over time. We demonstrate on CIFAR-10 images without data augmentation that this regularization can indeed counteract the negative effects of small learning rate, as we report a 4.72% increase in validation accuracy when adding noise to a small learning rate. Full details are in Section H.1.

We will also empirically demonstrate that the choice of large vs. small initial learning rate can indeed invert the learning order of different example types. We add a memorizable $7 \times 7$ pixel patch to a subset of CIFAR-10 images following the scenario presented in (1.1), such that around 20% of images have no patch, 16% of images contain only a patch, and 64% contain both CIFAR-10 data and patch. We generate the patches so that they are not easily separable, as in our constructed $\mathcal{Q}$, but they are low in variation and therefore easy to memorize. Precise details on producing the data, including a visualization of the patch, are in Section H.2. We train on the modified dataset using WideResNet16 using 3 methods: large learning rate with annealing at the 30th epoch, small initial learning rate, and small learning rate with noise annealed at the 30th epoch.

Figure 3 depicts the validation accuracy vs. epoch on clean (no patch) and patch-only images. From the plots, it is apparent that the small learning rate picks up the signal in the patch very quickly, whereas the other two methods only memorize the patch after annealing.

From the validation accuracy on clean images, we can deduce that the small learning rate method is indeed learning the CIFAR images using a small fraction of all the available data, as the validation accuracy of a small LR model when training on the full dataset is around 83%, but the validation on clean data after training with the patch is 70%. We provide additional arguments in Section H.2.

## 7 Conclusion

In this work, we show that the order in which a neural net learns to fit different types of patterns plays a crucial role in generalization. To demonstrate this, we construct a distribution on which models trained with large learning rates generalize provably better than those trained with small learning rates due to learning order. Our analysis reveals that more SGD noise, or larger learning rate, biases the model towards learning "generalizing" kernels rather than "memorizing" kernels. We confirm on articifially modified CIFAR-10 data that the scale of the learning rate can indeed influence learning order and generalization. Inspired by these findings, we propose a mitigation strategy that injects noise before the activations and works both theoretically for our construction and empirically. The design of better algorithms for regularizing learning order is an exciting question for future work.

## Acknowledgements

CW acknowledges support from a NSF Graduate Research Fellowship.

## Footnotes

[1]While these papers are framed as a study of large-batch training, a number of them explicitly acknowledge the connection between large batch size and small learning rate.

[2]Let $\tau_0'$ be the solution to (3.3) holding $\tau_\xi, \eta_1, \lambda$ fixed. If the standard deviation of the initialization is chosen to be smaller than $\tau_0'$, then standard deviation of the noise will grow to $\tau_0'$. Otherwise if the initialization is chosen to be larger, the contribution of the noise will decrease to the level of $\tau_0'$ due to regularization. In typical analysis of SGD with spherical noises, often as long as either the noise or the learning rate is small enough, the proof goes through. However, here we will make explicit use of the large learning rate or the large noise to show better generalization performance.

[3]Or in a non-asymptotic language, we assume that $N, d$ are sufficiently large compared to $\kappa$: $N, d \gg \text{poly}(\kappa)$

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
