[Supplementary Material]

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

# A  Basic Properties and Toolbox

In this section, we collect a few basic properties of the neural networks we are studying. In section G, we provide two lemmas on Gaussian random variables and perturbation theory of the matrices.

**Proposition A.1.**

$$[\nabla\widehat{L}(U)]_i = \widehat{\mathbb{E}}\left[\ell'(f(u,U;(x,y)))\mathbb{1}([U]_ix)x\right] \tag{A.1}$$

**Proposition A.2.** *Let $[\nabla\widehat{L}(U)]_i$ be the $i$-th row of $\nabla\widehat{L}(U)$. We have that $\|[\nabla\widehat{L}(U)]_i\|_2 \lesssim 1/\sqrt{m}$.*

**Proposition A.3.** *For any $t$, if $\gamma_s = \eta$ for every $s \leq t$, then we have that $\|[\overline{U}_t]_i\|_2 \lesssim \min\{\frac{1}{\sqrt{m}\lambda}, \eta t/\sqrt{m}\}$ and $\|\overline{U}_t\|_F \lesssim \frac{1}{\lambda}$.*

*Proof.* By equation (3.4) and Proposition A.2, we have that

$$\|[\overline{U}_t]_i\|_2 = \sum_s \eta(1-\eta\lambda)^{t-s}\|[\nabla\widehat{L}(U_s)]_i\|_2 \leq \frac{1}{\sqrt{m}}\sum_s \eta(1-\eta\lambda)^{t-s} \lesssim \min\left\{\frac{1}{\sqrt{m}\lambda}, \frac{\eta t}{\sqrt{m}}\right\}$$

$\square$

**Proposition A.4.** *Suppose that matrix $\widetilde{U} \in \mathbb{R}^{m\times d}$ is a random variable whose columns have i.i.d distribution $\mathcal{N}(0, \tau^2 I_{m\times m})$ and $u \in \mathbb{R}^m$ such that each entry of $u$ is i.i.d. uniform in $\{-m^{-1/2}, m^{1/2}\}$. For every $x$, we have that w.h.p. over the randomness of $\widetilde{U}$ and $u$ that*

$$\left|N_{\widetilde{U}}(u,\widetilde{U};x)\right| \lesssim \tau\|x\|_2 \log d \tag{A.2}$$

*Proof of Proposition A.4.* By definition, we have that

$$N_{\widetilde{U}}(u,\widetilde{U};x) = \sum_{i\in[m]} u_i[[\widetilde{U}]_ix]_+ \tag{A.3}$$

By definition, $\widetilde{U} \in \mathbb{R}^{m\times d}$ where each entry is i.i.d. $\mathcal{N}(0,\tau^2)$, which implies that when $m \geq d$, w.h.p. $\|\widetilde{U}\|_2 = O(\tau\sqrt{m})$.

Hence $\|[\widetilde{U}x]_+\|_2 \leq \|\widetilde{U}x\|_2 \lesssim \tau\sqrt{m}\|x\|_2$. Now, since each $u_i$ is i.i.d. uniform $\{-m^{-1/2}, m^{1/2}\}$, using the randomness of $u_i$ we know that w.h.p.

$$\left|\sum_{i\in[m]} u_i[[\widetilde{U}]_ix]_+\right| \lesssim \frac{\log m}{\sqrt{m}}\|[\widetilde{U}x]_+\|_2 \lesssim \tau\|x\|_2 \log d \tag{A.4}$$

$\square$

**Proposition A.5.** *Under the same setting as Lemma A.8, we will also have w.h.p over the randomness of $\widetilde{U}$ and $u$, $\forall \overline{U} \in \mathbb{R}^{d\times m}$,*

$$\left|N_U(u,\widetilde{U};x) - N_{\widetilde{U}}(u,\widetilde{U};x)\right| \lesssim B\|\overline{U}\|_F^{5/3}\tau^{-2/3}m^{-1/6} \tag{A.5}$$

*Thus, it also follows that*

$$|N_U(u,\widetilde{U};x)| \lesssim B\|\overline{U}\|_F^{5/3}\tau^{-2/3}m^{-1/6} + \tau B \log d \tag{A.6}$$

*Proof.* We know that for every $i$ where $\mathbb{1}([U]_ix) \neq \mathbb{1}([\widetilde{U}]_ix)$, it holds that $|[\widetilde{U}]_ix| \leq |[\overline{U}]_ix|$. This implies that

$$\left|N_U(u,\widetilde{U};x) - N_{\widetilde{U}}(u,\widetilde{U};x)\right| \leq \frac{1}{\sqrt{m}}\sum_{i\in[m]}|\mathbb{1}([U]_ix) - \mathbb{1}([\widetilde{U}]_ix)||[\overline{U}]_ix| \tag{A.7}$$

$$\leq \frac{1}{\sqrt{m}}\|\mathbb{1}(Ux) - \mathbb{1}(\widetilde{U}x)\|_1 \max_i |[\overline{U}]_ix| \tag{A.8}$$

$$\lesssim B\|\overline{U}\|_F^{4/3}\tau^{-4/3}m^{1/6}\max_i\|[\overline{U}]_i\|_2 \tag{A.9}$$

Here in the last inequality we applied Lemma A.8. The second statement follows from Proposition A.4 and triangle inequality. $\square$

We have the following Rademacher complexity bound:

**Lemma A.6** (Lemma G5 and 5.9 of [3]). *Let $U = \overline{U} + \widetilde{U}$, where $\widetilde{U} \in \mathbb{R}^{m \times d}$ is a random variable whose columns have i.i.d distribution $\mathcal{N}(0, \tau_0^2 I_{m \times m})$ and $u \in \mathbb{R}^m$ such that each entry of $u$ is i.i.d. uniform in $\{-m^{-1/2}, m^{1/2}\}$. W.h.p. over the samples $\{x^{(i)}\}$ and the randomness of $u, \widetilde{U}$, we have that for every $\rho \in [0, 1/\lambda]$:*

$$\mathcal{R} := \frac{1}{\sqrt{N}} \sum_{i \in [N]} \mathbb{E}_\sigma \left[ \left\| \sup_{\|\bar{U}\|_F^2 \leq \rho^2} \sigma_i N_U(u, \bar{U}; x^{(i)}) \right\| \right] \leq O(\rho + \varepsilon_s) \tag{A.10}$$

## A.1 Preliminaries on Decoupling the Iterates

In this section, we collect useful statements which will help with decoupling the signal $\overline{U}$ from the noise $\widetilde{U}$ in our analysis. First, we observe that if the noise updates in the system stabilize at initialization, the marginal distribution of $U_t$ is always the same as the initialization.

**Proposition A.7.** *Under Assumption 3.1, suppose we run Algorithm 1. Then for any $t$ before annealing the learning rate, $\widetilde{U}_t$ has marginal distribution $\mathcal{N}(0, \tau_0^2 I_{m \times m} \otimes I_{d \times d})$. In other words, each entry of $\widetilde{U}_t$ follows $\mathcal{N}(0, \tau_0^2)$ and they are independent with each others.*

One nice aspect of the signal-noise decomposition is as follows: we use tools from [6] to show that if the signal term $\overline{U}$ is small, then using only the noise component $\widetilde{U}$ to compute the activations roughly preserves the output of the network. This facilitates our analysis of the network dynamics.

**Lemma A.8.** *[Lemma 5.2 of [6]] Let $x \in \mathbb{R}^d$ be a fixed example with $\|x\|_2 \leq B$. For every $\tau > 0$, let $U = \overline{U} + \widetilde{U}$ where $\widetilde{U} \in \mathbb{R}^{m \times d}$ is a random variable whose columns have i.i.d distribution $\mathcal{N}(0, \tau^2 I_{m \times m})$ and $u \in \mathbb{R}^m$ such that each entry of $u$ is i.i.d. uniform in $\{-m^{-1/2}, m^{1/2}\}$. We have that, w.h.p over the randomness of $\widetilde{U}$ and $u$, $\forall \overline{U} \in \mathbb{R}^{d \times m}$,*

$$\left| N_U(u, \overline{U}; x) - N_{\widetilde{U}}(u, \overline{U}; x) \right| \lesssim B \|\overline{U}\|_F \tau^{-2} m^{-1/6} \tag{A.11}$$

*Moreover, we have that $\|\mathbb{1}(Ux) - \mathbb{1}(\widetilde{U}x)\|_1 \lesssim \|\overline{U}\|_F^{4/3} \tau^{-4/3} m^{2/3}$.*

As we will often apply (A.11) with $\|\overline{U}\|_F \lesssim \frac{1}{\lambda}$, for notational simplicity we denote throughout the paper $\varepsilon_s = \left(\frac{1}{\lambda \tau_0}\right)^{4/3} m^{-1/3}$. By our choice of $m \geq \text{poly}(d/\tau_0)$ we know that $\varepsilon_s \leq d^{-\Theta(1)}$.

# B Proof Sketches

## B.1 Proof Sketches for Large Learning Rate

We first introduce notations that will be useful in these proofs. We will explicitly decouple the noise in the weights from the signal by abstracting the loss as a function of only the signal portion $\overline{U}_t$ of the weights. Let us define the following:

$$f_t(B; x) = N_{U_t}(u, B + \widetilde{U}_t; x) \tag{B.1}$$

Moreover, we define

$$K_t(B) \triangleq \frac{1}{N} \sum_{i=1}^{N} \ell(f_t(B; \cdot); (x^{(i)}, y^{(i)})) \tag{B.2}$$

By definition, we know that

$$L_t = \widehat{L}(U_t) = K_t(\overline{U}_t) \tag{B.3}$$

$$\nabla_U \widehat{L}(U_t) = \nabla K_t(\overline{U}_t) \tag{B.4}$$

Now the proof of Lemma 4.1 relies on the following two results, which we state below and prove in Section D.1. The first says that there is a common target for the signal part of the network that is a good solution for all of the $K_t$.

**Lemma B.1.** *In the setting of Lemma 4.1, there exists a solution $U^\star$ satisfying a) $\|U^\star\|_F^2 \leq O\left(d\log^2\frac{1}{\varepsilon_1}\right)$ and b) for every $t \geq 0$*

$$K_t(U^\star) \leq q\log 2 + \epsilon_1/2 \tag{B.5}$$

Now the second statement is a general one proving that gradient descent on a sequence of convex, but changing, functions will still find a optimum provided these functions share the same solution.

**Theorem B.2.** *Suppose $K_1, \ldots, K_T : \mathbb{R}^d \to \mathbb{R}^*$ is a sequence of differentiable convex functions satisfying*

1. *$\exists z^\star$ and a constant $c^\star \in \mathbb{R}^*$ such that $K_t(z^\star) \leq c^\star, \forall t = 1, \ldots, T$, and that $\|z_0 - z^\star\|_2 \leq R, \|z^\star\|_2 \leq R$.*

2. *$K_t$'s are L-Lipschitz, i.e., $\|\nabla K_t(z)\|_2 \leq L, \forall z, t$*

*Let $K_t^\lambda(z) \triangleq K_t(z) + \frac{\lambda}{2}\|z\|_2^2$. Consider the following iterative algorithm that starts from $z_0 \in \mathbb{R}^d$,*

$$\forall t \geq 0, \ z_{t+1} = z_t - \eta\nabla K_t^\lambda(z_t) \tag{B.6}$$

*For every $\mu > 0$, we have that for $\lambda R^2 \leq \frac{1}{100}\mu$ and $\eta \leq \frac{\mu}{100(\lambda^2 R^2 + L^2)}, \eta T > \frac{R^2}{\mu}$, there is a $t^\star \in [T]$ such that:*

$$K_{t^\star}(z_{t^\star}) \leq c^\star + \mu \tag{B.7}$$

*Furthermore, the iterates satisfy $\|z_t - z^\star\|_2 \leq R$ for all $t \leq t^\star$.*

Combining these two statements leads to the proof of Lemma 4.1.

*Proof of Lemma 4.1.* We can apply Theorem B.2 with $K_t$ defined in (B.2) and $z^\star = U^\star$ defined in Lemma B.1, using $R = O\left(d\log^2\frac{1}{\varepsilon_1}\right)$. We note that $\eta_1$ satisfies the conditions of Theorem B.2 by our parameter choices, which completes the proof. $\qquad\square$

To prove Lemma 4.2, we will essentially argue in Section D.2 that the change in activations caused by the noise will prevent the model from learning $\mathcal{Q}$ with a large learning rate. This is because the examples in $\mathcal{Q}$ require a very specific configuration of activation patterns to learn correctly, and the noise will prevent the model from maintaining this configuration.

Now after we anneal the learning rate, in order to conclude Lemmas 4.3 and 4.4, the following must hold: 1) the network learns the $\mathcal{Q}$ component of the distribution and 2) the network does not forget the $\mathcal{P}$ component that it previously learned. To prove the latter, we rely on the following lemma stating that the activations do not change much with a small learning rate:

**Lemma B.3.** *The activation patterns do not change much after annealing the learning rate: for every $t_0, t \leq \frac{1}{\eta_2\lambda}$, for any $x$ and for any row $[U_t]_i$ of the weight matrix $U$, we have that*

$$\|\mathbb{1}([U_{t_0+t}]x) - \mathbb{1}([U_{t_0}]x)\|_1 \lesssim \sqrt{\frac{\eta_2}{\eta_1}}m + \varepsilon_s m \tag{B.8}$$

*Moreover, for all $i \in [m]$, $\left\|[\overline{U}_t]_i\right\|_2 \leq \frac{1}{\lambda\sqrt{m}}$, it holds that w.h.p. for every $x$:*

$$\left|N_{U_{t_0+t}}(u, U_{t_0+t}; x) - N_{U_{t_0}}(u, \overline{U}_{t_0+t}; x)\right| \lesssim \frac{1}{\lambda} \times \left(\sqrt{\frac{\eta_2}{\eta_1}} + \varepsilon_s\right) + \tau_0\log d \tag{B.9}$$

We prove the above lemma in Section D.3. Now to complete the proof of Lemma 4.3, we will construct a target solution for all timesteps after annealing the learning rate based on the activations at time $t_0$ (as they do not change by much in subsequent time steps because of Lemma B.3) and reapply Theorem B.2. Finally, to prove Lemma 4.4, we use the fact that the $W_t$ component of the solution does not change by much, and therefore the loss on $\mathcal{M}_1$ is still low.

## B.2 Proof Sketches for Small Learning Rate

The proof of Lemma 5.1 proceeds similarly as the proof of Lemma 4.3: we will show the existence of a target solution of $K_t$ for all iterations, and use Theorem B.2 to prove convergence to this target solution.

Now to sketch the proof of Lemma 5.2, we will first define the following notation: define $\ell'_{j,t} = \ell'(-y^{(j)}N_{U_t}(u, U_t; x^{(j)}))$ to be the derivative of the loss at time $t$ on example $j$. Let $\rho_t$ be the average of the absolute value of the derivative.

$$\rho_t = \frac{1}{N} \sum_{j \in \mathcal{M}_2} |\ell'_{i,t}| \tag{B.10}$$

The next two statements argue that $\rho_t$ can be large only in a limited number of time steps. As the training loss converges quickly with small learning rate, this will be used to argue that the $\mathcal{P}$ components of examples in $\mathcal{M}_2$ provide a very limited signal to $W_t$. The proofs of these statements are in Section E.2.

We first show the following lemma that says that if $\rho_t$ is large (which means the loss is large as well), then the total gradient norm has to be big. This lemma holds because there is little noise in the $\mathcal{Q}$ component of the distribution, and therefore the gradient of $V_t$ will be large if $\rho_t$ is large.

**Lemma B.4.** *For every $t \leq \frac{1}{\eta_2 \lambda}$, we have that if $\rho_t = \Omega\left(\frac{1}{N}\right)$, then w.h.p.*

$$\|\nabla \widehat{L}(U_t)\|_F^2 \geq \Omega\left(r\rho_t^4\right) \tag{B.11}$$

Now we use the above lemma to bound the number of times when $\rho_t$ is large.

**Proposition B.5.** *In the setting of Lemma 5.2, let $\mathcal{T}$ be the set of iterations where $\rho_t \geq \varepsilon_2'^2 \varepsilon_3^2$, where $\varepsilon_3$ is defined in Lemma 5.2. Then w.h.p, $|\mathcal{T}| \lesssim \frac{1}{r\varepsilon_2'^8 \varepsilon_3^8 \eta_2}$.*

Now if $\rho_t$ is small, the gradient accumulated on $W_t$ from examples in $\mathcal{M}_2$ must be small. We formalize this argument in our proof of Lemma 5.2 in Section E.2.

Lemma 5.3 will then follow by explicitly decomposing $\overline{W}_t$ into a component in $\text{span}\{x_1^{(i)}\}_{i \in \bar{\mathcal{M}}_2}$ and some remainder, which is shown to be small by Lemma 5.2. This is presented in the below lemma, which is proved in Section E.3.

**Lemma B.6.** *There exists real numbers $\{\alpha_k\}_{k \in \bar{\mathcal{M}}_2}$ such that for every $j \in [m]$, we have*

$$[\overline{W}_t]_j = w_j \sum_{k \in \bar{\mathcal{M}}_2} \alpha_k x_1^{(k)} \mathbb{1}([W_0]_j x_1^{(k)}) + [\overline{W}_t']_j$$

*with $\|\overline{W}_t'\|_F \leq \widetilde{O}\left(\varepsilon_3 \sqrt{d}\right)$.*

This allows us to conclude Lemma 5.3 via computations carried out in Section E.3.

Finally, to complete the proof of Theorem 3.5, we will argue in Section C.2 that a classifier $r_t$ of the form given by (5.2) cannot have small generalization error because it will be too heavily influenced by the noise in $x_1$.

# C  Proof of Main Theorems

## C.1  Proof of Theorem 3.4

We start with the following lemma that shows that if $g$ has small training error on $\bar{\mathcal{M}}_1$, then the output of $g$ on $x_2$ is large compared to $\|x_2\|$. This is because for the loss to be low, $g$ must have a good margin on $x_2$. However, as the norm of $x_2$ is roughly uniform in $[0, 1]$, the examples with small norm will force $g$ to have larger output.

**Lemma C.1** (Signal of $g$). *W.h.p. for every $t \geq 0$ and every $\delta \geq \frac{1}{\sqrt{qN}}$, as long as $\widehat{L}_{\bar{\mathcal{M}}_1}(g_{t_0+t}) \leq \delta$, we have that: for every $(x, y)$,*

$$y g_{t_0+t}(x_2) \gtrsim \frac{\|x\|_2}{\delta} \tag{C.1}$$

*Proof of Lemma C.1.* We use $\bar{\mathcal{M}}_1^{(1)}$ to denote the set of all $x_2^{(i)} \in \bar{\mathcal{M}}_1$ such that $x_2^{(i)} = \alpha(z - \zeta)$. Similarly, we use $\bar{\mathcal{M}}_1^{(2)}$ to denote the set of all $x_2^{(i)} \in \bar{\mathcal{M}}_1$ such that $x_2^{(i)} = \alpha(z + \zeta)$, and use $\bar{\mathcal{M}}_1^{(3)}$ to denote the set of all $x_2^{(i)} \in \bar{\mathcal{M}}_1$ such that $x_2^{(i)} = \alpha z$.

Let $g_{t_0+t}(z + \zeta) = \rho_1, g_{t_0+t}(z - \zeta) = \rho_2, g_{t+t_0}(z) = \rho_3$. By the positive homogeneity of ReLU, we know that for every $x_2 \in \bar{\mathcal{M}}_1^{(i)}$, it holds:

$$g_{t_0+t}(x_2) = \|x_2\|_2 \rho_i \tag{C.2}$$

Since $\widehat{L}_{\bar{\mathcal{M}}_1}(g_{t_0+t}) \le \delta$, it holds that w.h.p. for every $i \in [3]$,

$$\widehat{L}_{\bar{\mathcal{M}}_1^{(i)}}(g_{t_0+t}) \le 4\delta \tag{C.3}$$

Hence, at most $40\delta$ fraction of $x_2 \in \bar{\mathcal{M}}_1^{(i)}$ satisfies $\ell(g_{t_0+t}; (x_2, y)) \ge \frac{1}{10}$. Since $\|x_2\|_2$ is uniform on $[0, 1]$, this implies that as long as $\delta \ge \frac{1}{\sqrt{qN}}$, w.h.p., $80\delta$ fraction of the $x_2 \in \bar{\mathcal{M}}_1^{(i)}$ satisfies that $\|x_2\|_2 = O(\delta)$. Among of these examples, at least $40\delta$ fraction of them should satisfy $\ell(g_{t_0+t}; (x_2, y)) \le \frac{1}{10}$, which implies that $\|x_2\|\rho_i \gtrsim 1$. This implies that $\rho_i \gtrsim 1/\delta$ and the conclusion follows from equality (C.2).

$\square$

Our proof of Theorem 3.4 now amounts to carefully checking that all examples in $\mathcal{M}_2$ are classified correctly, and the classifier $r_{t_0+t}$ will generalize well on $\bar{\mathcal{M}}_2$.

*Proof of Theorem 3.4.* By Lemma 4.4, we know that for $t = \widetilde{O}\left(\frac{1}{\varepsilon_1^3 \eta_2 r}\right)$ we have $\widehat{L}_{\bar{\mathcal{M}}_1}(g_{t_0+t}) = O(\sqrt{\varepsilon_1/q^3})$. Thus applying Lemma C.1, we obtain that as long as $\varepsilon_1 \ge \frac{1}{\sqrt{N}}$ (which is implied by Assumption 3.3)

$$yg_{t_0+t}(x_2) \ge \Omega\left(\frac{\|x\|_2 \sqrt{q^3}}{\sqrt{\varepsilon_1}}\right) \tag{C.4}$$

On the other hand for $r_{t_0+t}$, by Lemma 4.1 and Lemma 4.3 we know that $\|\overline{W}_{t_0+t}\|_F = \widetilde{O}(\sqrt{d})$. Let us define $\mathcal{D}_{x_1}$ to be the marginal distribution of $x_1$. We know that $x_1 = \alpha w^\star + \beta$ where w.h.p. $|\alpha| = \tilde{O}(d^{-1/2})$ and $\beta \sim \mathcal{N}(0, 1/d \times (I - w^\star(w^\star)^\top))$. Hence we have that w.h.p. over $x_1 \sim \mathcal{D}_{x_1}$, $\|\overline{W}_{t_0+t} x_1\|_2 \le |\alpha| \|\overline{W}_{t_0+t}\|_F + d^{-1/2} \|\beta\|_2 \|\overline{W}_{t_0+t}\|_F \le \widetilde{O}(d^{-1/2}) \|\overline{W}_{t_0+t}\|_F \le \widetilde{O}(1)$.
This implies that for $x_1 \sim \mathcal{D}_{x_1}$, applying Lemma A.8 gives us

$$|r_{t_0+t}(x_1)| = |N_{U_{t_0+t}}(u, U_{t_0+t}; x_1)| \tag{C.5}$$
$$\lesssim |N_{U_{t_0+t}}(u, \overline{U}_{t_0+t}; x_1)| + \frac{\varepsilon_s}{\lambda} + \tau_0 \log d \qquad \text{(by Proposition A.5)}$$
$$\lesssim \|u\|_2 \|\overline{W}_{t_0+t} x_1\|_2 + \frac{\varepsilon_s}{\lambda} + \tau_0 \log d = \widetilde{O}(1) \qquad \text{(by our choice of } \tau_0, m)$$

Hence as long as $\|x_2\|_2 = \tilde{\Omega}(\sqrt{\varepsilon_1/q^3} \log \frac{1}{\varepsilon_1})$, it holds that

$$y(r_{t_0+t}(x_1) + g_{t_0+t}(x_2)) = \tilde{\Omega}(1) \times \log \frac{1}{\varepsilon_1} \tag{C.6}$$

This implies that $\ell(r_{t_0+t} + g_{t_0+t}; (x, y)) \le \varepsilon_1$. Otherwise, when $\|x_2\|_2 = \widetilde{O}\left(\sqrt{\varepsilon_1/q^3}\right)$, we also know that w.h.p. $\ell(r_{t_0+t} + g_{t_0+t}; (x, y)) \le \ell(r_{t_0+t}; (x, y)) = \tilde{O}(1)$, since $yg_{t_0+t}(x_2) \ge 0$. On the other hand by Lemma 4.4, we also know that

$$\widehat{L}_{\mathcal{M}_1}(r_{t_0+t}) = O(\sqrt{\varepsilon_1/q}) \tag{C.7}$$

Moreover, applying Lemma A.6 on $r_{t_0+t}$ with $\|W_{t_0+t}\|_F^2 \leq \|W_{t_0}\|_F^2 + \|W_{t_0+t} - W_{t_0}\|_F^2 \lesssim \left(d \log^2 \frac{1}{\varepsilon}\right)$ by Lemma 4.2 and Lemma 4.3, we have that

$$\mathbb{E}_{(x,y)\sim\mathcal{D}}\left[\ell(r_{t_0+t};(x,y)) \mid x_1 \neq 0\right] \lesssim \sqrt{\varepsilon_1/q} + \kappa \log \frac{1}{\varepsilon_1} \lesssim \kappa \log \frac{1}{\varepsilon_1} \tag{C.8}$$

where we used the fact that $\varepsilon_1 \leq \kappa^2 p^2 q^3$.

It follows thats

$$\mathbb{E}\left[\ell(r_{t_0+t} + g_{t_0+t};(x,y))\right] \tag{C.9}$$
$$\leq \Pr[x_2 = 0]\mathbb{E}\left[\ell(r_{t_0+t};(x,y))\right] + \Pr[x_2 \neq 0]\mathbb{E}\left[\ell(r_{t_0+t} + g_{t_0+t};(x,y))\right] \tag{C.10}$$
$$\leq \mathbb{E}\left[\ell(r_{t_0+t};(x,y)) \mid x_1 \neq 0\right]\Pr[x_2 = 0] + \widetilde{O}(1)\Pr\left[x_2 \neq 0, \|x_2\|_2 = O\left(\sqrt{\varepsilon_1/q^3}\right)\right] + \varepsilon_1 \tag{C.11}$$

$$\leq \widetilde{O}\left(\sqrt{\varepsilon_1/q^3}\right) + \varepsilon_1 \leq O\left(p\kappa \log \frac{1}{\varepsilon_1}\right) \tag{C.12}$$

Here the last step uses the definition of $\varepsilon_1$ that $\varepsilon_1 \leq \kappa^2 p^2 q^3$. $\qquad\square$

## C.2 Proof of Theorem 3.5

We will prove Theorem 3.5 using Lemma 5.3 by roughly arguing that the predictions made by $r_t$ will be heavily influenced by a vector $\alpha$ in the low rank span of examples from $\bar{\mathcal{M}}_2$. With high probability, this vector $\alpha$ will be noisy and not align well with the ground truth $w^\star$, leading to mispredictions.

*Proof of Theorem 3.5.* Recall that $\varepsilon_2'$ denotes the stopping criterion used in Theorem 3.5 and $\varepsilon_3 = d^{-1/32}\frac{1}{\varepsilon_2'^2}$. Using Lemma 5.3, we know that w.h.p.

$$r_t(x_1) - r_t(-x_1) = 2\langle\alpha, x_1\rangle \pm \widetilde{O}(\varepsilon_3) \tag{C.13}$$

Consider the matrix $M = (x_1^{(i)})_{i\in\bar{\mathcal{M}}_2} \in \mathbb{R}^{d\times Np}$. By definition, we know that $M = M_0 + M_1$ where $M_0 = w^\star\beta^\top$ where $\beta_i \in \{-d^{-1/2}, d^{-1/2}\}$ and $M_1$ is a Gaussian random matrix with each entry i.i.d. $\mathcal{N}(0, 1/d)$.

By Lemma G.2 we know that w.h.p. over the randomness of $x_1^{(i)}$'s, for $\alpha \in \text{span}\{x_1^{(i)}\}_{i\in\bar{\mathcal{M}}_2}$ we have as long as $Np \leq d/2$: $\frac{\langle\alpha, w^\star\rangle}{\|\alpha\|_2\|w^\star\|_2} \leq 0.9$. For every randomly chosen $x_1$, we can also write $x_1 = \gamma w^\star + \beta$ where $\beta\perp w^\star$ so $\beta$ is independent of $\gamma$, hence

$$\langle\alpha, x_1\rangle = \gamma\langle\alpha, w^\star\rangle + \langle\alpha, \beta\rangle \tag{C.14}$$

Note that $\langle\alpha, \beta\rangle \sim \mathcal{N}(0, \sigma^2\|\alpha\|_2^2/d)$ with $\sigma \geq 0.1$, and with probability at least $0.1$, $\gamma \leq 2\|\alpha\|_2/\sqrt{d}$. This implies that with probability at least $\Omega(1)$ over a randomly chosen $x_1$ we can have:

$$\langle w^\star, x_1\rangle = \gamma < 0, \quad |\gamma| \leq 2\|\alpha\|_2/\sqrt{d} \tag{C.15}$$

For $\beta$, we know that with probability at least $\Omega(1)$, we have:

$$\langle\alpha, \beta\rangle \geq 3\|\alpha\|_2/\sqrt{d} \tag{C.16}$$

Moreover, since $\beta$ is independent of $\gamma$, we know that with probability $\Omega(1)$ both events can happen, in which case:

$$\langle w^\star, x_1\rangle < 0, \quad \langle\alpha, x_1\rangle = \gamma\langle\alpha, w^\star\rangle + \langle\alpha, \beta\rangle \geq \|\alpha\|_2/\sqrt{d} \tag{C.17}$$

Thus, since $\|\alpha\|_2 = \Omega(\sqrt{Np})$ by Lemma 5.3, we know that as long as

$$\frac{\sqrt{p}}{\kappa} = \frac{\sqrt{Np}}{\sqrt{d}} = \tilde{\Omega}(\varepsilon_3) \tag{C.18}$$

which is implied by $\varepsilon_3 = \widetilde{O}\left(\frac{\sqrt{p}}{\kappa}\right)$, it holds that $\langle \alpha, x_1 \rangle \geq \widetilde{\Omega}(\varepsilon_3)$. This implies that

$$r_t(x_1) = r_t(-x_1) + 2\langle \alpha, x_1 \rangle \pm \widetilde{O}(\varepsilon_3) \tag{C.19}$$
$$\geq r_t(-x_1) \tag{C.20}$$

However, since $\langle w^\star, x_1 \rangle < 0$, we know that either $r_t(x_1) < 0$, which results in $r_t(-x_1) < 0$ but $\langle w^\star, -x_1 \rangle > 0$. So when $x_2 = 0$, the network classifies $(-x_1, 0)$ incorrectly. On the other hand, we have when $r_t(x_1) > 0$ the network will classify $(x_1, 0)$ incorrectly. Since $\langle w^\star, x_1 \rangle < 0$ and $r_t(x_1) \geq r_t(-x_1)$ holds with probability $\Omega(1)$, this shows that the test error is at least $\Omega(p)$. $\qquad\square$

## D  Proofs for Large Learning Rate Lemmas

### D.1  Proofs for Lemma 4.1

To prove Lemma 4.1, we will show that the network will learn all examples with $\mathcal{P}$ component while the learning rate is large. The key to the proof is that although the large learning rate noise only allows the network to search over coarse kernels, $\mathcal{P}$ is still learnable by these kernels because of its linearly-separable structure. To make this precise, we decompose the weights $U_t$ Into the signal and noise components, and show that there exists a fixed "target" signal matrix which will classify $\mathcal{P}$ correctly no matter the noise matrix.

Recall our definitions of $f_t(B; x)$, $K_t(B)$ in (B.1) and (B.2), and that

$$L_t = \widehat{L}(U_t) = K_t(\overline{U}_t) \tag{D.1}$$
$$\nabla_U \widehat{L}(U_t) = \nabla K_t(\overline{U}_t) \tag{D.2}$$

Recall that Lemma B.1 leverages the linearly-separable structure of $\mathcal{P}$ to find a "target" signal matrix that correctly classifies $\mathcal{P}$ w.h.p over the noise matrix. We state its proof below.

*Proof of Lemma B.1.* By proposition A.3, $\|\overline{U}_t\|_F \leq O\left(\frac{1}{\lambda}\right)$. We apply Lemma A.8 as follows: by Proposition A.7, $\widetilde{U}_t$'s entry has marginal distribution $\mathcal{N}(0, \tau_0^2)$ and therefore the column of $\widetilde{U}_t$ has distribution $\mathcal{N}(0, \tau_0^2 I_{m\times m})$. Since w.h.p. $\|x\|_2 \lesssim \sqrt{\log d}$, the coupling Lemma A.8 gives

$$\|\mathbb{1}(U_t x) - \mathbb{1}(\widetilde{U}_t x)\|_0 \leq \varepsilon_s m \tag{D.3}$$

On the other hand, we also have by Proposition A.5, using the fact that $\max_i \|[\overline{U}_i]\|_2 \lesssim \frac{1}{\sqrt{m}\lambda}$, w.h.p.

$$\left| N_{U_t}(u, \widetilde{U}_t; x) \right| \lesssim \tau_0 \log d + \frac{\varepsilon_s}{\lambda} \lesssim \tau_0 \log d \tag{D.4}$$

Here in the last inequality we used the fact that the network is sufficiently over-parameterized so that $\varepsilon_s = \widetilde{O}(\tau_0 \lambda)$.

Using (D.4), noting that our choice of $m, \lambda, \tau_0$ satisfies $\tau_0 \log d = o(\varepsilon_1)$, we conclude

$$\left| N_{U_t}(u, \widetilde{U}_t; x) \right| \leq \varepsilon_1/20 \tag{D.5}$$

Now, let us consider $U^* = (W^*, V^*)$ given by $V^* = 0$ and an $W^* \in \mathbb{R}^{m\times d}$ defined as: for all $i \in [m]$, $W_i^* = 20 w_i \sqrt{d} w^\star \log \frac{1}{\varepsilon_1} \in \mathbb{R}^d$. We will have $\|U^*\|_F^2 = O\left(d^2 \log \frac{1}{\varepsilon_1}\right)$. We first decompose $f_t(U^*; x)$ into

$$f_t(U^*, x) = N_{U_t}(u, U^* + \widetilde{U}_t; x) \tag{D.6}$$
$$= N_{U_t}(u, \widetilde{U}_t; x) + N_{U_t}(u, U^*; x) \tag{D.7}$$

For the term $N_{U_t}(u, U^*; x)$, we know that

$$N_{U_t}(u, U^*; x) = N_{W_t}(w, W^*; x) = 20\langle w^\star, x_1 \rangle \sqrt{d} \log \frac{1}{\varepsilon_1} \times \sum_{i=1}^{m/2} w_i^2 \mathbb{1}([W_t]_i x_1) \tag{D.8}$$

$$= 20\langle w^\star, x_1 \rangle \sqrt{d} \log \frac{1}{\varepsilon_1} \times \frac{1}{m}\|\mathbb{1}(W_t x_1)\|_1 \tag{D.9}$$

By Lemma A.8, we know that $\left|\mathbb{1}(W_t x) - \mathbb{1}(\widetilde{W}_t x)\right|_1 \leq O\left(\varepsilon_s m\right)$ and that $20\langle w^\star, x_1\rangle\sqrt{d}\log\frac{1}{\varepsilon_1} \lesssim \sqrt{d}\log d$, which implies that

$$N_{U_t}(u, U^*; x) = 20\langle w^\star, x_1\rangle\sqrt{d}\log\frac{1}{\varepsilon_1} \times \frac{1}{m}\|\mathbb{1}(\widetilde{W}_t x_1)\|_1 \pm O\left(\sqrt{d}\varepsilon_s\log d\right) \tag{D.10}$$

Note that entries of $\widetilde{W}_t x_1$ are i.i.d. random Bernoulli$(1/2)$, thus we know that w.h.p.

$$\frac{2}{m}\|\mathbb{1}(\widetilde{W}_t x_1)\|_1 = \frac{1}{2} \pm O(m^{-1/2}\sqrt{\log d}) = \frac{1}{2} \pm O(m^{-1/3}) \tag{D.11}$$

Thus, by our choice that $m^{-1/3} = O(\varepsilon_1)$ and $\sqrt{d}\varepsilon_s = O(\varepsilon_1)$,

$$\left|N_{U_t}(u, U^*; x) - 5\langle w^\star, x_1\rangle\log\frac{1}{\varepsilon_1}\right| \leq \frac{\varepsilon_1}{20} \tag{D.12}$$

By (D.5), this also implies that

$$\left|N_{U_t}(u, \widetilde{U}_t + U^*; x) - 5\langle w^\star, x_1\rangle\log\frac{1}{\varepsilon_1}\right| \leq \frac{\varepsilon_1}{10} \tag{D.13}$$

By definition of $w^\star$, we know that

$$\frac{1}{N}\sum_{i=1}^N \ell\left(5\langle w^\star, x_1^{(i)}\rangle\log\frac{1}{\varepsilon_1}; (x^{(i)}, y^{(i)})\right) \leq q\log 2 + \varepsilon_1/5 \tag{D.14}$$

Thus, from the fact that $\ell$ is 1-Lipschitz, it follows that

$$K_t(U^*) \leq q\log 2 + \varepsilon_1/2 \tag{D.15}$$

$\square$

Now we wish to argue that even though the noise matrix is changing, gradient descent will still find the fixed target signal matrix $U^\star$. This leverages the fact that once we fix the activation patterns, we can view each step of the optimization as gradient descent with respect to a convex, but changing, function. Below we provide a proof of Theorem B.2, which allows for optimization of this changing function.

*Proof of Theorem B.2.* For the sake of contradiction, we assume that $K_t(z_t) \geq c^\star + \mu$ for all $t \leq T$. Using the definition of $K_t^\lambda$, we have that the update rule of $z_t$ can be written as

$$z_{t+1} = z_t - \eta\nabla K_t(z_t) - \eta\lambda z_t \tag{D.16}$$
$$= (1 - \eta\lambda)z_t - \eta\nabla K_t(z_t) \tag{D.17}$$

It follows that

$$\|z_{t+1} - z^\star\|_2^2 = \|(1 - \eta\lambda)(z_t - z^\star) - \eta(\lambda z^\star + \nabla K_t)\|_2^2 \tag{D.18}$$
$$= \|(1 - \eta\lambda)(z_t - z^\star)\|_2^2 + \|\eta(\lambda z^\star + \nabla K_t)\|_2^2 - 2\eta(1 - \eta\lambda)\langle\nabla K_t(z_t), z_t - z^\star\rangle$$
$$- 2\eta\lambda(1 - \eta\lambda)\langle z_t - z^\star, z^\star\rangle \qquad \text{(expanding)}$$
$$\leq \|(1 - \eta\lambda)(z_t - z^\star)\|_2^2 + 2\eta^2(\lambda^2 R^2 + L^2) - 2\eta(1 - \eta\lambda)(K_t(z_t) - K_t(z^\star))$$
$$\qquad \text{(by convexity of } K_t)$$
$$+ 2\eta\lambda(1 - \eta\lambda)\|z_t\|R + 2\eta\lambda(1 - \eta\lambda)R^2 \tag{D.19}$$

Assuming that $\|z_t - z^\star\|_2 \leq R$, we have that as long as $\lambda R^2 \leq \frac{1}{100}\mu$ and $\eta \leq \frac{\mu}{100(\lambda^2 R^2 + L^2)}$, we have:

$$\|z_{t+1} - z^\star\|_2^2 \leq \|(z_t - z^\star)\|_2^2 + 2\eta^2(\lambda^2 R^2 + L^2) - 2\eta(1 - \eta\lambda)\mu + 6\eta\lambda R^2 \tag{D.20}$$
$$\leq \|(z_t - z^\star)\|_2^2 - \eta\mu \tag{D.21}$$

Therefore, by induction,

$$\|z_T - z^\star\|_2^2 \leq \|(z_0 - z^\star)\|_2^2 - T\eta\mu \leq R^2 - T\eta\mu < 0 \tag{D.22}$$

which is a contradiction.

$\square$

## D.2 Proof of Lemma 4.2

We define $\tilde{g}_t$ to be the neural network operating on $x_2$ with activation pattern computed from $\widetilde{V}_t$ and and weights using $\overline{V}_t$:

$$\tilde{g}_t(x) = \tilde{g}_t(x_2) = N_{\widetilde{V}_t}(v, \overline{V}_t; x) \tag{D.23}$$

In the full proof of Lemma 4.2 at the end of the section, we will show that $\tilde{g}_t$ is very close to $g_t$ and therefore we focus on $\tilde{g}_t$ in most parts of the section, and show that it satisfies the almost-linearity condition in Lemma 4.2.

In this section, we will often consider the activation patterns on the inputs $z, z - \zeta, z + \zeta$ at various time steps. For convenience, we have the following definition:

**Definition D.1.** *For any $s$, and vector $w$, let $\mathcal{E}_s^w \triangleq \{i \in [m] : [\widetilde{V}_s]_i w \geq 0\}$ denote the set of neurons that have positive pre-activation on the input $w$ (with weights $\widetilde{V}_s$), and $\bar{\mathcal{E}}_s^w \triangleq \{i \in [m] : [\widetilde{V}_s]_i w < 0\}$ be the set of neurons with negative pre-activations on the input $w$. (We will mostly be interested in the quantities $\mathcal{E}^{z-\zeta}, \bar{\mathcal{E}}^{z-\zeta}, \mathcal{E}^{z+\zeta}, \bar{\mathcal{E}}^{z+\zeta}$ and their intersections.)*

For a set $\mathcal{E} \subset [m]$, we will use $\mathbb{1}(\mathcal{E}) \in \{0,1\}^m$ to denote the indicator vector for the set $\mathcal{E}$. With this notation, we have that

$$\mathbb{1}(\mathcal{E}_s^x) = \mathbb{1}(\widetilde{V}_s x) \tag{D.24}$$

We start by providing a decomposition of $\tilde{g}_t(z - \zeta) + \tilde{g}_t(z + \zeta) - 2\tilde{g}_t(z)$, and a bound based on how much the activation of $z, z - \zeta, z + \zeta$ differs.

**Lemma D.2.** *Let $Q_t \triangleq diag(v)\overline{V}_t$. Then, we have that*

$$\tilde{g}_t(z - \zeta) + \tilde{g}_t(z + \zeta) - 2\tilde{g}_t(z)$$
$$= (\mathbb{1}(\mathcal{E}_t^{z-\zeta}) + \mathbb{1}(\mathcal{E}_t^{z+\zeta}) - 2\mathbb{1}(\mathcal{E}_t^z))^\top Q_t z + (\mathbb{1}(\mathcal{E}_t^{z+\zeta}) - \mathbb{1}(\mathcal{E}_t^{z-\zeta}))^\top Q_t \zeta \tag{D.25}$$

*Proof.* We fix $t$ and drop the subscript of $t$ throughout the proof. Recall the definition of $\tilde{g}_t$ in equation (D.23), we have

$$\tilde{g}(x) := N_{\widetilde{V}}(v, \overline{V}; x) = v^\top \left( \mathbb{1}(\widetilde{V}x) \odot \overline{V}x \right)$$
$$= \mathbb{1}(\widetilde{V}x)^\top Q x \qquad \text{(by the definition of } Q = \text{diag}(v)\overline{V})$$

Therefore,

$$\tilde{g}(z - \zeta) + \tilde{g}(z + \zeta) - 2\tilde{g}(z) = \mathbb{1}(\mathcal{E}^{z-\zeta})^\top Q(z - \zeta) + \mathbb{1}(\mathcal{E}^{z+\zeta})^\top Q(z + \zeta) - 2\mathbb{1}(\mathcal{E}^z)^\top Q z$$
$$= (\mathbb{1}(\mathcal{E}^{z-\zeta}) + \mathbb{1}(\mathcal{E}^{z+\zeta}) - 2\mathbb{1}(\mathcal{E}^z))^\top Q z + (\mathbb{1}(\mathcal{E}^{z+\zeta}) - \mathbb{1}(\mathcal{E}^{z-\zeta}))^\top Q \zeta$$
$$\square$$

Towards bounding the terms in equation (D.25), we will need to reason about the activations patterns of $z, z - \zeta, z + \zeta$ at various time steps. We first show that the activation patterns of $z - \zeta$ and $z + \zeta$ have to agree in most of neurons except an $\approx r$ fraction of them. This will be useful to show that the second term of the RHS of equation (D.25) is small.

**Proposition D.3.** *In the setting of Lemma D.2, w.h.p over the randomness of the initialization and all the randomness in the algorithm, for every $t \leq \text{poly}(d), i \in [m]$, $i \in \mathcal{E}_t^{z-\zeta} \oplus \mathcal{E}_t^{z+\zeta}$ implies that $|[\widetilde{V}_t]_i z| \lesssim \tau_0 r \sqrt{\log d}$. Moreover, the size of the set $\mathcal{E}_t^{z-\zeta} \oplus \mathcal{E}_t^{z+\zeta}$ is bounded by*

$$|\mathcal{E}_t^{z-\zeta} \oplus \mathcal{E}_t^{z+\zeta}| \lesssim rm\sqrt{\log d} \tag{D.26}$$

*Proof.* Recall that $[\widetilde{V}_t]_i \in \mathbb{R}^{1 \times d}$ denote the $i$-th row of the matrix $\widetilde{V}_t$. Recall that $i \in \mathcal{E}_t^{z-\zeta} \oplus \mathcal{E}_t^{z+\zeta}$ means that $[\widetilde{V}_t]_i(z - \zeta)$ and $[\widetilde{V}_t]_i(z + \zeta)$ have different signs, which in turn implies that

$$|[\widetilde{V}_t]_i z| \leq |[\widetilde{V}_t]_i \zeta| \tag{D.27}$$

Recall that $\|\zeta\|_2 = r$ and by Proposition A.7 $[\widetilde{V}_t]_i$ has distribution $\mathcal{N}(0, \tau_0^2 I_{d\times d})$. Therefore, by standard Gaussian concentration and union bound, with high probability over the randomness of the initialization and the algorithm, for all $t \le \mathrm{poly}(d)$,

$$|[\widetilde{V}_t]_i\zeta| \lesssim \tau_0\|\zeta\|_2\sqrt{\log d} = \tau_0 r\sqrt{\log d}. \tag{D.28}$$

This proves the first part of the lemma.

Moreover, note that $\Pr\left[|[\widetilde{V}_t]_i z| \le \tau_0 r\sqrt{\log d}\right] \lesssim r\sqrt{\log d}$. By the independence between $[\widetilde{V}_t]_i$'s and standard concentration inequalities (Bernstein inequality), we have that with high probability, there are at most $rm\sqrt{\log d} + \log d$ entries $i \in [m]$ satisfying $|[\widetilde{V}_t]_i z| \le \tau_0 r\sqrt{\log d}$. Together with the first part of the lemma, and that $m$ is sufficiently large so that $rm\sqrt{\log d} + \log d \lesssim rm\sqrt{\log d}$, we complete the proof of equation (D.26). $\qquad\square$

We use the lemma above to conclude that the second term in the decomposition (D.25) is at most on the order of $r^2/\lambda$.

**Proposition D.4.** *In the setting of Lemma D.2, we have that*

$$\|(\mathbb{1}(\mathcal{E}_t^{z+\zeta}) - \mathbb{1}(\mathcal{E}_t^{z-\zeta}))^\top Q_t\zeta\|_2 \lesssim \frac{r^2\sqrt{\log d}}{\lambda}. \tag{D.29}$$

*Proof.*

$$|(\mathbb{1}(\mathcal{E}_t^{z+\zeta}) - \mathbb{1}(\mathcal{E}_t^{z-\zeta}))^\top Q_t\zeta| \le \|(\mathbb{1}(\mathcal{E}_t^{z+\zeta}) - \mathbb{1}(\mathcal{E}_t^{z-\zeta}))^\top Q_t\|_2\|\zeta\|_2 \tag{D.30}$$

By the definition of our algorithm, before annealing the learning rate, we have

$$[Q_t]_i = v_i \cdot [\overline{V}_t]_i = v_i\sum_{s=1}^t \eta_1(1 - \eta_1\lambda)^{t-s}[\nabla_V\widehat{L}(U_{s-1})]_i. \tag{D.31}$$

Using Proposition A.3 and that $|v_i| = \frac{1}{\sqrt{m}}$, we have that $\|[Q_t]_i\|_2 \lesssim \frac{1}{\lambda m}$. It follows that

$$\|(\mathbb{1}(\mathcal{E}_t^{z+\zeta}) - \mathbb{1}(\mathcal{E}_t^{z-\zeta}))^\top Q_t\|_2 \le |\mathcal{E}_t^{z-\zeta} \oplus \mathcal{E}_t^{z+\zeta}| \cdot \max_i \|[Q_t]_i\|_2 \lesssim \frac{r\sqrt{\log d}}{\lambda}. \tag{D.32}$$

Equation above and equation (D.30) complete the proof. $\qquad\square$

Next we will reason about the first term of the RHS of equation (D.25). Note that this is less obvious than the bound for the second term of RHS because both $Q$ and $z$ don't depend on the scale of $r$, whereas the norm of $\mathbb{1}(\mathcal{E}_t^{z-\zeta}) + \mathbb{1}(\mathcal{E}_t^{z+\zeta}) - 2\mathbb{1}(\mathcal{E}_t^z)$ only linearly depends on $r$. However, it is still the case that the first term of RHS of (D.25) scales in $r^2$ because of the subtle interactions between $\mathbb{1}(\mathcal{E}_t^{z-\zeta}) + \mathbb{1}(\mathcal{E}_t^{z+\zeta}) - 2\mathbb{1}(\mathcal{E}_t^z)$ and $Q_t$, as demonstrated in the proofs below.

The following lemma decomposes $Q$ into a sum of the contribution of the gradient from all the previous steps.

**Proposition D.5.** *In the setting of Lemma D.2, let $\Delta Q_t \triangleq diag(v)\nabla_V\widehat{L}(U_t)$. ($\Delta Q_t$ can be viewed as the raw change of $Q_t$ at the time step $t$ without considering the effect of the regularizer.) We have that*

$$|(\mathbb{1}(\mathcal{E}_t^{z-\zeta}) + \mathbb{1}(\mathcal{E}_t^{z+\zeta}) - 2\mathbb{1}(\mathcal{E}_t^z))^\top Q_t z| \le \eta_1\sum_{s=1}^t \|(\mathbb{1}(\mathcal{E}_t^{z-\zeta}) + \mathbb{1}(\mathcal{E}_t^{z+\zeta}) - 2\mathbb{1}(\mathcal{E}_t^z))^\top \Delta Q_{s-1}\|_2$$

*Proof.* Denote $a = \mathbb{1}(\mathcal{E}_t^{z-\zeta}) + \mathbb{1}(\mathcal{E}_t^{z+\zeta}) - 2\mathbb{1}(\mathcal{E}_t^z)$ for notational simplicity. By definition of our algorithm, we have

$$a^\top Q_t = a^\top \mathrm{diag}(v)\sum_{s=1}^t \eta_1(1 - \eta_1\lambda)^{t-s}\nabla_V\widehat{L}(U_{s-1}) = a^\top\sum_{s=1}^t \eta_1(1 - \eta_1\lambda)^{t-s}\Delta Q_{s-1} \tag{D.33}$$

It follows that

$$\|a^\top Q_t\|_2 \le \eta\sum_{s=1}^t \|a^\top \Delta Q_{s-1}\|_2.$$

Using the fact that $\|z\|_2 \le 1$ we complete the proof. $\qquad\square$

In the sequel, we will bound from above the quantity $\|(\mathbb{1}(\mathcal{E}_t^{z-\zeta}) + \mathbb{1}(\mathcal{E}_t^{z+\zeta}) - 2\mathbb{1}(\mathcal{E}_t^z))^\top \Delta Q_{s-1}\|_2$ for every $s$. One important fact is that the following proposition which shows that $\Delta Q_s$ has a lot of repetitive rows that enable additional cancellation in addition to the cancellation in $\mathbb{1}(\mathcal{E}_t^{z-\zeta}) + \mathbb{1}(\mathcal{E}_t^{z+\zeta}) - 2\mathbb{1}(\mathcal{E}_t^z)$.

**Proposition D.6.** *Define the analog of $\mathcal{E}_s^w$ with $V_t$ to compute the activation pattern: for any $s$, and vector $w$, let $\mathcal{G}_s^w \triangleq \{i \in [m] : [V_s]_i w \geq 0\}$ and define $\bar{\mathcal{G}}_s^w \triangleq \{i \in [m] : [V_s]_i w < 0\}$ similarly.*

*Suppose at some iteration $s$, $z - \zeta$ and $z + \zeta$ have the same activation pattern at neuron $i$ and $j$ in the sense that $i, j \in \mathcal{G}_s^{z-\zeta} \cap \mathcal{G}_s^{z+\zeta}$, or $i, j \in \bar{\mathcal{G}}_s^{z-\zeta} \cap \bar{\mathcal{G}}_s^{z+\zeta}$. Then the corresponding gradient update at that iteration for the weight vectors associated with $i$ and $j$ are the same up to a potential sign flip:*

$$[\Delta Q_s]_i = v_i [\nabla_V \widehat{L}(U_s)]_i = v_j [\nabla_V \widehat{L}(U_s)]_j = [\Delta Q_s]_j \tag{D.34}$$

*Moreover, suppose we have that $i, j$ satisfy that $[\widetilde{V}_s]_i x \gtrsim \tau_0 r \sqrt{\log d}$ and $[\widetilde{V}_s]_j x \gtrsim \tau_0 r \sqrt{\log d}$ (or $[\widetilde{V}_s]_i x \lesssim -\tau_0 r \sqrt{\log d}$ and $[\widetilde{V}_s]_j x \lesssim -\tau_0 r \sqrt{\log d}$) for $x \in \{z - \zeta, z + \zeta\}$, then the same conclusion holds for $i$ and $j$.*

*Proof.* Note that by definition, $[\Delta Q_s]_i = v_i [\nabla_V \widehat{L}(U_s)]_i$, and thus it suffices to prove that $v_i [\nabla_V \widehat{L}(U_s)]_i = v_j [\nabla_V \widehat{L}(U_s)]_j$. By Proposition A.1, we have that

$$[\nabla_V \widehat{L}(U_s)]_i = \widehat{\mathbb{E}} \left[ \ell'(f(u, U_s; (x, y))) v_i \mathbb{1}([V_s]_i x_2) x_2 \right] \tag{D.35}$$

Note that $x_2$ can only take (a positive scaling of) four values $z - \zeta, z, z + \zeta, 0$. We claim that for every choice of these four values, for the $i, j$ satisfying the condition of the lemma, we have

$$\ell'(f(u, U_s; (x, y))) \mathbb{1}([V_s]_i x_2) x_2 = \ell'(f(u, U_s; (x, y))) \mathbb{1}([V_s]_j x_2) x_2 \tag{D.36}$$

Note that the equation above together with $v_i^2 = v_j^2 = 1$ suffices to complete the proof.

Equation (D.36) is true for $x_2 = 0$. Suppose without loss of generality, $i, j \in \mathcal{G}_s^{z-\zeta} \cap \mathcal{G}_s^{z+\zeta}$. Then we know that $i, j \in \mathcal{G}_s^z$ because $[V_s]_i(z - \zeta) + [V_s]_i(z + \zeta) = 2[V_s]_i z$. Therefore $\mathbb{1}([V_s]_i x_2) = \mathbb{1}([V_s]_j x_2) = 1$ for all $x_2 \in \{z - \zeta, z, z + \zeta\}$. Thus we proved equation (D.36) and complete the proof of the first part of the lemma.

Now to prove the second part of the lemma, suppose $i, j$ satisfy that $[\widetilde{V}_s]_i x \gtrsim \tau_0 r \sqrt{\log d}$ and $[\widetilde{V}_s]_j x \gtrsim \tau_0 r \sqrt{\log d}$ for $x \in \{z - \zeta, z + \zeta\}$. Using $\|[\widetilde{V}_s]_i\|_2 \leq \frac{1}{\lambda \sqrt{m}}$ from Proposition A.3, we have that $[V_s]_i z \geq [\widetilde{V}_s]_i z - |[\overline{V}_s]_i z| \gtrsim \tau_0 r \sqrt{\log d} - O(\frac{1}{\lambda \sqrt{m}}) \geq \tau_0 r \sqrt{\log d}$ where used the assumption that $1/\lambda = \text{poly}(d)$ and $m = \text{poly}(d/\tau_0)$. Therefore, we conclude that $i, j \in \mathcal{G}_s^{z-\zeta} \cap \mathcal{G}_s^{z+\zeta}$. Now by the first lemma of the lemma we complete the proof. □

Now we are ready to bound the first term on the RHS of equation D.25, which is the crux of the proofs in this section. The key here is to get a bound that scales quadratically in $r$.

**Proposition D.7.** *In the setting of Lemma D.2, let $\Delta Q_s$ be defined in Proposition D.5. Then, we have that*

$$\|(\mathbb{1}(\mathcal{E}_t^{z-\zeta}) + \mathbb{1}(\mathcal{E}_t^{z+\zeta}) - 2\mathbb{1}(\mathcal{E}_t^z))^\top \Delta Q_s\|_2 \lesssim \frac{r^2 \sqrt{\log d}}{\sqrt{\lambda \eta_1(s - t)}} \tag{D.37}$$

*As a direct corollary of the equation above and Proposition D.5, we have that*

$$|(\mathbb{1}(\mathcal{E}_t^{z-\zeta}) + \mathbb{1}(\mathcal{E}_t^{z+\zeta}) - 2\mathbb{1}(\mathcal{E}_t^z))^\top Q_t z| \lesssim \frac{r^2 \sqrt{\log d}}{\lambda} \tag{D.38}$$

*Proof.* By the set operations and the facts that $\mathcal{E}_t^{z-\zeta} \cap \mathcal{E}_t^{z+\zeta} \subset \mathcal{E}_t^z$ and that $\mathcal{E}_t^z \subset \mathcal{E}_t^{z-\zeta} \cup \mathcal{E}_t^{z+\zeta}$, we have that

$$\mathbb{1}(\mathcal{E}_t^{z-\zeta}) + \mathbb{1}(\mathcal{E}_t^{z+\zeta}) - 2\mathbb{1}(\mathcal{E}_t^z) = \left( \mathbb{1}(\mathcal{E}_t^{z-\zeta} \backslash \mathcal{E}_t^z) - \mathbb{1}(\mathcal{E}^z \backslash \mathcal{E}_t^{z+\zeta}) \right) + \left( \mathbb{1}(\mathcal{E}_t^{z+\zeta} \backslash \mathcal{E}_t^z) - \mathbb{1}(\mathcal{E}_t^z \backslash \mathcal{E}_t^{z-\zeta}) \right)$$
$$\tag{D.39}$$

Define

$$\mathcal{F}_s^+ = \{i \in [m] : [\widetilde{V}_s]_i z \gtrsim \tau_0 r \sqrt{\log d}\}$$
$$\mathcal{F}_s^- = \{i \in [m] : [\widetilde{V}_s]_i z \lesssim -\tau_0 r \sqrt{\log d}\}$$
$$\mathcal{F}_s^c = \{i \in [m] : |[\widetilde{V}_s]_i z| \lesssim \tau_0 r \sqrt{\log d}\} \tag{D.40}$$

where the $\lesssim, \gtrsim$ notations hide universal constants that make the first conclusion of Proposition D.3 true. By the second part of Proposition D.3 (or more directly equation (D.28)), we have that $\mathcal{F}_s^+ \subset \mathcal{E}_s^{z-\zeta} \cap \mathcal{E}_s^{z+\zeta}$, and $\mathcal{F}_s^- \subset \bar{\mathcal{E}}_s^{z-\zeta} \cap \bar{\mathcal{E}}_s^{z+\zeta}$. By Proposition D.6, we have that for any $i, j \in \mathcal{F}_s^-$, $[\Delta Q_s]_i = [\Delta Q_s]_j$. For notational simplicity, let $A = \mathcal{E}_t^{z+\zeta} \backslash \mathcal{E}_t^z$ and $B = \mathcal{E}_t^z \backslash \mathcal{E}_t^{z-\zeta}$. Therefore it follows that

$$\left\| \left( \mathbb{1}(\mathcal{E}_t^{z+\zeta} \backslash \mathcal{E}_t^z) - \mathbb{1}(\mathcal{E}_t^z \backslash \mathcal{E}_t^{z-\zeta}) \right)^\top \Delta Q_s \right\|_2 = \left\| \sum_{i \in A} [\Delta Q_s]_i - \sum_{i \in B} [\Delta Q_s]_i \right\|_2$$

$$= \left\| \sum_{i \in A \cap \mathcal{F}_s^+} [\Delta Q_s]_i - \sum_{i \in B \cap \mathcal{F}_s^+} [\Delta Q_s]_i \right\|_2 + \left\| \sum_{i \in A \cap \mathcal{F}_s^-} [\Delta Q_s]_i - \sum_{i \in B \cap \mathcal{F}_s^-} [\Delta Q_s]_i \right\|_2$$

$$+ \left\| \sum_{i \in A \cap \mathcal{F}_s^c} [\Delta Q_s]_i - \sum_{i \in B \cap \mathcal{F}_s^c} [\Delta Q_s]_i \right\|_2$$

$$\leq \frac{1}{m} \left( \left| |A \cap \mathcal{F}_s^+| - |B \cap \mathcal{F}_s^+| \right| + \left| |A \cap \mathcal{F}_s^-| - |B \cap \mathcal{F}_s^-| \right| + |A \cap \mathcal{F}_s^c| + |B \cap \mathcal{F}_s^c| \right) \tag{D.41}$$

where in the last inequality we use that for any $i, j \in \mathcal{F}_s^-$, $[\Delta Q_s]_i = [\Delta Q_s]_j$, and the fact that $\|[\Delta Q_s]_i\|_2 = \frac{1}{\sqrt{m}} \|[\nabla_V \widehat{L}(U_s)]_i\|_2 \leq 1/m$ (by Proposition A.2.)

Next, we first bound

$$|A \cap \mathcal{F}_s^+| - |B \cap \mathcal{F}_s^+| = \sum_{i \in [m]} \mathbf{1}(i \in \mathcal{E}_t^{z+\zeta}, i \notin \mathcal{E}_t^z, i \in \mathcal{F}_s^+) - \mathbf{1}(i \in \mathcal{E}_t^z, i \notin \mathcal{E}_t^{z-\zeta}, i \in \mathcal{F}_s^+). \tag{D.42}$$

Note that the distribution of $([\widetilde{V}_s]_i, [\widetilde{V}_t]_i$'s are independent across the choice of $i$. Thus we will compute $\Pr[i \in \mathcal{E}_t^{z+\zeta}, i \notin \mathcal{E}_t^z, i \in \mathcal{F}_s^+] - \Pr[i \in \mathcal{E}_t^z, i \notin \mathcal{E}_t^{z-\zeta}, i \in \mathcal{F}_s^+]$ and then apply concentration concentration inequality for the sum. Note that the event here depends on three quantities $[\widetilde{V}_s]_i z$, $[\widetilde{V}_t]_i z$, and $[\widetilde{V}_t]_i \zeta$. First of all, $[\widetilde{V}_t]_i \zeta$ is independent of these other two because $\zeta$ is orthogonal to $z$ and $[\widetilde{V}_t]_i$ and $[\widetilde{V}_s]_i$ have spherical covariance matrices.

By the definition of $\widetilde{V}_s, \widetilde{V}_t$, we can express their relationship by writing $[\widetilde{V}_t]_i z = (1 - \eta_1 \lambda)^{t-s} [\widetilde{V}_s]_i z + [\Xi_{t,s}]_i z$, where $\Xi_{t,s} = \eta_1 \sum_{j \in [t-s]} (1 - \eta_1 \lambda)^{t-s-j} \xi_{s+j}$. Recall that by proposition A.7, we have $[\widetilde{V}_s]_i z \sim \mathcal{N}(0, \tau_0^2)$ and $[\Xi_{t,s}]_i z$ are two independent Gaussians. Let $\sigma_{t,s}$ be the variance of $[\Xi_{t,s}]_i z$. We compute $\sigma_{t,s}$ by observing that

$$\tau_0^2 = \mathsf{Var}([\widetilde{V}_t]_i z) = \mathsf{Var}((1 - \eta_1 \lambda)^{t-s} [\widetilde{V}_s]_i z) + \mathsf{Var}([\Xi_{t,s}]_i z) = (1 - \eta_1 \lambda)^{2(t-s)} \tau_0^2 + \sigma_{s,t}^2$$

Solving the equation we obtain that

$$\sigma_{s,t} = \sqrt{\tau_0^2 (1 - (1 - \eta_1 \lambda)^{2(t-s)})} \geq \tau_0 \sqrt{\lambda \eta_1 (s - t)} \tag{D.43}$$

Note that $\zeta^\top z = 0$, thus $[\widetilde{V}_s]_i z$ is independent of $[\widetilde{V}_t]_i \zeta$ conditioned on $[\widetilde{V}_t]_i z$, for every $s \leq t$. For notational simplicity, let $Y_1 = [\widetilde{V}_s]_i z$, $Y_2 = [\widetilde{V}_t]_i z$, and $Y_3 = [\widetilde{V}_t]_i \zeta$, and $\kappa = O(\tau_0 r \sqrt{\log d})$ where the big O notation hide the same constant factor used in defining $\mathcal{F}_s^+$ in equation (D.40). Let $Y_4 = [\Xi_{t,s}]_i z = Y_1 - \beta Y_2$ where $\beta = \eta_1 (1 - \eta_1 \lambda)^{t-s} \gtrsim 1$ (because $t \leq 1/(\eta_1 \lambda)$). Note that by the calculation above, $Y_4$ has standard deviation $\sigma_{s,t}$ which is bounded from below by $\tau_0 \sqrt{\lambda \eta_1 (s - t)}$.

Then, we have that

$$\Pr[i \in \mathcal{E}_t^{z+\zeta}, i \notin \mathcal{E}_t^z, i \in \mathcal{F}_s^+] = \Pr[Y_2 + Y_3 \geq 0, Y_2 \leq 0, Y_1 \geq \kappa] \tag{D.44}$$

$$= \Pr[Y_2 + Y_3 \geq 0, Y_2 \leq 0, Y_4 \geq \kappa - \beta Y_2] \tag{D.45}$$

$$= \mathop{\mathbb{E}}_{Y_2}[\Pr[Y_2 + Y_3 \geq 0, Y_2 \leq 0, Y_4 \geq \kappa - \beta Y_2 \mid Y_2]]$$

(by the law of total expecation)

$$= \mathop{\mathbb{E}}_{Y_2}[\mathbf{1}(Y_2 \leq 0)\Pr[Y_3 \geq -Y_2 \mid Y_2] \cdot \Pr[Y_4 \geq \kappa - \beta Y_2 \mid Y_2]]$$

(because $Y_1, Y_3, Y_4$ are independent conditioned on $Y_2$.)

Similarly, we have that

$$\Pr[i \in \mathcal{E}_t^z, i \notin \mathcal{E}_t^{z-\zeta}, i \in \mathcal{F}_s^+] = \Pr[Y_2 \geq 0, Y_2 - Y_3 \leq 0, Y_1 \geq \kappa]$$

$$= \Pr[-Y_2 \geq 0, -Y_2 - Y_3 \leq 0, -Y_1 \geq \kappa]$$

$$((Y_1, Y_2, Y_3) \text{ has the same distribution as } (-Y_1, -Y_2, Y_3)))$$

$$= \mathop{\mathbb{E}}_{Y_2}[\mathbf{1}(Y_2 \leq 0)\Pr[Y_3 \geq -Y_2 \mid Y_2] \cdot \Pr[Y_4 \leq -\kappa - \beta Y_2 \mid Y_2]]$$

(because $Y_1, Y_3, Y_4$ are independent conditioned on $Y_2$.)

$$= \mathop{\mathbb{E}}_{Y_2}[\mathbf{1}(Y_2 \leq 0)\Pr[Y_3 \geq -Y_2 \mid Y_2] \cdot \Pr[Y_4 \geq \kappa + \beta Y_2 \mid Y_2]]$$

(because $(Y_4, Y_2)$ has the same distribution as $(-Y_4, Y_2)$.)

Therefore, we have that

$$\left|\Pr[i \in \mathcal{E}_t^{z+\zeta}, i \notin \mathcal{E}_t^z, i \in \mathcal{F}_s^+] - \Pr[i \in \mathcal{E}_t^z, i \notin \mathcal{E}_t^{z-\zeta}, i \in \mathcal{F}_s^+]\right| \tag{D.46}$$

$$= \mathop{\mathbb{E}}_{Y_2}[\mathbf{1}(Y_2 \leq 0)\Pr[Y_3 \geq -Y_2 \mid Y_2]\Pr[\kappa - \beta Y_2 \leq Y_4 \leq \kappa + \beta Y_2 \mid Y_2]] \tag{D.47}$$

$$\lesssim \mathop{\mathbb{E}}_{Y_2}\left[\mathbf{1}(Y_2 \leq 0)\Pr[Y_3 \geq -Y_2 \mid Y_2]\frac{|Y_2|}{\sigma_{s,t}}\right] \quad \text{(because the density of } Y_4 \text{ is bounded by } O(1/\sigma_{s,t}))$$

$$\lesssim \mathop{\mathbb{E}}_{Y_2}\left[\mathbf{1}(Y_2 \leq 0)\exp(-|Y_2|^2/2(r^2\tau_0^2))\frac{|Y_2|}{\sigma_{s,t}}\right] \quad \text{(because } Y_3 \text{ has variance } r^2\tau_0^2)$$

$$\lesssim \int_{-\infty}^{0} 1/\tau_0 \cdot \exp(-z^2/(2r^2\tau_0^2))\exp(-z^2/\tau_0^2)|z|/\sigma_{s,t}dz \lesssim r^2\tau_0/\sigma_{s,t}$$

$$\lesssim \frac{r^2}{\sqrt{\lambda\eta_1(s-t)}} \tag{D.48}$$

Now by equation (D.42) and standard concentration inequality, and the fact that $m$ is sufficiently large, we have that with high probability,

$$\left||A \cap \mathcal{F}_s^+| - |B \cap \mathcal{F}_s^+|\right| \lesssim \frac{r^2m}{\sqrt{\lambda\eta_1(s-t)}} + \log d \lesssim \frac{r^2m}{\sqrt{\lambda\eta_1(s-t)}} \tag{D.49}$$

Similarly, we can prove that

$$\left||A \cap \mathcal{F}_s^-| - |B \cap \mathcal{F}_s^-|\right| \lesssim \frac{r^2m}{\sqrt{\lambda\eta_1(s-t)}} \tag{D.50}$$

Finally, we have that

$$\Pr[i \in \mathcal{E}_t^{z+\zeta}, i \notin \mathcal{E}_t^z, i \in \mathcal{F}_s^c] = \Pr\left[Y_2 + Y_3 \geq 0, Y_2 \leq 0, |Y_1| \leq \kappa\right] \tag{D.51}$$

$$= \mathbb{E}\left[\Pr\left[Y_2 + Y_3 \geq 0, Y_2 \leq 0, |Y_4 - \beta Y_2| \leq \kappa\right]\right]$$

(by the law of total expecation)

$$= \underset{Y_2}{\mathbb{E}}\left[\mathbf{1}(Y_2 \leq 0) \Pr\left[Y_3 \geq -Y_2 \mid Y_2\right] \cdot \kappa/\sigma_{s,t}\right]$$

(because the density of $Y_4$ is bounded by $O(1/\sigma_{s,t})$)

$$\lesssim \underset{Y_2}{\mathbb{E}}\left[\mathbf{1}(Y_2 \leq 0) \exp(-|Y_2|^2/2(r^2\tau_0^2)) \frac{\kappa}{\sigma_{s,t}}\right]$$

(because $Y_3$ has variance $r^2\tau_0^2$)

$$\lesssim \kappa r\tau_0/\sigma_{s,t} \lesssim \frac{r^2\sqrt{\log d}}{\sqrt{\lambda\eta_1(s-t)}} \tag{D.52}$$

Using standard concentration inequality and the fact that $m$ is sufficiently large, we have that with high probability,

$$|A \cap \mathcal{F}_s^c| \lesssim \frac{r^2 m\sqrt{\log d}}{\sqrt{\lambda\eta_1(s-t)}} + \log d \lesssim \frac{r^2 m\sqrt{\log d}}{\sqrt{\lambda\eta_1(s-t)}} \tag{D.53}$$

We can also prove the same bound for $|B \cap \mathcal{F}_s^c|$ analogously. Using equation (D.41) and the several equations above, we conclude that

$$\left\|\left(\mathbf{1}(\mathcal{E}_t^{z+\zeta}\backslash\mathcal{E}_t^z) - \mathbf{1}(\mathcal{E}_t^z\backslash\mathcal{E}_t^{z-\zeta})\right)^\top \Delta Q_s\right\|_2 \lesssim \frac{r^2\sqrt{\log d}}{\sqrt{\lambda\eta_1(s-t)}} \tag{D.54}$$

Thus equation (D.37) follows from equation (D.39) and proving a bound for $\left(\mathbf{1}(\mathcal{E}_t^{z+\zeta}\backslash\mathcal{E}_t^z) - \mathbf{1}(\mathcal{E}_t^z\backslash\mathcal{E}_t^{z-\zeta})\right)^\top \Delta Q_s$ similarly to the equation above. To prove equation (D.38), we use Proposition D.5, and equation (D.37) to obtain that

$$|(\mathbf{1}(\mathcal{E}_t^{z-\zeta}) + \mathbf{1}(\mathcal{E}_t^{z+\zeta}) - 2\mathbf{1}(\mathcal{E}_t^z))^\top Q_t z| \leq \eta_1 \sum_{s=1}^{t} \|(\mathbf{1}(\mathcal{E}_t^{z-\zeta}) + \mathbf{1}(\mathcal{E}_t^{z+\zeta}) - 2\mathbf{1}(\mathcal{E}_t^z))^\top \Delta Q_{s-1}\|_2$$

$$\lesssim \eta_1 \sum_{s=1}^{t} \frac{r^2\sqrt{\log d}}{\sqrt{\lambda\eta_1(s-t)}} \lesssim r^2\sqrt{\log d}\sqrt{t\eta_1/\lambda} \tag{D.55}$$

$$\lesssim r^2\sqrt{\log d}/\lambda \tag{D.56}$$

where the last step uses that the condition that $t \leq 1/(\eta_1\lambda)$.

$\square$

Now combining the Propositions above we are ready to prove Lemma 4.2.

*Proof of Lemma 4.2.* Using triangle inequality, Proposition A.8, and equation (A.6) of Proposition A.5, we have that for any $x$ of norm $O(1)$,

$$|g_t(x) - \tilde{g}_t(x)| \leq |N_{V_t}(v, \overline{V}_t; x) - N_{\widetilde{V}_t}(v, \overline{V}_t; x)| + |N_{V_t}(v, \widetilde{V}_t; x)| \tag{D.57}$$

$$\leq \|\overline{V}_t\|_F \tau_0^{-2} m^{-1/6} + \|\overline{V}_t\|_F^{5/3} \tau_0^{-2/3} m^{-1/6} + \tau_0 \log d$$

(by Proposition A.8, and equation (A.6) of Proposition A.5)

$$\leq 1/\mathrm{poly}(d)$$

(because $\tau_0 = 1/\mathrm{poly}\left(\frac{d}{\varepsilon}\right)$ and $m \geq \mathrm{poly}\left(\frac{d}{\varepsilon\tau_0}\right)$ and $\|\overline{V}_t\| \lesssim 1/\lambda$ by Proposition A.3.)

Thus we can only focus on $\tilde{g}_t$. Using Lemma D.2, we have that

$$|\tilde{g}_t(z-\zeta) + \tilde{g}_t(z+\zeta) - 2\tilde{g}_t(z)|$$

$$\leq |(\mathbf{1}(\mathcal{E}_t^{z-\zeta}) + \mathbf{1}(\mathcal{E}_t^{z+\zeta}) - 2\mathbf{1}(\mathcal{E}_t^z))^\top Q_t z| + |(\mathbf{1}(\mathcal{E}_t^{z+\zeta}) - \mathbf{1}(\mathcal{E}_t^{z-\zeta}))^\top Q_t \zeta| \tag{D.58}$$

$$\lesssim \frac{r^2\sqrt{\log d}}{\lambda} + \frac{r^2\sqrt{\log d}}{\lambda} \qquad \text{(by equation (D.38) of Proposition D.7 and Proposition D.4)}$$

which completes the proof.

$\square$

## D.3 Proof of Lemma B.3

The proof of Lemma B.3 relies on the fact that a smaller learning rate preserves the noise generated from the timestep before annealing. This allows us to reason that the new activations are similar to the original before reducing the learning rate.

*Proof of Lemma B.3.* By definition, we have that

$$[U_{t_0}]_i = [\overline{U}_{t_0}]_i + [\widetilde{U}_{t_0}]_i$$
$$[U_{t_0+t}]_i = [\overline{U}_{t_0+t}]_i + [\widetilde{U}_{t_0+t}]_i = [\overline{U}_{t_0+t}]_i + (1 - \eta_2\lambda)^t[\widetilde{U}_{t_0}]_i + [\Xi_t]_i \qquad (D.59)$$

where $\Xi_t := \eta_2 \sum_{j \leq t}(1 - \lambda\eta_2)^{t-j}\xi_{t_0+j}$.

By properties of a sum of Independent Gaussians, we have $[\Xi_t]_i \sim \mathcal{N}(0, \sigma_t^2 I)$ where $\sigma_t$ is the standard deviation of each entry of $\Xi_t$. We also have that $\Xi_t$ is independent of $\widetilde{U}_{t_0}$. Moreover, for every $t \leq \frac{1}{\eta_2\lambda}$, the standard deviation $\sigma_t$ can be bounded by

$$\sigma_t^2 = \eta_2^2 \sum_{j \leq t}(1 - \lambda\eta_2)^{2(t-j)}\tau_\xi^2 \leq \eta_2^2\tau_\xi^2 t$$
$$= \frac{\eta_2^2(\tau_0^2 - (1 - \eta_1\lambda)^2\tau_0^2)}{\eta_1^2}t \leq \frac{2\eta_2^2\lambda\tau_0^2 t}{\eta_1} \leq \frac{2\eta_2\tau_0^2}{\eta_1} \qquad (D.60)$$

(Note that since $\eta_2 \ll \eta_1$, we should expect that the standard deviations satisfy $\sigma_t \ll \sigma_0$. That is, the additional randomness introduced in the pre-activation is small.)

On the other hand, for every $t \leq \frac{1}{\eta_2\lambda}$, the contribution of $\widetilde{U}_{t_0}$ to $U_{t+t_0}$ is still present because the entry of $(1 - \eta_2\lambda)^t[\widetilde{U}_{t_0}]_i$ has variance at least on the order of the variance of the entries of $[\widetilde{U}_{t_0}]_i$, which is $\gtrsim \tau_0^2$. This also implies that the variance of the entries of $\widetilde{U}_{t_0+t}$ is lower bounded by the variance of $(1 - \eta_2\lambda)^t[\widetilde{U}_{t_0}]_i$. This in turn is lower bounded by $\tau_0^2$ up to constant factor.

Therefore, using the decomposition (D.59) and the bounds above, we should expect that the sign of $U_{t_0+t}$ strongly correlates with the the sign of $U_{t_0}$, which will be formally shown below. Using Lemma A.8, we have that the activation pattern is mostly decided by the noise part ($\widetilde{U}_{t+t_0}$ and $\widetilde{U}_{t_0}$), in the sense that for every $x$,

$$\|\mathbb{1}(U_{t_0}x) - \mathbb{1}(\widetilde{U}_{t_0}x)\|_1 \lesssim \|\overline{U}_{t_0}\|_F^{4/3}\tau_0^{-4/3}m^{2/3} \leq \varepsilon_s m \qquad (D.61)$$

This can obtained by setting $\tilde{U} = \widetilde{U}_{t_0}, \overline{U} = \overline{U}_{t_0}, \tau = \tau_0$ in Lemma A.8, and using $\|\overline{U}_{t_0}\|_F \leq 1/\lambda$ from Proposition A.3. Similarly, setting $\tilde{U} = \widetilde{U}_{t_0+t}, \overline{U} = \overline{U}_{t_0+t}$, and letting $\tau$ be the standard deviation of entries of $\widetilde{U}_{t_0+t}$ (which has been shown to be $\gtrsim \tau_0$), we get

$$\|\mathbb{1}(U_{t_0+t}x) - \mathbb{1}(\widetilde{U}_{t_0+t}x)\|_1 \lesssim \|\overline{U}_{t_0+t}\|_F^{4/3}\tau^{-4/3}m^{2/3} \leq \varepsilon_s m \qquad (D.62)$$

Fixing $x$, we can decompose our target to

$$\|\mathbb{1}(U_{t_0+t}x) - \mathbb{1}(U_{t_0}x)\|_1 \leq \qquad (D.63)$$

$$\|\mathbb{1}(U_{t_0+t}x) - \mathbb{1}(\widetilde{U}_{t_0+t}x)\|_1 + \|\mathbb{1}(\widetilde{U}_{t_0+t}x) - \mathbb{1}(\widetilde{U}_{t_0}x)\|_1 + \|\mathbb{1}(\widetilde{U}_{t_0}x) - \mathbb{1}(U_{t_0}x)\|_1 \qquad (D.64)$$

We've bounded the first and third term on the RHS of the equation above. For the middle term, let $\alpha_i = (1 - \eta_2\lambda)^t[\widetilde{U}_{t_0}]_i x$ and $\beta_i = [\Xi_{t+t_0}]_i x$. Note that $[\widetilde{U}_{t+t_0}]_i x = \alpha_i + \beta_i$ and that $\alpha_i$ and $\beta_i$ are zero-mean independent Gaussian random variables with variance $\gtrsim \tau_0^2\|x\|^2$ and variance $\lesssim \eta_2\tau_0^2\|x\|^2/\eta_1$, respectively. The basic property of Gaussian random variable implies that

$$\Pr\left[\mathbb{1}(\alpha_i + \beta_i) \neq \mathbb{1}(\beta_i)\right] \lesssim \sqrt{\frac{\eta_2\tau_0^2\|x\|^2/\eta_1}{\tau_0^2\|x\|^2}} = \sqrt{\eta_2/\eta_1} \qquad (D.65)$$

Since $\alpha_i, \beta_i$'s are independent, by basic concentration inequality (e.g., Bernstein inequality or Hoeffding inequality), we have that with high probability

$$\|\mathbb{1}(\widetilde{U}_t x) - \mathbb{1}(\widetilde{U}_{t_0}x)\|_1 \lesssim \sqrt{\eta_2/\eta_1}m + \sqrt{m\log d} \lesssim \sqrt{\eta_2/\eta_1}m + m^{2/3} \qquad (D.66)$$

Combining the equation above with equation (D.61), (D.62),and (D.64) completes the proof for the first part.

For the second part, we can bound

$$\left| N_{U_{t_0+t}}(u, U_{t_0+t}; x) - N_{U_{t_0}}(u, \overline{U}_{t_0+t}; x) \right| \tag{D.67}$$

$$\leq \left| N_{U_{t_0+t}}(u, U_{t_0+t}; x) - N_{U_{t_0+t}}(u, \overline{U}_{t_0+t}; x) \right| + \left| N_{U_{t_0+t}}(u, \overline{U}_{t_0+t}; x) - N_{U_{t_0}}(u, \overline{U}_{t_0+t}; x) \right| \tag{D.68}$$

$$\lesssim \left| N_{U_{t_0+t}}(u, \widetilde{U}_{t_0+t}; x) \right| \tag{D.69}$$

$$+ \frac{1}{\sqrt{m}} \| \mathbb{1}([U_{t_0+t}]x) - \mathbb{1}([U_{t_0}]x) \|_1 \max_i \| [\overline{U}_{t_0+t}]_i \|_2 \tag{D.70}$$

$$\lesssim \left( \sqrt{\frac{\eta_2}{\eta_1}} + \varepsilon_s \right) \times \frac{1}{\lambda} + \tau_0 \log d \tag{D.71}$$

where the last inequality is due to $\max_i \| [\overline{U}_{t_0+t}]_i \|_2 = O(1/\sqrt{m}\lambda)$ by Proposition A.3, and bounding $\left| N_{U_{t_0+t}}(u, \widetilde{U}_{t_0+t}; x) \right| \lesssim \frac{\varepsilon_s}{\lambda} + \tau_0 \log d$ by Proposition A.5.

$\square$

We note that this lemma also applies to the setting when $t_0 = 0$, i.e. we start with an initial small learning rate and compare to the random initialization. This is useful for the proofs in the small initial learning rate setting.

## D.4 Proof of Lemma 4.3

We will now show that the network learns patterns from $\mathcal{Q}$ once the learning rate is annealed by constructing a common target for the network at every subsequent time step. We will then use Theorem B.2 to show that the optimization finds this target. Let us define

$$\varepsilon_0 := \frac{1}{N} \sum_{i \in \mathcal{M}_1} \ell(r_{t_0}; (x^{(i)}, y^{(i)})) \tag{D.72}$$

Formally, we first show the following proposition, which proves the existence of a target solution that has good accuracy on $\overline{\mathcal{M}}_1$ and does not unlearn the network's progress on $\mathcal{M}_1$:

**Lemma D.8.** *In the setting of Lemma 4.3, let $K_t(B)$ be defined in equation (B.2). Then, there exists a solution $U^*$ satisfying $\|U^*\|_F^2 = \widetilde{O}\left( \frac{1}{\varepsilon_1^2 r} \right)$ and*

$$K_{t_0+t}(\overline{U}_{t_0} + U^*) \leq \varepsilon_0 + \varepsilon_1 \tag{D.73}$$

To prove this proposition, we need the following lemma:

**Proposition D.9.** *Suppose $g_t$ satisfies that $|g_t(z + \zeta) + g_t(z - \zeta) - 2g_t(z)| \leq \delta$ for some $\delta \lesssim 1$. Then, we have that*

$$\widehat{L}_{\overline{\mathcal{M}}_1}(u, U) \geq \log 2 - O(\delta) - O(\log d/\sqrt{qN}) \tag{D.74}$$

*And moreover, if $\widehat{L}_{\overline{\mathcal{M}}_1}(u, U) \leq \log 2 + O(\delta')$ for some $\delta' \geq \delta$, then the prediction of $g_t$ on $z - \zeta, z, z + \zeta$ satisfies $|g_t(z - \zeta)|, |g_t(z + \zeta)|, |g_t(z)| = O(\sqrt{\delta' + \log d/\sqrt{qN}})$.*

*Proof.* For convenience, let us denote $g_t(z + \delta) = u, g_t(z - \delta) = v, g_t(z) = (u + v)/2 + \gamma$. By our assumption, we have that $|\gamma| \leq \delta$.

Let $h(z) := -\log \frac{1}{1+e^{-z}}$. We have that w.h.p, for $c = O(\log d/\sqrt{qN})$,

$$4L_{\overline{\mathcal{M}}_1}(u, U) \geq [h(-u) + h(-v) + 2h((u + v)/2 + \gamma)] \cdot (1 - c) \tag{D.75}$$

$$= [\Delta + 2h(-(u + v)/2) + 2h((u + v)/2 + \gamma)] \cdot (1 - c) \tag{D.76}$$

where $\Delta$ is defined as

$$\Delta = h(-u) + h(-v) - 2h(-(u + v)/2) \geq 0 \qquad \text{(by convexity of } h\text{)}$$

and the factor of $1 - c$ comes from the fact that the fraction of examples that are $z - \zeta, z + \zeta, z$ will be $1/4 \pm O(\log d/\sqrt{qN})$, $1/4 \pm O(\log d/\sqrt{qN})$, $1/2 \pm O(\log d/\sqrt{qN})$, respectively, w.h.p. Since the function $h(z)$ is a 2-Lip function, we know that

$$|h((u + v)/2 + \gamma) - h((u + v)/2)| \leq 2\gamma \tag{D.77}$$

It follows that

$$\begin{aligned}
4L_{\bar{\mathcal{M}}_1}(u, U) &\geq (\Delta + 2h(-(u + v)/2) + 2h((u + v)/2 + \gamma))(1 - c) \\
&\geq (2h(-(u + v)/2) + 2h((u + v)/2) - 4\gamma)(1 - c) \\
&\qquad\qquad \text{(because } \Delta \geq 0 \text{ and equation (D.77))} \\
&\geq 4\log 2 - 4\gamma - O(\log d/\sqrt{qN}) \qquad\qquad \text{(by convexity of } h) \\
&\geq 4\log 2 - O(\delta) - O(\log d/\sqrt{qN})
\end{aligned}$$

The equation above together with the assumption $\widehat{L}_{\bar{\mathcal{M}}_1}(u, U) \leq \log 2 + O(\delta')$ implies that

$$4\log 2 + O(\delta') \geq 4L_{\bar{\mathcal{M}}_1}(u, U) \geq (\Delta + 2h((u + v)/2) + 2h(-(u + v)/2) - O(\delta))(1 - c) \tag{D.78}$$

which implies that $h((u + v)/2) + h(-(u + v)/2) - 2h(0) + \Delta \leq O(\delta') + O(c)$. It follows that $h((u + v)/2) + h(-(u + v)/2) - 2h(0) \leq O(\delta') + O(c)$ and $\Delta \leq O(\delta') + O(c)$. Now we note that By the strict convexity of $h(z)$, we
can easily conclude that $|u|, |v| \leq O(\sqrt{\delta' + c})$. $\qquad\square$

Next, we will bound $\varepsilon_0$ and the value of $g_{t_0}$. This allows us to conclude that $g_{t_0}$ is small, so that it is easy to "unlearn" once the learning rate is annealed.

**Lemma D.10.** *Suppose the condition in Lemma 4.1 holds. Then*

$$|g_{t_0}(z)|, |g_{t_0}(z + \zeta)|, |g_{t_0}(z - \zeta)| \leq O(\sqrt{\varepsilon_1/q}) \tag{D.79}$$
$$\varepsilon_0 = O(\sqrt{\varepsilon_1/q}) \tag{D.80}$$

*Proof of Lemma D.10.* Since $L_{t_0} \leq q\log 2 + \varepsilon_1$, we know that $\widehat{L}_{\bar{\mathcal{M}}_1}(u, U_{t_0}) \leq \log 2 + 2\varepsilon_1/q$. Applying Proposition D.9 with $\delta' = \varepsilon_1$ and $\delta = O(r^2/\lambda) = O(\varepsilon_1)$, we have that $|g_{t_0}(z)|, |g_{t_0}(z + \zeta)|, |g_{t_0}(z - \zeta)| \leq O(\sqrt{\varepsilon_1/q})$ and $\widehat{L}_{\bar{\mathcal{M}}_1}(u, U_{t_0}) \geq \log 2 - \varepsilon_1$.
Hence we have that (since $\ell$ is 2-Lipschitz)

$$\varepsilon_0 = \frac{1}{N} \sum_{i \in \mathcal{M}_1} \ell(r_{t_0}; (x^{(i)}, y^{(i)})) \tag{D.81}$$

$$\leq \frac{1}{N} \sum_{i \in \mathcal{M}_1} \ell(r_{t_0} + g_{t_0}; (x^{(i)}, y^{(i)})) + \frac{2}{N} \sum_{i \in \mathcal{M}_1} |g_{t_0}(x^{(i)})_2| \tag{D.82}$$

$$\leq \left(L_{t_0} - q\widehat{L}_{\bar{\mathcal{M}}_1}(u, U_{t_0})\right) + O(\sqrt{\varepsilon_1/q}) \tag{D.83}$$

$$\leq O(\sqrt{\varepsilon_1/q}) \tag{D.84}$$

$\qquad\square$

Now we will complete the proof of Proposition D.8.

*Proof of Proposition D.8.* Let us define sets $\mathcal{E}_1, \mathcal{E}_2, \mathcal{E}_3$ as the following:

$$\mathcal{E}_1 = \{i \in [m] \mid \langle [V_{t_0}]_i, z - \zeta \rangle \geq 0, \langle [V_{t_0}]_i, z \rangle \geq 0, \langle [V_{t_0}]_i, z + \zeta \rangle < 0\} \tag{D.85}$$
$$\mathcal{E}_2 = \{i \in [m] \mid \langle [V_{t_0}]_i, z - \zeta \rangle \geq 0, \langle [V_{t_0}]_i, z \rangle < 0, \langle [V_{t_0}]_i, z + \zeta \rangle < 0\} \tag{D.86}$$
$$\mathcal{E}_3 = \{i \in [m] \mid \langle [V_{t_0}]_i, z - \zeta \rangle < 0, \langle [V_{t_0}]_i, z \rangle < 0, \langle [V_{t_0}]_i, z + \zeta \rangle \geq 0\} \tag{D.87}$$

Let us define weight matrix $V^* \in \mathbb{R}^{m \times d}$ as:

$$V_i^* = \begin{cases} \frac{20c \log(1/\varepsilon_1)}{r\varepsilon_1} v_i z & \text{if } i \in \mathcal{E}_1; \\ -\frac{40c \log(1/\varepsilon_1)}{r\varepsilon_1} v_i z & \text{if } i \in \mathcal{E}_2; \\ -\frac{20c \log\log(1/\varepsilon_1)}{r\varepsilon_1} v_i z & \text{if } i \in \mathcal{E}_3; \\ 0 & \text{otherwise.} \end{cases} \tag{D.88}$$

for some sufficiently large universal constant $c$.

Note that the random noise vector $[\widetilde{V}_{t_0}]_i$ will satisfy the condition for set $\mathcal{E}_i$ with probability proportional to the angle between $z-\zeta$ and $z$, which is $r \pm O(r^2)$ by Taylor approximation of arcsin. Thus, as $V_{t_0}$ and $\widetilde{V}_{t_0}$ differ in at most $\varepsilon_s m$ activations, w.h.p., $|\mathcal{E}_1|, |\mathcal{E}_2|, |\mathcal{E}_3| = \frac{1}{2\pi} rm \pm \widetilde{O}\left(r^2 m + \sqrt{m}\right) \pm \varepsilon_s m$. This implies that

$$\|V^*\|_F^2 = \widetilde{O}\left(\frac{1}{r\varepsilon_1^2}\right) \tag{D.89}$$

Now, for $x_2 = z - \zeta$, we have that

$$N_{V_{t_0}}(v, V^*, z - \zeta) = \frac{1}{m}\left(|\mathcal{E}_1|\frac{20c \log(1/\varepsilon_1)}{r\varepsilon_1} - \frac{40c \log(1/\varepsilon_1)}{r\varepsilon_1}|\mathcal{E}_2|\right) \leq -2c \log(1/\varepsilon_1)/\varepsilon_1 \tag{D.90}$$

and for $x_2 = z + \zeta$, we have that

$$N_{V_{t_0}}(v, V^*, z + \zeta) = -\frac{1}{m}|\mathcal{E}_3|\frac{20c \log(1/\varepsilon_1)}{r\varepsilon_1} \leq -2c \log(1/\varepsilon_1)/\varepsilon_1 \tag{D.91}$$

Now, for $x_2 = z$, we have that

$$N_{V_{t_0}}(v, V^*, z) = \frac{1}{m}|\mathcal{E}_1|\frac{20c \log(1/\varepsilon_1)}{r\varepsilon_1} \geq 2c \log(1/\varepsilon_1)/\varepsilon_1 \tag{D.92}$$

Hence we can also easily conclude that for every $x_2 \in \{\alpha(z - \zeta), \alpha z, \alpha(z + \zeta)\}$,

$$y N_{V_{t_0}}(v, V^*, x_2) \geq \frac{2c \log(1/\varepsilon_1)\|x_2\|_2}{\varepsilon_1} \tag{D.93}$$

Note that for every $i \in [m]$,

$$|\langle V_i^*, x_2 \rangle| \leq \frac{1}{\sqrt{m}} \widetilde{O}\left(\frac{1}{\varepsilon_1 r}\right) \tag{D.94}$$

Now applying Lemma B.3, with $\eta_2 = O(\eta_1 \lambda^2 (\varepsilon_1 r)^2)$, we have that for every $x_2$, w.h.p. $\|\mathbb{1}([V_{t_0+t}]x_2) - \mathbb{1}([V_{t_0}]x_2)\|_1 \lesssim \lambda \varepsilon_1 rm$. This implies that for every $t \leq \frac{1}{\eta_2 \lambda}$ and every $x_2 \in \{z - \delta, z + \delta, z\}$, w.h.p.

$$\left|\sum_{i \in [m]} v_i \langle V_i^*, x_2 \rangle \left[\mathbb{1}([V_{t_0+t}]_i x_2) - \mathbb{1}([V_{t_0}]_i x_2)\right]\right| \leq \frac{1}{m}\widetilde{O}\left(\frac{1}{\varepsilon_1 r}\right) \times O(\lambda \varepsilon_1 rm) \leq 1 \tag{D.95}$$

Combining with (D.93), this gives us

$$y N_{V_{t_0+t}}(v, V^*; x_2) = y\left(\sum_{i \in [m]} v_i \langle V_i^*, x_2 \rangle \mathbb{1}([V_{t_0+t}]_i x_2)\right) \geq \frac{c\|x_2\|_2}{\varepsilon_1} \log \frac{1}{\varepsilon_1} \tag{D.96}$$

On the other hand we have that by Lemma D.10, it holds that

$$|N_{V_{t_0}}(v, \overline{V}_{t_0}; x_2)| \leq |g_{t_0}(x_2)| + |N_{V_{t_0}}(v, \overline{V}_{t_0}; x_2) - N_{V_{t_0}}(v, V_{t_0}; x_2)| \tag{D.97}$$

$$\leq |g_{t_0}(x_2)| + |N_{V_{t_0}}(v, \widetilde{V}_{t_0}; x_2)| \tag{D.98}$$

$$\lesssim |g_{t_0}(x_2)| + \frac{\varepsilon_s}{\lambda} + \tau_0 \log d \leq O(1) \qquad \text{(applying Proposition A.5)}$$

Thus, we also have

$$\left| y N_{V_{t_0+t}}(v, \overline{V}_{t_0}; x_2) \right| = \left| \left( \sum_{i \in [m]} v_i \langle [\overline{V}_{t_0}]_i, x_2 \rangle \mathbb{1}([V_{t_0+t}]_i x_2) \right) \right| \tag{D.99}$$

$$\leq \left| \left( \sum_{i \in [m]} v_i \langle [\overline{V}_{t_0}]_i, x_2 \rangle \mathbb{1}([V_{t_0}]_i x_2) \right) \right| + \left| \left( \sum_{i \in [m]} v_i \langle [\overline{V}_{t_0}]_i, x_2 \rangle \left[ \mathbb{1}([V_{t_0+t}]_i x_2) - \mathbb{1}([V_{t_0}]_i x_2) \right] \right) \right| \tag{D.100}$$

Now the first term equals $|N_{V_{t_0}}(v, \overline{V}_{t_0}; x_2)| = O(1)$, and the second term is bounded by

$$\left| \left( \sum_{i \in [m]} v_i \langle [\overline{V}_{t_0}]_i, x_2 \rangle \left[ \mathbb{1}([V_{t_0+t}]_i x_2) - \mathbb{1}([V_{t_0}]_i x_2) \right] \right) \right| \leq \frac{1}{m} O\left( \frac{1}{\lambda} \right) \times O(\lambda \varepsilon_1 r m)$$

using Proposition A.3 to upper bound $\|[\overline{V}_{t_0}]_i\|_2$. Thus, it follows that $|y N_{V_{t_0+t}}(v, \overline{V}_{t_0}; x_2)| = O(1)$.
It follows that for every $x_2 \in \{z - \zeta, z, z + \zeta\}$ and its corresponding label $y$, as long as $\|x_2\|_2 \geq \varepsilon_1$,

$$y N_{V_{t_0+t}}(v, \overline{V}_{t_0} + V^*; x_2) \geq y N_{V_{t_0+t}}(v, V^*; x_2) - \left| y N_{V_{t_0+t}}(v, \overline{V}_{t_0}; x_2) \right| \tag{D.101}$$

$$\geq c \log(1/\varepsilon_1) - \left| y N_{V_{t_0+t}}(v, \overline{V}_{t_0}; x_2) \right| \tag{D.102}$$

$$\geq 3 \log(1/\varepsilon_1) \qquad \text{(choosing } c \text{ sufficiently large)}$$

Now we can compute

$$\left| N_{W_{t_0+t}}(w, \overline{W}_{t_0}, x_1) - r_{t_0}(x_1) \right| \tag{D.103}$$

$$\leq \left| N_{W_{t_0+t}}(w, \overline{W}_{t_0}, x_1) - N_{W_{t_0}}(w, \overline{W}_{t_0}, x_1) \right| + \left| N_{W_{t_0}}(w, \widetilde{W}_{t_0}, x_1) \right| \tag{D.104}$$

$$\leq \frac{1}{\sqrt{m}} \| \mathbb{1}(W_{t_0+t} x_1) - \mathbb{1}(W_{t_0} x_1) \|_1 \max_i \|[\overline{W}_{t_0}]_i\|_2 \|x_1\|_2 + \left| N_{W_{t_0}}(w, \widetilde{W}_{t_0}, x_1) \right|$$

$$\text{(by Lemma B.3 and } \|[\overline{W}_{t_0}]_i\|_2 = O\left( \frac{1}{\sqrt{m}} \frac{1}{\lambda} \right) \text{ from Proposition A.2)}$$

$$\lesssim \frac{\varepsilon_s}{\lambda} + \tau_0 \log d \leq q \varepsilon_1 \tag{D.105}$$

The last inequality follows from our choice of parameters such that $\tau_0 \log d \leq q \varepsilon_1$. Putting together Eq (D.101) and (D.103) and defining $U^* = (0, V^*)$, we have that

$$K_{t_0+t}(\overline{U}_{t_0} + U^*) = K_{t_0+t}((\overline{W}_{t_0}, \overline{V}_{t_0} + V^*)) \tag{D.106}$$

$$\leq \frac{|\mathcal{M}_1|}{N} \widehat{L}_{\mathcal{M}_1}(r_{t_0}) + O(q \varepsilon_1) + \frac{|\bar{\mathcal{M}}_1|}{N} \widehat{L}_{\bar{\mathcal{M}}_1}(N_{V_{t_0+t}}(v, \overline{V}_{t_0} + V^*; *))$$

$$\text{(by definition of } \mathcal{M}_1 \text{ and Lipschitz-ness of } \ell)$$

$$\leq \varepsilon_0 + \varepsilon_1 \tag{D.107}$$

This completes the proof.

$\square$

*Proof of Lemma 4.3.* By proposition D.8, there exists $V^*$ with $\|V^*\|_F^2 \leq \widetilde{O}\left( \frac{1}{r \varepsilon_1^2} \right)$ such that for every $t \leq \frac{1}{\eta_2 \lambda}$,

$$K_{t_0+t}((\overline{W}_{t_0}, \overline{V}_{t_0} + V^*)) \leq \varepsilon_0 + \varepsilon_1 \tag{D.108}$$

By Theorem B.2, with $z^* = (\overline{W}_{t_0}, V^*)$, starting from $z_0 = (\overline{W}_{t_0}, \overline{V}_{t_0})$, we can take $R^2 = \widetilde{O}\left( \frac{1}{r \varepsilon_1^2} \right)$, $L = 1, \mu = \varepsilon_1$ to conclude that the algorithm converges to $\varepsilon_0 + 2\varepsilon_1$ in $\widetilde{O}\left( \frac{1}{\eta_2 r \varepsilon_1^3} \right)$ iterations. Applying Lemma D.10 to bound $\varepsilon_0$ completes the proof. $\square$

## D.5 Proof of Lemma 4.4

By the 1-Lipschitzness of logistic loss, we know that

$$\left| \widehat{L}_{\mathcal{M}_1}(r_{t_0}) - \widehat{L}_{\mathcal{M}_1}(r_{t_0+t}) \right| \tag{D.109}$$

$$= \left| \frac{1}{|\mathcal{M}_1|} \sum_{i \in \mathcal{M}_1} \left( \ell(r_{t_0}; (x^{(i)}, y^{(i)})) - \ell(r_{t_0+t}; (x^{(i)}, y^{(i)})) \right) \right| \tag{D.110}$$

$$\leq \frac{1}{|\mathcal{M}_1|} \sum_{i \in \mathcal{M}_1} \left| r_{t_0}(x_1^{(i)}) - r_{t_0+t}(x_1^{(i)}) \right| \tag{D.111}$$

To bound this term, we can directly use Cauchy-Shwartz and obtain that:

$$\sum_{i \in \mathcal{M}_1} \left| r_{t_0+t}(x_1^{(i)}) - r_{t_0}(x_1^{(i)}) \right| \tag{D.112}$$

$$\leq \sqrt{N} \sqrt{ \sum_{i \in \mathcal{M}_1} \left( r_{t_0+t}(x_1^{(i)}) - r_{t_0}(x_1^{(i)}) \right)^2 } \tag{D.113}$$

We can further bound $r_{t_0+t}(x_1^{(i)}) - r_{t_0}(x_1^{(i)})$ by applying Lemma B.3, as from our choice of parameters $\eta_2 \leq \eta_1 \varepsilon_1^4 \lambda^2, \varepsilon_s/\lambda \leq \varepsilon_1^2, \tau_0 \log d \leq \varepsilon_1^2$:

$$\left| r_{t_0+t}(x_1^{(i)}) - r_{t_0}(x_1^{(i)}) \right| \tag{D.114}$$

$$\leq \left| N_{W_{t_0+t}}(w, W_{t_0+t}, x_1^{(i)}) - N_{W_{t_0}}(w, \overline{W}_{t_0+t}, x_1^{(i)}) \right| + \tag{D.115}$$

$$\left| N_{W_{t_0}}(w, \overline{W}_{t_0+t}, x_1^{(i)}) - N_{W_{t_0}}(w, \overline{W}_{t_0}, x_1^{(i)}) \right| + \left| N_{W_{t_0}}(w, \widetilde{W}_{t_0}, x_1^{(i)}) \right| \tag{D.116}$$

$$\leq \left| N_{W_{t_0}}(w, \overline{W}_{t_0+t}, x_1^{(i)}) - N_{W_{t_0}}(w, \overline{W}_{t_0}, x_1^{(i)}) \right| + O\left( \frac{1}{\lambda} \times \left( \sqrt{\frac{\eta_2}{\eta_1}} + \varepsilon_s \right) + \tau_0 \log d \right)$$
$$\text{(by Lemma B.3 and Proposition A.5)}$$

$$\leq \left| N_{W_{t_0}}(w, \overline{W}_{t_0+t}, x_1^{(i)}) - N_{W_{t_0}}(w, \overline{W}_{t_0}, x_1^{(i)}) \right| + \varepsilon_1^2 \tag{D.117}$$

Now, let us denote $X = (x^{(i)})_{i \in [N]}$ as the data matrix. By the standard Gaussian matrix spectral norm bound we know that w.h.p. $\|X\|_2^2 \leq 10\frac{N}{d}$.

This gives us:

$$\sqrt{N} \sqrt{ \sum_{i \in \mathcal{M}_1} \left( N_{W_{t_0}}(w, \overline{W}_{t_0+t}, x_1^{(i)}) - N_{W_{t_0}}(w, \overline{W}_{t_0}, x_1^{(i)}) \right)^2 }$$

$$\leq \sqrt{N} \sqrt{ \|\overline{W}_{t_0+t} - \overline{W}_{t_0}\|_F^2 \|X\|_2^2 } \qquad \text{(expanding the expression of } N_{W_{t_0}}(w, W_{t_0+t}, x_1^{(i)}))$$

$$\leq \sqrt{N} \sqrt{ 10 \left( \|\overline{W}_{t_0+t} - \overline{W}_{t_0}\|_F^2 \right) \frac{N}{d} } \tag{D.118}$$

$$\leq N \tilde{O}\left( \frac{1}{\sqrt{dr}\varepsilon_1} \right) \leq N\varepsilon_1 \tag{D.119}$$

Here in (D.119), we use the assumption $dr \geq \tilde{\Omega}\left( \frac{1}{\varepsilon_1^4} \right)$ in Theorem 3.4 along with the fact that by Lemma 4.3, we have that

$$\left\| \overline{W}_{t_0+t} - \overline{W}_{t_0} \right\|_F^2 \leq \tilde{O}\left( \frac{1}{r\varepsilon_1^2} \right) \tag{D.120}$$

Thus, using (D.119), it follows that

$$\sum_{i \in \mathcal{M}_1} \left| r_{t_0+t}(x_1^{(i)}) - r_{t_0}(x_1^{(i)}) \right|$$

$$\lesssim \sqrt{N} \sqrt{\sum_{i \in \mathcal{M}_1} \left( N_{W_{t_0}}(w, \overline{W}_{t_0+t}, x_1^{(i)}) - N_{W_{t_0}}(w, \overline{W}_{t_0}, x_1^{(i)}) \right)^2} + N\varepsilon_1^2 \leq N\varepsilon_1$$

By (D.109) and our definition of $\varepsilon_0$ as

$$\varepsilon_0 := \frac{|\mathcal{M}_1|}{N} \widehat{L}_{\mathcal{M}_1}(r_{t_0}) = (1-q)\widehat{L}_{\mathcal{M}_1}(r_{t_0}) \tag{D.121}$$

we must have

$$\left| \widehat{L}_{\mathcal{M}_1}(r_{t_0+t}) - \frac{\varepsilon_0}{1-q} \right| \leq \varepsilon_1/2 \tag{D.122}$$

Using the bound on $\varepsilon_0$ that $\varepsilon_0 = O(\sqrt{\varepsilon_1/q})$ by Lemma D.10, we conclude the bound on $\widehat{L}_{\mathcal{M}_1}(r_{t_0+t})$.

In the end, by $\widehat{L}_{\bar{\mathcal{M}}_1}(g_{t_0+t}) \leq \widehat{L}_{t_0+t}$ and the assumption that $\widehat{L}_{t_0+t} \leq O(\sqrt{\varepsilon_1/q})$, it must hold that (since $|\bar{\mathcal{M}}_1| = qN$)

$$\widehat{L}_{\bar{\mathcal{M}}_1}(g_{t_0+t}) \lesssim \sqrt{\frac{\varepsilon_1}{q^3}} \tag{D.123}$$

so we can complete the proof.

# E    Proofs for Small Learning Rate

## E.1    Proof of Lemma 5.1

We first show the following Lemma:

**Lemma E.1.** *In the setting of theorem 3.5, there exists a solution $U^\star$ satisfying a) $\|U^\star\|_F^2 \leq \widetilde{O}(\frac{1}{\varepsilon_2'^2 r} + Np)$ and b) for every $t \leq \frac{1}{\eta_2 \lambda}$,*

$$K_t(U^\star) \leq \varepsilon_2' \tag{E.1}$$

*Proof of Lemma E.1.* We can construct the matrix $U^\star$ as follows: let $X = (x_1^i)_{i \in \bar{\mathcal{M}}_2} \in \mathbb{R}^{d \times Np}$ and $Y = (y^{(i)})_{i \in \bar{\mathcal{M}}_2} \in \mathbb{R}^{1 \times Np}$. If we define $s = X(X^\top X)^{-1} y^\top \in \mathbb{R}^{d \times 1}$, we know that $s^\top X = y$ with $\|s\|_2 = O(\sqrt{Np})$. Thus, we can define $V^*$ as in Lemma 4.3 with $t_0 = 0$, and $W_i^* = 10 \log \frac{1}{\varepsilon_2'} s w_i$, and we can see that for every $t \leq \frac{1}{\eta_2 \lambda}$, it holds that

$$K_t((W^*, V^*)) \leq \varepsilon_2' \tag{E.2}$$

$\square$

To prove Lemma 5.1, we can apply an identical analysis as 4.3 to show that for $t' = \widetilde{O}\left(\frac{1}{\eta_2 \varepsilon_2'^3 r}\right)$, $\widehat{L}_{\mathcal{M}_2}(U_{t'}) \leq \varepsilon_2'$. The rest of the proof follows from combining Theorem B.2 and Lemma E.1.

## E.2    Proof of Lemma 5.2

We will use the following Lemma from [6].

**Lemma E.2** (Lemma 6.3 of [6]). *For every $v_1, v_2, v_3$, let $g \sim \mathcal{N}(0, I)$ in $\mathbb{R}^d$, then we have:*

$$\mathbb{E}_g \left[ \|v_1 \mathbb{1}(\langle g, z - \zeta \rangle)(z - \zeta) + v_2 \mathbb{1}(\langle g, z + \zeta \rangle)(z + \zeta) + v_3 \mathbb{1}(\langle g, z \rangle)z\|_2^2 \right] \tag{E.3}$$

$$\gtrsim r \left( v_1^2 + v_2^2 + v_3^2 \right) \tag{E.4}$$

Recall the expression $\rho_t$ defined in (B.10). We first prove Lemma B.4 here, which says that if $\rho_t$ is large (which means the loss is large as well), then the total gradient norm has to be big.

*Proof of Lemma B.4.* For notation simplicity, let's fix $t$ and let

$$Q_j = \ell'_{j,t} \tag{E.5}$$

The gradient with respect to $V$ can be computed by

$$\nabla_{[V]_k} \widehat{L}(U_t) = \frac{1}{N} \sum_{j \in \mathcal{M}_2} Q_j v_k \mathbb{1}(\langle [V_t]_k, x_2^{(j)} \rangle) x_2^{(j)} \tag{E.6}$$

Let us denote the set $\mathcal{S}_{2,1}^{(\alpha_0)}, \mathcal{S}_{2,2}^{(\alpha_0)}, \mathcal{S}_{2,3}^{(\alpha_0)}$ as:

$$\mathcal{S}_{2,1}^{(\alpha_0)} = \left\{ j \in [m] \mid x_2^{(j)} = \alpha_j(z - \zeta) \text{ for some } \alpha_j \geq \alpha_0 \right\} \tag{E.7}$$

$$\mathcal{S}_{2,2}^{(\alpha_0)} = \left\{ j \in [m] \mid x_2^{(j)} = \alpha_j(z + \zeta) \text{ for some } \alpha_j \geq \alpha_0 \right\} \tag{E.8}$$

$$\mathcal{S}_{2,3}^{(\alpha_0)} = \left\{ j \in [m] \mid x_2^{(j)} = \alpha_j z \text{ for some } \alpha_j \geq \alpha_0 \right\} \tag{E.9}$$

We then have that

$$N m v_k \nabla_{[V]_k} L_t \tag{E.10}$$

$$= \sum_{j \in \mathcal{S}_{2,1}^{(0)}} \alpha_j Q_j \mathbb{1}(\langle [V_t]_k, z - \zeta \rangle)(z - \zeta) + \sum_{j \in \mathcal{S}_{2,2}^{(0)}} \alpha_j Q_j \mathbb{1}(\langle [V_t]_k, z + \zeta \rangle)(z + \zeta) \tag{E.11}$$

$$+ \sum_{j \in \mathcal{S}_{2,3}^{(0)}} \alpha_j Q_j \mathbb{1}(\langle [V_t]_k, z \rangle) z \tag{E.12}$$

For each $k \in [m]$, let us define

$$\tilde{L}_k \triangleq \sum_{j \in \mathcal{S}_{2,1}^{(0)}} \alpha_j Q_j \mathbb{1}(\langle [\tilde{V}_t]_k, z - \zeta \rangle)(z - \zeta) + \sum_{j \in \mathcal{S}_{2,2}^{(0)}} \alpha_j Q_j \mathbb{1}(\langle [\tilde{V}_t]_k, z + \zeta \rangle)(z + \zeta) \tag{E.13}$$

$$+ \sum_{j \in \mathcal{S}_{2,3}^{(0)}} \alpha_j Q_j \mathbb{1}(\langle [\tilde{V}_t]_k, z \rangle) z \tag{E.14}$$

i.e., the loss gradient using activations computed by the noise component of $V_t$ scaled by a factor of $N m v_k$.

By the Geometry of ReLU Lemma E.2, we have that w.h.p.

$$\mathbb{E}_{[\tilde{V}_t]_k} \left[ \left\| \tilde{L}_k \right\|_2^2 \right] \geq r\Omega \left( \left( \sum_{j \in \mathcal{S}_{2,1}^{(0)}} \alpha_j Q_j \right)^2 + \left( \sum_{j \in \mathcal{S}_{2,2}^{(0)}} \alpha_j Q_j \right)^2 + \left( \sum_{j \in \mathcal{S}_{2,3}^{(0)}} \alpha_j Q_j \right)^2 \right) \tag{E.15}$$

$$\geq r\Omega \left( \left( \sum_{j \in \mathcal{M}_2} \alpha_j |Q_j| \right)^2 \right) \tag{E.16}$$

Where the last inequality is obtained since for every $j \in \mathcal{S}_{2,j'}^{(0)}$, $Q_j$ has the same sign.

Since each $[\tilde{V}_t]_k$ are independent and $|\alpha_j Q_j|, \|z\|_2, \|\zeta\|_2 = O(1)$, by concentration, we know that taking a union bound over all choices of $Q_j$, w.h.p.

$$\|\tilde{L}\|_F^2 \geq mr\Omega \left( \left( \sum_j \alpha_j |Q_j| \right)^2 \right) - \widetilde{O}(m^{1/2} N^4) \tag{E.17}$$

where $\tilde{L}$ denotes the matrix where each $\tilde{L}_k$ is a row. By Coupling Lemma A.8, we note that as

$$\frac{1}{N^2m}\|\tilde{L}\|_F^2 - \|\nabla\hat{L}(U_t)\|_F^2 \lesssim \frac{1}{Nm}\sum_k\sum_j Q_j^2|\mathbb{1}(\langle[V_t]_k, x_2^{(j)}\rangle) - \mathbb{1}(\langle[\tilde{V}_t]_k, x_2^{(j)}\rangle)| \lesssim O(\varepsilon_s)$$

we therefore also have w.h.p.:

$$\|\nabla\hat{L}(U_t)\|_F^2 \geq \frac{1}{N^2m}\|\tilde{L}\|_F^2 - O(\varepsilon_s) \tag{E.18}$$

$$\geq \frac{r}{N^2}\Omega\left(\left(\sum_j \alpha_j|Q_j|\right)^2\right) - \tilde{O}(m^{-1/2}N^2) - O(\varepsilon_s) \tag{E.19}$$

Note that $\alpha_j \sim U(0,1)$, and therefore for every fixed $\alpha_0 \geq \frac{1}{\sqrt{N}}$, w.h.p. there are $O(N\alpha_0)$ many $\alpha_j$ such that $\alpha_j \leq \alpha_0$. For each of them, we also know that $|Q_j| \leq 1$, which implies that

$$\left(\sum_j \alpha_j|Q_j|\right)^2 \geq \alpha_0^2\left(\sum_{j:\alpha_j\geq\alpha_0}|Q_j|\right)^2 \tag{E.20}$$

$$\geq \alpha_0^2\left(\left(\sum_j|Q_j|\right) - O(N\alpha_0^2)\right)^2 \tag{E.21}$$

$$\geq \alpha_0^2(N(\rho_t - O(\alpha_0^2)))^2 \tag{E.22}$$

Picking $\alpha_0 = \Theta(\sqrt{\rho_t})$, we complete the proof by our choice of $m \geq N^{10}\frac{1}{(\lambda\tau_0)^4}$. □

Now we prove Proposition B.5, which bounds the number of iterations in which $\rho_t$ can be large.

*Proof of Proposition B.5.* Consider the function $\mathcal{F}_s(x) := N_{U_0}(u, \overline{U}_s; x)$, and let us define $\mathcal{G}_{s+1}(x) := N_{U_0}(u, \overline{U}_s - \frac{\eta_2}{1-\eta_2\lambda}\nabla\hat{L}(U_s); x)$. We have that since $\overline{U}_{s+1} = (1-\eta_2\lambda)\overline{U}_s - \eta_2\nabla\hat{L}(U_s)$,

$$\hat{L}(\mathcal{F}_{s+1}) = \hat{L}((1-\eta_2\lambda)\mathcal{G}_{s+1}) \leq (1+\eta_2\lambda)\hat{L}(\mathcal{G}_{s+1}) \tag{E.23}$$

Here we use the fact that for logistic loss $\ell$, $\ell((1-\alpha)z) \leq (1+\alpha)\ell(z)$ for every $z \in \mathbb{R}, \alpha \in [0, 0.1]$.

Now, by standard gradient descent analysis, we have that (as the logistic loss has Lipschitz derivative and the data have bounded norm):

$$\hat{L}(\mathcal{G}_{s+1}) \leq \hat{L}(\mathcal{F}_s) - \frac{\eta_2}{1-\eta_2\lambda}\langle\nabla\hat{L}(\mathcal{F}_s), \nabla\hat{L}(U_s)\rangle + 2\eta_2^2\|\nabla\hat{L}(U_s)\|_F^2 \tag{E.24}$$

$$\leq \hat{L}(\mathcal{F}_s) - \frac{\eta_2}{1-\eta_2\lambda}\langle\nabla\hat{L}(\mathcal{F}_s), \nabla\hat{L}(U_s)\rangle + O(\eta_2^2) \quad \text{(by Proposition A.2)}$$

Next, we will bound $\|\nabla\hat{L}(U_s) - \nabla\hat{L}(\mathcal{F}_s)\|_F$. We can compute

$$\|\nabla\hat{L}(U_s) - \nabla\hat{L}(\mathcal{F}_s)\|_F^2 \tag{E.25}$$

$$\leq \frac{1}{N^2m}\sum_{k\in[m]}\|\sum_j\left(\ell'(-y^{(j)}N_{U_s}(u, U_s; x^{(j)}))\mathbb{1}([U_s]_kx^{(j)}) - \right. \tag{E.26}$$

$$\ell'(-y^{(j)}N_{U_0}(u, \overline{U}_s; x^{(j)}))\mathbb{1}([U_0]_kx^{(j)}))x^{(j)}\|_2^2 \tag{E.27}$$

$$\leq \frac{1}{Nm}\sum_{k\in[m]}\sum_j\|(\ell'(-y^{(j)}N_{U_s}(u, U_s; x^{(j)}))\mathbb{1}([U_s]_kx^{(j)}) - \tag{E.28}$$

$$\ell'(-y^{(j)}N_{U_0}(u, \overline{U}_s; x^{(j)}))\mathbb{1}([U_0]_kx^{(j)}))x^{(j)}\|_2^2 \tag{E.29}$$

where the last step followed via Cauchy-Schwarz. Now by the Lipschitzness of $\ell'$, we have the bound

$$\ell'(-y^{(j)}N_{U_s}(u,U_s;x^{(j)}))\mathbb{1}([U_s]_kx^{(j)}) - \ell'(-y^{(j)}N_{U_0}(u,\overline{U}_s;x^{(j)}))\mathbb{1}([U_0]_kx^{(j)}) \lesssim$$
$$|N_{U_s}(u,U_s;x^{(j)})) - N_{U_0}(u,\overline{U}_s;x^{(j)})| + |\mathbb{1}([U_s]_kx^{(j)}) - \mathbb{1}([U_0]_kx^{(j)})|$$

Plugging this back into (E.29), by the coupling Lemma B.3 we obtain the bound

$$\|\nabla\widehat{L}(U_s) - \nabla\widehat{L}(\mathcal{F}_s)\|_F^2 \lesssim \frac{1}{\lambda}(\varepsilon_s + \sqrt{\eta_2/\eta_1}) + \tau_0\log d := \varepsilon_c^2$$

This implies that for $\eta_2\lambda < 0.1$,

$$\widehat{L}(\mathcal{G}_{s+1}) \leq \widehat{L}(\mathcal{F}_s) - \frac{1}{2}\eta_2\|\nabla\widehat{L}(U_s)\|_F^2 + O(\eta_2^2 + \eta_2\varepsilon_c) \tag{E.30}$$

Hence, we have

$$\widehat{L}(\mathcal{F}_{s+1}) \leq (1+\eta_2\lambda)\widehat{L}(\mathcal{F}_s) - \frac{1}{2}(1+\eta_2\lambda)\eta_2\|\nabla\widehat{L}(U_s)\|_F^2 + O(\eta_2^2 + \eta_2\varepsilon_c) \tag{E.31}$$

which implies that for every $t \leq \frac{1}{\eta_2\lambda}$, as long as $\eta_2, \varepsilon_c = O(\lambda)$, we have:

$$\eta_2\sum_{s\leq t}\|\nabla\widehat{L}(U_s)\|_F^2 \lesssim \widehat{L}(\mathcal{F}_0) \lesssim 1 \tag{E.32}$$

By Lemma B.4, we have that if $\rho_t \geq {\varepsilon_2'}^2\varepsilon_3^2$, then $\|\nabla\widehat{L}(U_s)\|_F^2 \geq r{\varepsilon_2'}^8\varepsilon_3^8$. It follows that there will be at most $O(\frac{1}{r{\varepsilon_2'}^8\varepsilon_3^8\eta_2})$ such $t$.

$$\square$$

Finally, we complete the proof of Lemma 5.2 by noting that $\rho_t$ cannot be large for very many iterations, and therefore $W_t$ will not obtain much signal from the $\mathcal{P}$ component of examples in $\mathcal{M}_2$.

*Proof of Lemma 5.2.* We have,

$$\left\|\sum_{j\in\mathcal{M}_2}\nabla_W\widehat{L}_j(U_t)\right\|_2^2 = \sum_{k\in[m]}\left\|\sum_{j\in\mathcal{M}_2}\ell_{j,t}'w_k\mathbb{1}(\langle[W_t]_k,x_1^{(j)}\rangle)x_1^{(j)}\right\|_2^2$$

Now we note that the above can be reformulated as a matrix multiplication between the matrix of data $X$ and the vector with entry $\ell_{j,t}'w_k\mathbb{1}(\langle[W_t]_k,x_1^{(j)}\rangle)$ in the $j$-th coordinate for $j \in \mathcal{M}_2$ and 0 elsewhere. Thus,

$$\left\|\sum_{j\in\mathcal{M}_2}\nabla_W\widehat{L}_j(U_t)\right\|_2^2 \leq \sum_{k\in[m]}\|X\|_2^2\left(\sum_{j\in\mathcal{M}_2}\left(\ell_{j,t}'w_k\mathbb{1}(\langle[W_t]_k,x_1^{(j)}\rangle)\right)^2\right)$$
$$\text{(definition of spectral norm)}$$

$$\leq \sum_{k\in[m]}\|X\|_2^2\left(\sum_{j\in\mathcal{M}_2}\left(\ell_{j,t}'w_k\right)^2\right)$$

$$= \|X\|_2^2\sum_{j\in\mathcal{M}_2}\left(\ell_{j,t}'\right)^2 \qquad \text{(because } w_k \in \{\pm 1/\sqrt{m}\})$$

$$\lesssim \|X\|_2^2\sum_{j\in\mathcal{M}_2}|\ell_{j,t}'| \qquad \text{(because the } \ell \text{ is } O(1)\text{-Lipschitz)}$$

$$\lesssim N/d \cdot N\rho_t \tag{E.33}$$

The last line followed from the spectral norm bound on matrix $X$. Let $\mathcal{T}$ be defined as in Proposition B.5. It follows that

$$\left\|\overline{W}_t^{(2)}\right\|_F \leq \eta_2 \sum_{s \leq t} \left\|\left(\frac{1}{N}\sum_{j \in \mathcal{M}_2} \nabla_W \ell(f_s; (x^{(j)}, y^{(j)}))\right)\right\|_F \tag{E.34}$$

$$= \eta_2 \sum_{s \in \mathcal{T}} \left\|\left(\frac{1}{N}\sum_{j \in \mathcal{M}_2} \nabla_W \ell(f_s; (x^{(j)}, y^{(j)}))\right)\right\|_F + \tag{E.35}$$

$$\eta_2 \sum_{s \notin \mathcal{T}} \left\|\left(\frac{1}{N}\sum_{j \in \mathcal{M}_2} \nabla_W \ell(f_s; (x^{(j)}, y^{(j)}))\right)\right\|_F$$

$$\leq \eta_2 \sum_{s \in \mathcal{T}} \left\|\left(\frac{1}{N}\sum_{j \in \mathcal{M}_2} \nabla_W \ell(f_s; (x^{(j)}, y^{(j)}))\right)\right\|_F + \eta_2 t O\left(\frac{\varepsilon_2' \varepsilon_3}{\sqrt{d}}\right)$$

(by definition of $\mathcal{T}$ and equation (E.33))

Note that we can additionally bound the first term by $\eta_2 |\mathcal{T}| O(\frac{1}{\sqrt{d}})$ as $\rho_t \leq 1$ by the Lipschitzness of $\ell$. Thus, applying our bound on $|\mathcal{T}|$, we get

$$\left\|\overline{W}_t^{(2)}\right\|_F \leq O\left(\frac{1}{r\sqrt{d}\varepsilon_2'^8 \varepsilon_3^8} + \frac{\eta_2 \varepsilon_2' \varepsilon_3 t}{\sqrt{d}}\right) \tag{E.36}$$

Now the conclusion of the lemma follows by the assumption that $t = O(d/\eta_2 \varepsilon_2')$ and our choice of $\eta_2$ and $\frac{1}{\varepsilon_2'^8 \varepsilon_3^8 r} \leq \varepsilon_2' d$ in Theorem 3.5.

$\square$

## E.3 Proof of Lemma 5.3

We now prove the decomposition lemma of $\overline{W}_t$, Lemma B.6. Recall our definition of $\overline{W}_t^{(2)}$ as

$$\overline{W}_t^{(2)} = \frac{1}{N}\eta_2 \sum_{s \leq t}(1 - \eta_2\lambda)^{t-s}\sum_{i \in \mathcal{M}_2}\nabla_W \widehat{L}_{\{i\}}(U_s) \tag{E.37}$$

*Proof of Lemma B.6.* For each step, we know that for every $j \in [m]$,

$$\nabla_{W_j}\hat{L}(U_s) = w_j \frac{1}{N}\sum_{i \in [N]}\ell_{i,s}' \mathbb{1}([W_s]_j x_1^{(i)})x_1^{(i)}$$

Thus, multiplying by $\eta_2(1 - \eta_2\lambda)^{t-s}$ and summing, following our definition of $\overline{W}_t^{(2)}$ in (5.1), we get

$$[\overline{W}_t]_j = [\overline{W}_t^{(2)}]_j + w_j \frac{1}{N}\eta_2 \sum_{s \leq t}(1 - \eta_2\lambda)^{t-s}\sum_{i \in \bar{\mathcal{M}}_2}\ell_{i,s}' \mathbb{1}([W_s]_j x_1^{(i)})x_1^{(i)} \tag{E.38}$$

$$= [\overline{W}_t^{(2)}]_j + \tag{E.39}$$

$$w_j \frac{1}{N}\eta_2 \sum_{s \leq t}(1 - \eta_2\lambda)^{t-s} \cdot \left(\sum_{i \in \bar{\mathcal{M}}_2}\ell_{i,s}' \mathbb{1}([W_0]_j x_1^{(i)})x_1^{(i)} + \right. \tag{E.40}$$

$$\left.\sum_{i \in \bar{\mathcal{M}}_2}\ell_{i,s}'\left[\mathbb{1}([W_s]_j x_1^{(i)}) - \mathbb{1}([W_0]_j x_1^{(i)})\right]x_1^{(i)}\right) \tag{E.41}$$

Now we focus on bounding the bottom term. We can see that

$$\sum_{j\in[m]}\left\|w_j\frac{1}{N}\sum_{i\in\bar{\mathcal{M}}_2}\ell'_{i,s}\left[\mathbb{1}([W_s]_jx_1^{(i)})-\mathbb{1}([W_0]_jx_1^{(i)})\right]x_1^{(i)}\right\|_2^2 \tag{E.42}$$

$$\leq \frac{1}{mN}\sum_{j\in[m]}\sum_{i\in\bar{\mathcal{M}}_2}\left\|\ell'_{i,s}\left[\mathbb{1}([W_s]_jx_1^{(i)})-\mathbb{1}([W_0]_jx_1^{(i)})\right]x_1^{(i)}\right\|_2^2$$

(since $w_j=\pm 1/\sqrt{m}$ and by Cauchy-Schwarz)

$$\lesssim \frac{1}{mN}\sum_{i\in\bar{\mathcal{M}}_2}\sum_{j\in[m]}\left\|\left[\mathbb{1}([W_s]_jx_1^{(i)})-\mathbb{1}([W_0]_jx_1^{(i)})\right]x_1^{(i)}\right\|_2^2 \qquad \text{(by Lipschitzness of } \ell)$$

By Auxiliary Coupling Lemma B.3 with $t_0=0$, we know that for $s\leq\frac{1}{\eta_2\lambda}$, w.h.p.

$$\sum_{j\in[m]}\left\|\left[\mathbb{1}([W_s]_jx_1^{(i)})-\mathbb{1}([W_0]_jx_1^{(i)})\right]x_1^{(i)}\right\|_2^2 \leq \left\|\mathbb{1}(W_sx_1^{(i)})-\mathbb{1}(W_0x_1^{(i)})\right\|_1\|x_1^{(i)}\|_2^2 \tag{E.43}$$

$$\leq \widetilde{O}\left(\varepsilon_s m+\sqrt{\frac{\eta_2}{\eta_1}}m\right) \tag{E.44}$$

Thus, we have

$$\sum_{j\in[m]}\left\|w_j\frac{1}{N}\sum_{i\in\bar{\mathcal{M}}_2}\ell'_{i,s}\left[\mathbb{1}([W_s]_jx_1^{(i)})-\mathbb{1}([W_0]_jx_1^{(i)})\right]x_1^{(i)}\right\|_2^2 \tag{E.45}$$

$$\lesssim \frac{1}{mN}\sum_{i\in\bar{\mathcal{M}}_2}\sum_{j\in[m]}\left\|\left[\mathbb{1}([W_s]_jx_1^{(i)})-\mathbb{1}([W_0]_jx_1^{(i)})\right]x_1^{(i)}\right\|_2^2 \tag{E.46}$$

$$\leq \widetilde{O}\left(\varepsilon_s+\sqrt{\frac{\eta_2}{\eta_1}}\right) \tag{E.47}$$

Now, we can express the weight

$$[\overline{W}_t]_j = w_j\sum_{k\in\bar{\mathcal{M}}_2}\alpha_k x_1^{(k)}\mathbb{1}([W_0]_jx_1^{(k)})+[\overline{W}'_t]_j \tag{E.48}$$

for some real values $\{\alpha_k\}_{k\in\bar{\mathcal{M}}_2}$ with

$$\alpha_k = \eta_2\sum_{s\leq t}(1-\eta_2\lambda)^{t-s}\ell'_{k,s} \tag{E.49}$$

and

$$[\overline{W}'_t]_j = [\overline{W}_t^{(2)}]_j + w_j\frac{1}{N}\eta_2\sum_{s\leq t}(1-\eta_2\lambda)^{t-s}\sum_{i\in\bar{\mathcal{M}}_2}\ell'_{i,s}\left[\mathbb{1}([W_t]_jx_1^{(i)})-\mathbb{1}([W_0]_jx_1^{(i)})\right]x_1^{(i)} \tag{E.50}$$

By the above calculation, (E.47), and Lemma 5.2, we have:

$$\|\overline{W}'_t\|_F \leq \|\overline{W}_t^{(2)}\|_F + \frac{1}{\lambda}\widetilde{O}\left(\sqrt{\varepsilon_s+\sqrt{\frac{\eta_2}{\eta_1}}}\right) \leq \widetilde{O}\left(\varepsilon_3\sqrt{d}\right) \tag{E.51}$$

where the last inequality followed by our choice of parameters. $\qquad\square$

Using the decomposition lemma, the conclusion of Lemma 5.3 now follows via computation.

*Proof of Lemma 5.3.* We first show that the network output on $x_1^{(i)}$ is close to that of some kernel prediction function by applying Lemma B.6. We vector-multiply the equality $[\overline{W}_t]_j = w_j \sum_{k\in\bar{\mathcal{M}}_2} \alpha_k x_1^{(k)} \mathbb{1}([W_0]_j x_1^{(k)}) + [\bar{W}_t']_j$ on both sides by $w_j \mathbb{1}([W_0]_j x_1^{(i)})$ and sum over all $j$ to get:

$$\left| \sum_{j\in[m]} w_j \langle [\overline{W}_t]_j, x_1^{(i)} \rangle \mathbb{1}([W_0]_j x_1^{(i)}) - \frac{1}{m} \sum_{j\in[m]} \sum_{k\in\bar{\mathcal{M}}_2} \alpha_k \langle x_1^{(k)}, x_1^{(i)} \rangle \mathbb{1}([W_0]_j x_1^{(k)}) \mathbb{1}([W_0]_j x_1^{(i)}) \right| \tag{E.52}$$

$$= \left| \sum_{j\in[m]} w_j \langle [\bar{W}_t']_j, x_1^{(i)} \rangle \mathbb{1}([W_0]_j x_1^{(i)}) \right| \tag{E.53}$$

$$\leq \sqrt{\sum_{j\in[m]} \langle [\bar{W}_t']_j, x_1^{(i)} \rangle^2} \qquad \text{(by Cauchy-Schwarz)}$$

$$= \|\overline{W}_t' x_1^{(i)}\|_2 \tag{E.54}$$

Let us define the function $\mathfrak{U}$ as:

$$\mathfrak{U}(x_1) := \frac{1}{m} \sum_{j\in[m]} \sum_{k\in\bar{\mathcal{M}}_2} \alpha_k \langle x_1^{(k)}, x_1 \rangle \mathbb{1}([W_0]_j x_1^{(k)}) \mathbb{1}(\langle [W_0]_j, x_1 \rangle) \tag{E.55}$$

Note that $\mathfrak{U}$ is some kernel prediction function. Since each $[W_0]_j$ is distributed as a vector of i.i.d. spherical Gaussians, we know that for fixed $x_1^{(k)}, x_1$:

$$\mathbb{E}\left[ \mathbb{1}([W_0]_j x_1^{(k)}) \mathbb{1}(\langle [W_0]_j, x_1 \rangle) \right] = \frac{1}{2\pi} \arccos \Theta(x_1^{(k)}, x_1) \tag{E.56}$$

In the above equation $\Theta(x_1^{(k)}, x_1^{(i)})$ is the principle angle between $x_1^{(k)}, x_1^{(i)}$. Since each $[W_0]_j$ is i.i.d., with basic concentration bounds, we know that w.h.p.

$$\mathfrak{U}(x_1^{(i)}) = \sum_{k\in\bar{\mathcal{M}}_2} \alpha_k \langle x_1^{(k)}, x_1^{(i)} \rangle \frac{1}{2\pi} \arccos \Theta(x_1^{(k)}, x_1^{(i)}) \pm O(m^{-1/6})$$

$$= \frac{1}{2} \alpha_i \|x_1^{(i)}\|_2^2$$

$$+ \sum_{k\in\bar{\mathcal{M}}_2, k\neq i} \alpha_k \langle x_1^{(k)}, x_1^{(i)} \rangle \frac{1}{4} \left( 1 - \frac{1}{2\pi} \frac{\langle x_1^{(k)}, x_1^{(i)} \rangle}{\|x_1^{(k)}\|_2 \|x_1^{(i)}\|_2} \pm O\left( \frac{\langle x_1^{(k)}, x_1^{(i)} \rangle}{\|x_1^{(k)}\|_2 \|x_1^{(i)}\|_2} \right)^3 \right)$$

$$\pm O(m^{-1/6}) \qquad \text{(by Taylor expansion of } \arccos)$$

$$= \frac{1}{2} \alpha_i \|x_1^{(i)}\|_2^2 \tag{E.57}$$

$$+ \sum_{k\in\bar{\mathcal{M}}_2, k\neq i} \alpha_k \langle x_1^{(k)}, x_1^{(i)} \rangle \frac{1}{4} \left( 1 - \frac{1}{2\pi} \frac{\langle x_1^{(k)}, x_1^{(i)} \rangle}{\|x_1^{(k)}\|_2 \|x_1^{(i)}\|_2} \pm \widetilde{O}\left( d^{-3/2} \right) \right) \pm O(m^{-1/6})$$

The last inequality uses the fact that w.h.p. for $k \neq i$, $\frac{\langle x_1^{(k)}, x_1^{(i)} \rangle}{\|x_1^{(k)}\|_2 \|x_1^{(i)}\|_2} = \widetilde{O}(d^{-1/2})$.

Let us define $\alpha = \frac{1}{4}\sum_{k\in\bar{\mathcal{M}}_2}\alpha_k x_1^{(k)}$; then

$$\left| \sum_{k\in\bar{\mathcal{M}}_2} \alpha_k \langle x_1^{(k)}, x_1^{(i)}\rangle \frac{1}{4}\left(1 - \frac{1}{2\pi}\frac{\langle x_1^{(k)}, x_1^{(i)}\rangle}{\|x_1^{(k)}\|_2\|x_1^{(i)}\|_2}\right)\right| \tag{E.58}$$

$$\leq |\langle \alpha, x_1^{(i)}\rangle| + \frac{1}{8\pi}\sum_{k\in\bar{\mathcal{M}}_2}|\alpha_k|\frac{\langle x_1^{(k)}, x_1^{(i)}\rangle^2}{\|x_1^{(k)}\|_2\|x_1^{(i)}\|_2} \tag{E.59}$$

$$\leq |\langle \alpha, x_1^{(i)}\rangle| + |\alpha_i\langle x_1^{(i)}, x_1^{(i)}\rangle| + \frac{1}{d}\widetilde{O}\left(\sum_{k\in\bar{\mathcal{M}}_2, k\neq i}|\alpha_k|\right) \tag{E.60}$$

Since the training loss is at $\varepsilon_2 \leq p/10$, we know that $\frac{1}{|\bar{\mathcal{M}}_2|}\sum_{i\in\bar{\mathcal{M}}_2}|\mathfrak{U}(x_1^{(i)})| \geq 1$ (or else the loss would not be low).

Since $|\mathfrak{U}(x_1^{(i)})| \leq |\langle \alpha, x_1^{(i)}\rangle| + \frac{3}{2}|\alpha_i|\|x_1^{(i)}\|_2^2 + \frac{1}{d}\widetilde{O}\left(\sum_{k\in\bar{\mathcal{M}}_2, k\neq i}|\alpha_k|\right) + O(m^{-1/6})$, we can get:

$$\frac{1}{|\bar{\mathcal{M}}_2|}\sum_{i\in\bar{\mathcal{M}}_2}\left(|\langle \alpha, x_1^{(i)}\rangle| + |\alpha_i| + \frac{1}{d}\widetilde{O}\left(\sum_{k\in\bar{\mathcal{M}}_2, k\neq i}|\alpha_k|\right)\right) \geq \frac{1}{2} \tag{E.61}$$

Since $Np \leq d$, this implies that

$$\frac{1}{|\bar{\mathcal{M}}_2|}\sum_{i\in\bar{\mathcal{M}}_2}\left(|\langle \alpha, x_1^{(i)}\rangle| + \widetilde{O}(|\alpha_i|)\right) \geq \frac{1}{2} \tag{E.62}$$

Thus, either $\frac{1}{|\bar{\mathcal{M}}_2|}\sum_{i\in\bar{\mathcal{M}}_2}|\langle \alpha, x_1^{(i)}\rangle| \geq \frac{1}{4}$, which implies that

$$\left\|(x_1^{(i)})_{i\in\bar{\mathcal{M}}_2}\alpha\right\|_2^2 = \sum_{i\in\bar{\mathcal{M}}_2}|\langle \alpha, x_1^{(i)}\rangle|^2 \geq \frac{|\bar{\mathcal{M}}_2|}{16} \tag{E.63}$$

Since w.h.p., $\|(x_1^{(i)})_{i\in\bar{\mathcal{M}}_2}\|_2 \leq O(1)$, we know that $\|\alpha\|_2 = \tilde{\Omega}(\sqrt{|\bar{\mathcal{M}}_2|}) = \tilde{\Omega}(\sqrt{Np})$.

The other possibility is that $\sum_{i\in\bar{\mathcal{M}}_2}\widetilde{O}(|\alpha_i|) \geq |\bar{\mathcal{M}}_2|/4$, which also implies that $\|\alpha\|_2 = \tilde{\Omega}(\sqrt{|\bar{\mathcal{M}}_2|}) = \tilde{\Omega}(\sqrt{Np})$ from Cauchy-Schwarz.

We now ready to conclude the proof: for randomly chosen $x_1$, it holds that

$$N_{W_0}(w, \overline{W}_t, x_1) \tag{E.64}$$

$$= \frac{1}{m}\sum_{j\in[m]}\sum_{k\in\bar{\mathcal{M}}_2}\alpha_k\langle x_1^{(k)}, x_1\rangle\mathbb{1}([W_0]_j x_1^{(k)})\mathbb{1}([W_0]_j x_1) \pm \|\overline{W}_t' x_1\|_2 \tag{E.65}$$

$$= \frac{1}{m}\sum_{j\in[m]}\sum_{k\in\bar{\mathcal{M}}_2}\alpha_k\langle x_1^{(k)}, x_1\rangle\mathbb{1}([W_0]_j x_1^{(k)})\mathbb{1}([W_0]_j x_1) \pm \widetilde{O}\left(\frac{\|\overline{W}_t'\|_F}{\sqrt{d}}\right) \tag{E.66}$$

$$= \frac{1}{m}\sum_{j\in[m]}\sum_{k\in\bar{\mathcal{M}}_2}\alpha_k\langle x_1^{(k)}, x_1\rangle\mathbb{1}([W_0]_j x_1^{(k)})\mathbb{1}([W_0]_j x_1) \pm \widetilde{O}(\varepsilon_3) \tag{E.67}$$

Now using the same expansion of $\mathfrak{U}$ as before gives

$$\mathfrak{U}(x_1) := \frac{1}{m}\sum_{j\in[m]}\sum_{k\in\bar{\mathcal{M}}_2}\alpha_k\langle x_1^{(k)}, x_1\rangle\mathbb{1}([W_0]_j x_1^{(k)})\mathbb{1}([W_0]_j x_1) \tag{E.68}$$

$$= \sum_{k\in\bar{\mathcal{M}}_2}\alpha_k\langle x_1^{(k)}, x_1\rangle\frac{\arccos(\Theta(x_1^{(k)}, x_1))}{2\pi} \pm O(m^{-1/6}) \tag{E.69}$$

Now we note that as the nonzero degrees in the polynomial expansion of $\arccos$ are all odd, we have

$$\mathfrak{U}(x_1) - \mathfrak{U}(-x_1) = 2\langle \alpha, x_1 \rangle \pm O(m^{-1/6}) \tag{E.70}$$

The end result is that by Lemma B.3, it will hold that:

$$r_t(x_1) = N_{W_0}(w, \overline{W}_t, x_1) \pm O\left(\frac{1}{\lambda} \times \left(\varepsilon_s + \sqrt{\frac{\eta_2}{\eta_1}}\right) + \tau_0 \log d\right) \tag{E.71}$$

$$= \mathfrak{U}(x_1) \pm \widetilde{O}(\varepsilon_3) \qquad \text{(by our choice of parameters)}$$

This implies that

$$r_t(x_1) - r_t(-x_1) = 2\langle \alpha, x_1 \rangle \pm \widetilde{O}(\varepsilon_3) \tag{E.72}$$

$\square$

# F  General case

## F.1  Mitigation strategy

Instead of using large learning rate and annealing to a small learning rate, the regularization effect also exists if we use a small learning rate ($\eta_2$) and large pre-activation noise and then decay the noise. Hence the update is given as:

$$U_{t+1} = U_t - \eta_2 \nabla_U(\widehat{L}_\lambda(u, U_t) + \xi_t) \tag{F.1}$$

where $\xi_t \sim N(0, \tau_\xi^2 I_{m \times m} \otimes I_{d \times d})$. However, the output of the network is given as:

$$f_t(x) = u^\top \left(\mathbb{1}(U_t x + \Xi_t) \odot (U_t x + \Xi_t)\right) \tag{F.2}$$

Here $\Xi_t \sim \mathcal{N}(0, \tau_t^2 I_{m \times m})$ is a (freshly random) gaussian variable at each iteration.

The following theorem holds:

**Theorem F.1** (General case). *The same conclusion as in Theorem 3.4 holds if we first use noise level $\tau_t = \tau_0$ and then anneal to $\tau_t = 0$ after $\widetilde{O}\left(\frac{d}{\eta_1 \varepsilon_1}\right)$ iterations.*

## F.2  Extension to two layer convolution network

We are also able to extend our results to convolutional networks. We consider a convolution network with $\frac{m}{k}$ channels, patch size $d$ and stride $d/k$ for some $k \leq d$. Thus, the $i$-th patch consists of input $x_{(i)} = (x_{(i-1)d/k+1}, \cdots, x_{(i-1)d/k+d})$. Hence for $u \in \mathbb{R}^m, U \in \mathbb{R}^{\frac{m}{k} \times d}$, where $u = (u_1, \cdots, u_k)$ for each $u_i \in \mathbb{R}^{\frac{m}{k}}$, the network is given as:

$$N_U(u, U; x) = \sum_{i \in [k]} u_i^\top [U x_{(i)}]_+ \tag{F.3}$$

For every $A \in \mathbb{R}^{\frac{m}{k} \times d}$, we also use the notation

$$N_A(u, U; x) = \sum_{i \in [k]} u_i^\top \mathbb{1}(A x_{(i)}) U x_{(i)} \tag{F.4}$$

$$N_A(u_i, U; x) = u_i^\top \mathbb{1}(A x_{(i)}) U x_{(i)} \tag{F.5}$$

We make a simplifying assumption that $z, \zeta$ are only supported on the last $d/k$ coordinates. The main theorem can be stated as the follows:

**Theorem F.2** (General case). *The same conclusions as in Theorem 3.4 and Theorem 3.5 hold if we replace the value of $r$ by $r/k$ and $d$ by $dk$ in both the theorem and in Assumption 3.3.*

Following the notation, we still denote
$$g_t(x) = g_t(x_{(k)}) = N_{U_t}(u, U_t; (0, x_{(k)})) \tag{F.6}$$
$$r_t(x) = r_t(x_{(1)}) = N_{U_t}(u, U_t; (x_{(1)}, 0)) \tag{F.7}$$

We use this definition so that $N_{U_t}(u, U_t; x) = g_t(x) + r_t(x)$ for every $t \geq 0$.

We denote $u = (u_1, \cdots, u_k)$ for the weight of the second layer associated with each convolution.

The main difference between the convolution setting and the simple case is that there is only one hidden weight that is shared across channels. However, since the output layers of these channels have different weights, we can disentangle these channels and think of them as updating "separately", which is given as the following two lemmas.

**Lemma F.3** (disentangle convolution 1). *For every fixed $x \in \mathbb{R}^{2d}$ and matrices $U_1, \cdots, U_k : \mathbb{R}^{\frac{m}{k} \times d}$ that can depend on $\widetilde{U}_t$ but not depend on $u$, with each $\|U_i\|_F \leq O\left(\frac{1}{\lambda}\right)$, we have w.h.p. over the randomness of $u, \widetilde{U}_t$:*

$$\left| N_{U_t}(u, \sum_{i \in [k]} u_i \odot U_i; x) - \sum_{i \in [k]} N_{U_t}(u_i, u_i \odot U_i; x) \right| \leq \widetilde{O}\left( k^2 \frac{\|x\|_2}{\lambda m^{1/2}} + k\varepsilon_s \|x\|_2 \right) \tag{F.8}$$

*Here $u_i \odot U_i = ((u_i)_j (U_i)_j)_{j \in [\frac{m}{k}]}$.*

**Lemma F.4** (disentangle convolution 2). *For every $s, t$, w.h.p. over the randomness of $u, \widetilde{U}_t, \widetilde{U}_s$, every $i, i' \in [k]$ with $i \neq i'$, and every $x, x' \in \mathbb{R}^d$, if we define $U_i = u_i \odot \mathbb{1}([U_s]x')x'^\top$, then as long as $\|\overline{U}_s\|_F, \|\overline{U}_t\|_F = O\left(\frac{1}{\lambda}\right)$, the following holds:*

$$|N_{U_t}(u_{i'}, U_i; x)| \leq \widetilde{O}\left( \frac{d^2 \|x\|_2 \|x'\|_2}{m^{1/2}} + \|x'\|_2 \|x\|_2 \sqrt{\varepsilon_s} + \|x\|_2 \varepsilon_s \right) \tag{F.9}$$

To apply this lemma, we can see that $u_i \odot \mathbb{1}([U_s]x')x'^\top$ is (a scaling of) the gradient coming from channel $i$ on input $x'$ at iteration $s$. This lemma says that it will have negligible effect on the output of channel $i' \neq i$ for (any) later iterations $t$. Hence at each iteration, every channel is updating almost separately.

*Proof of Lemma F.3.* By Lemma A.8, we know that

$$\left| N_{U_t}(u, \sum_{i \in [k]} u_i \odot U_i; x) - \sum_{i \in [k]} N_{U_t}(u, u_i \odot U_i; x) \right| \tag{F.10}$$

$$\leq \left| N_{\widetilde{U}_t}(u, \sum_{i \in [k]} u_i \odot U_i; x) - \sum_{i \in [k]} N_{\widetilde{U}_t}(u_i, u_i \odot U_i; x) \right| + O\left(k\varepsilon_s \|x\|_2\right) \tag{F.11}$$

Now, we can directly decompose

$$N_{\widetilde{U}_t}(u, \sum_{i \in [k]} u_i \odot U_i; x) = \sum_{i \in [k]} N_{\widetilde{U}_t}(u_i, u_i \odot U_i; x) \tag{F.12}$$

$$+ \sum_{i \in [k]} \sum_{i' \in [k], i' \neq i} N_{\widetilde{U}_t}(u_{i'}, u_i \odot U_i; x) \tag{F.13}$$

Since $U_i$ does not depend on the randomness of $u_{i'}$ but only $\widetilde{U}_t$, fixing $\widetilde{U}_t, U_i$ we know that since each entry of $u_{i'}$ i.i.d. mean zero, we have:

$$\mathbb{E}_{u_{i'}}\left[ N_{\widetilde{U}_t}(u_{i'}, u_i \odot U_i; x) \right] = 0 \tag{F.14}$$

Applying basic concentration bounds on $N_{\widetilde{U}_t}(u_{i'}, u_i \odot U_i; x)$, it holds that w.h.p. $|N_{\widetilde{U}_t}(u_{i'}, u_i \odot U_i; x)| \leq \widetilde{O}\left(\frac{\|x\|_2}{\lambda m}\right)$. Putting this back into Eq (F.12), we complete the proof.

$\square$

*Proof of Lemma F.4.* By Lemma A.8, we know that

$$|N_{U_t}(u_{i'}, U_i; x)| \leq \left|N_{\widetilde{U}_t}(u_{i'}, U_i; x)\right| + O(\varepsilon_s) \tag{F.15}$$

Hence, by definition, we have that

$$N_{\widetilde{U}_t}(u_{i'}, U_i; x) = N_{\widetilde{U}_t}(u_{i'}, u_i \odot \mathbb{1}([U_s]x')x'^\top; x) \tag{F.16}$$

Again by Lemma A.8, we know that $\|\mathbb{1}([U_s]) - \mathbb{1}(\widetilde{U}_s)\|_1 \leq \varepsilon_s m$, hence we have since the absolute value of each entry of $u_i$ is $m^{-1/2}$:

$$\left|N_{\widetilde{U}_t}(u_{i'}, u_i \odot \mathbb{1}([U_s]x')x'^\top; x)\right| \leq \left|N_{\widetilde{U}_t}(u_{i'}, u_i \odot \mathbb{1}([\widetilde{U}_s]x')x'^\top; x)\right| + \|x'\|_2\|x\|_2\sqrt{\varepsilon_s} \tag{F.17}$$

Now for fixed $x', x$, for $\left|N_{\widetilde{U}_t}(u_{i'}, u_i \odot \mathbb{1}([\widetilde{U}_s]x')x'^\top; x)\right|$, since $\mathbb{1}([\widetilde{U}_s]x')x'^\top$ does not depend on the randomness of $u_{i'}$, following the previous lemma we can show that with probability at least $1 - e^{-d^2}$, $\left|N_{\widetilde{U}_t}(u_{i'}, u_i \odot \mathbb{1}([\widetilde{U}_s]x')x'^\top; x)\right| \leq \widetilde{O}\left(\frac{\|x\|_2\|x'\|_2 d^2}{\lambda m}\right)$. Now, taking union bound over an epsilon-net of $x', x \in \mathbb{R}^d$ we conclude that for every $x, x'$, w.h.p. $\left|N_{\widetilde{U}_t}(u_{i'}, u_i \odot \mathbb{1}([\widetilde{U}_s]x')x'^\top; x)\right| \leq \widetilde{O}\left(\frac{\|x\|_2\|x'\|_2 d^2}{\lambda m}\right)$. Putting this back to Eq (F.17) we complete the proof. $\qquad\square$

We set $\varepsilon_c = \widetilde{O}\left(kd^4 \frac{1}{\lambda m^{1/2}}\right)$, and with this lemma, we can restate Lemma B.1, Lemma D.8 and Lemma E.1 in the following way: Suppose $\varepsilon_c \leq \min\{\varepsilon_1/10, \varepsilon_2'/10\}$ for every $x$ in the training set. Then the following lemmas hold by directly applying Lemma F.3.

**Corollary F.5.** *In the setting of Theorem F.2, there exists a solution $U^\star$ satisfying a) $\|U^\star\|_F^2 \leq O(dk \log^2(1/\varepsilon))$ and b) for every $t \geq 0$:*

$$K_t(U^\star) \leq q\log 2 + \epsilon_1/2 \tag{F.18}$$

**Corollary F.6.** *In the setting of Theorem F.2, there exists a solution $U^*$ satisfying $\|U^*\|_F^2 = \widetilde{O}\left(\frac{k}{\varepsilon_1^2 r}\right)$ and for every $t \leq \frac{1}{\eta_2 \lambda}$:*

$$K_{t_0+t}(\overline{U}_{t_0} + U^*) \leq \varepsilon_0 + \varepsilon_1 \tag{F.19}$$

**Corollary F.7.** *In the setting of Theorem F.2, there exists a solution $U^\star$ satisfying a) $\|U^\star\|_F^2 \leq \widetilde{O}\left(\frac{k}{\varepsilon_2'^2 r} + Npk\right)$ and b) for every $t \leq \frac{1}{\eta_2 \lambda}$,*

$$K_t(U^\star) \leq \varepsilon_2' \tag{F.20}$$

To prove these Lemmas, we can simply define $U^* = \sqrt{k}W^* + \sqrt{k}V^*$ for $W^*, V^*$ given in the original proof and apply Lemma F.3. The reason we need $k$ here is because there are $\frac{m}{k}$ channels instead of $m$, so the square norm scales up by a factor of $k$.

Now the next two convergence theorems follow directly from Lemma 4.1 and Lemma 4.3 and apply with initial learning rate $\eta_1$.

**Corollary F.8.** *In the setting of Theorem F.2 with initial learning rate $\eta_1$, at some step $t_0 \leq \widetilde{O}\left(\frac{dk}{\eta_1 \varepsilon_1}\right)$, the training loss $\widehat{L}(u, U_{t_0})$ becomes smaller than $q\log 2 + \epsilon_1$. Moreover, we have $\|\overline{U}_{t_0}\|_F^2 = O\left(dk \log^2(1/\varepsilon_1)\right)$.*

**Corollary F.9.** *In the setting of Theorem F.2, with initial learning rate $\eta_1$, there exists $t = \widetilde{O}\left(\frac{k}{\varepsilon_1^3 \eta_2 r}\right)$, such that after $t_0 + t$ iterations we have that*

$$L_{t_0+t} = O\left(\sqrt{\varepsilon_1/q}\right) \tag{F.21}$$

*Moreover, $\|\overline{U}_{t_0+t} - \overline{U}_{t_0}\|_F^2 \leq \widetilde{O}\left(\frac{k}{\varepsilon_1^2 r}\right)$*

The following statement applies when we use a small initial learning rate and follows from the proof of Lemma 5.1.

**Corollary F.10.** *In the setting of Theorem F.2, with initial learning rate $\eta_2$, there exists $t$ with*

$$t = \widetilde{O}\left(\frac{k}{\eta_2 \varepsilon_2'^3 r} + \frac{Npk}{\eta_2 \varepsilon_2'}\right) \tag{F.22}$$

*such that $L_t \leq \varepsilon_2'$ after $t$ iterations. Moreover, we have that $\|\overline{U}_t\|_F^2 \leq \widetilde{O}\left(\frac{k}{\varepsilon_2'^2 r} + Npk\right)$*

Now, the following lemma directly adapts from Lemma 4.2 by applying Lemma F.4:

**Lemma F.11.** *In the setting of Theorem F.2 with initial learning rate $\eta_1$, w.h.p., for every $t \leq \frac{1}{\eta_1 \lambda}$,*

$$|g_t(z+\zeta) + g_t(z-\zeta) - 2g_t(z)| \leq \widetilde{O}\left(\frac{r^2}{\lambda}\right) \tag{F.23}$$

With these lemmas, we can directly conclude the following:

**Corollary F.12.** *In the setting of Lemma F.9 with initial learning rate $\eta_1$, the following holds:*

$$\widehat{L}_{\mathcal{M}_1}(r_{t_0+t}) = O(\sqrt{\varepsilon_1/q}) \tag{F.24}$$

$$\widehat{L}_{\bar{\mathcal{M}}_1}(g_{t_0+t}) = O(\sqrt{\varepsilon_1/q^3}) \tag{F.25}$$

**Corollary F.13.** *In the setting with initial learning rate $\eta_2$, for every $\varepsilon_3 > 0$ such that $\frac{1}{\varepsilon_2'^8 \varepsilon_3^8 r} \leq \varepsilon_2' dk$, there exists $\alpha \in \mathbb{R}^d$ such that $\alpha \in span\{x_1^{(i),(j)}\}_{i \in \bar{\mathcal{M}}_2, j \in [k]}$ and $\alpha = \tilde{\Omega}(\sqrt{Np})$ such that w.h.p. over a randomly chosen $x_1 \sim \mathcal{N}(0, I/d)$, we have that*

$$r_t(x_1) - r_t(-x_1) = 2\langle \alpha, x_1 \rangle \pm \widetilde{O}\left(\varepsilon_3 + \frac{Npk}{d^{3/2}}\right) \tag{F.26}$$

*Here $x_1^{(i),(j)} = ([x_1^{(i)}]_s)_{s \in \{(j-1)d/k+1, (j-1)d/k+2, \cdots, d\}}$*

The final proof of Theorem F.2 follows directly from the proof of Theorem 3.4 and Theorem 3.5.

# G   Toolbox

**Lemma G.1.** *Let $X_1, X_2 \sim \mathcal{N}(0,1)$ and $a, b > 0$ such that $a^2 + b^2 = 1$. Then for every $\gamma_1, \gamma_2 \in \mathbb{R}$, we have that*

$$\left|\Pr\left[X_1 \geq \gamma_1 \mid aX_1 + bX_2 = \gamma_2\right] - \Pr\left[X_1 \geq \gamma_1 \mid aX_1 + bX_2 = 0\right]\right| \lesssim \frac{a|\gamma_2|}{b} \tag{G.1}$$

$$\Pr\left[|X_1| \leq \gamma_1 \mid aX_1 + bX_2 = \gamma_2\right] \lesssim \frac{|\gamma_1|}{b} \tag{G.2}$$

*Proof of Lemma G.1.* Without loss of generality, we assume $a\gamma_2/b \geq 0$. Let $Y_1 = aX_1 + bX_2$ and $Y_2 = bX_1 - aX_2$. We have that $Y_1, Y_2$ are independent random Gaussian variables with marginal distribution $\mathcal{N}(0,1)$. Moreover, $X_1 = aY_1 + bY_2$. Thus, $X_1 \mid aX_1 + bX_2 = \gamma_2$ is the same as $aY_1 + bY_2 \mid Y_1 = \gamma_2$, which has distribution $\mathcal{N}(a\gamma_2, b^2)$. Let $Z$ be a standard Gaussian, then

$$\left|\Pr\left[X_1 \geq \gamma_1 \mid aX_1 + bX_2 = \gamma_2\right] - \Pr\left[X_1 \geq \gamma_1 \mid aX_1 + bX_2 = 0\right]\right|$$

$$= \left|\Pr\left[bZ + a\gamma_2 \geq \gamma_1\right] - \Pr\left[bZ \geq \gamma_1\right]\right| = \left|\Pr\left[\frac{\gamma_1}{b} \geq Z \geq \frac{\gamma_1}{b} - \frac{a\gamma_2}{b}\right]\right|$$

$$\lesssim \left|\frac{a\gamma_2}{b}\right| \qquad \text{(beacuse the density of } \mathcal{N}(0,1) \text{ is bounded by } O(1))$$

Moreover,

$$\Pr\left[|X_1| \leq \gamma_1 \mid aX_1 + bX_2 = \gamma_2\right] = \Pr\left[|bZ + a\gamma_2| \leq \gamma_1\right] \lesssim |\gamma_1|/b \tag{G.3}$$

$\square$

Table 1: Validation accuracies for WideResNet16 trained and tested on original CIFAR-10 images without data augmentation.

| Method | Val. Acc |
|---|---|
| Large LR + anneal | 90.41% |
| Small LR + noise | 89.65% |
| Small LR | 84.93% |

**Lemma G.2.** *Let $M = M_0 + M_1$ where $M_1 \in \mathbb{R}^{d,d'}$ with $d' \le d$ is a matrix with each entry i.i.d. $\mathcal{N}(0, 1/d)$ and $M_0 = w^\star \beta^\top$ where $\|\beta\|_2 \le 1$ can depend on $M_1$. Then for every vector $z \in \mathbb{R}^{d'}$ we have that:*

$$\frac{\langle w^\star, Mz \rangle}{\|Mz\|_2} \le 0.9 \tag{G.4}$$

*Proof of Lemma G.2.* Note that $Mz = w^\star \langle \beta, z \rangle + M_1 z$. Since $M_1$ is a random gaussian matrix and $d' \le d$, we know that w.h.p. for every $z$ we have $\frac{\langle w^\star M_1 z \rangle}{\|M_1 z\|_2} \le \frac{\sqrt{2}}{2}$.

This implies that

$$\|Mz\|_2^2 = |\langle \beta, z \rangle|^2 + \|M_1 z\|_2^2 + 2 \langle \beta, z \rangle \langle w^\star, M_1 z \rangle \tag{G.5}$$

$$\ge |\langle \beta, z \rangle|^2 + \langle w^\star M_1 z \rangle^2 + 2 \langle \beta, z \rangle \langle w^\star, M_1 z \rangle + \frac{1}{2} \|M_1 z\|_2^2 \tag{G.6}$$

$$= (\langle \beta, z \rangle + \langle w^\star, M_1 z \rangle)^2 + \frac{1}{2} \|M_1 z\|_2^2 \tag{G.7}$$

$$= \langle w^\star, Mz \rangle^2 + \frac{1}{2} \|M_1 z\|_2^2 \tag{G.8}$$

This completes the proof.

$\square$

# H  Additional Details for Experiments

In this section we provide additional details on the experimental results of Section 6. All of our models were trained using a single NVIDIA TitanXp GPU and our code is implemented via PyTorch. We note that for all our experiments, the mean pixel is subtracted from the CIFAR image and then the image is divided by the standard deviation pixel. We use mean and standard deviation values in the PyTorch WideResNet implementation: https://github.com/xternalz/WideResNet-pytorch.

## H.1  Additional Details for Noise Mitigation Strategy

In this section, we provide additional details for the mitigation strategy for a small learning rate described in Section 6. In Table 1, we demonstrate on CIFAR-10 images without data augmentation that this regularization can indeed counteract the negative effects of small learning rate, as we report a 4.72% increase in validation accuracy when adding noise to a small learning rate.

We train for all models for 200 epochs, annealing the learning rates by a factor of 0.2 at the 60th, 120th, and 150th epoch for all models. The large learning rate model uses an initial learning rate of 0.1, whereas the small learning rate model uses initial learning rate of 0.01. The large learning rate is a standard hyperparameter setting for the WideResNet16 architecture, and we chose the small learning rate by scaling this value down. The other hyperparameter settings are standard. We remove data augmentation from the training set to isolate the effect of adding noise.

We add noise before every time we apply the relu activation. As it is costly to add i.i.d. noise that is the size of the entire hidden layer, we sample Gaussian noise that has shape equal to the last two dimensions of the 4 dimensional hidden layer, where the first two dimensions are batch size and number of channels, and duplicate this over the first 2 dimensions. We sample different noise for every batch.

Figure 4: Visualizations of CIFAR-10 images with patches added.

Our annealing schedule simply multiplies the noise level by a constant factor at every iteration. We tune the standard deviation of the noise to $0.2$ and the annealing rate to $0.995$ every iteration. We show results from a single trial as the small LR with noise algorithm already shows substantial improvement over vanilla small LR.

### H.2 Additional Details on Patch-Augmented CIFAR-10

We first describe in greater detail our method for producing the patch. First, the split of our data is the following: of the 50000 CIFAR-10 training images, 10000 will contain no patch and 40000 will have a patch. We generate this split randomly before training and keep it fixed. During a single epoch, we iterate through all images, loading the 10000 clean images the same way each time. For the remaining 40000 examples, we use a patch-only image with probability 0.2 and a patch mixed with CIFAR image with probability 0.8. Thus, 20% of the updates are on clean images, 16% of updates are on patches only, and 64% of updates are on mixed images, but the actual split of the data is slightly different because of our implementation.

The patch will be located in the center of the image. We visualize the patches in Figure 4. We generate the patch as follows: before training begins, we sample a random vector $z$ with i.i.d entries from $\mathcal{N}(0, \sigma_z^2)$ as well as $\zeta_i \sim [-\beta, \beta]$ for classes $i = 1, \ldots, 10$. Then to generate patch-only images, we add a scalar multiple of $\zeta_i$ to $z$ if the example belongs to class $i$. This scalar multiple is in the range $[-\alpha, \alpha]$ for some $\alpha$ we tune. We set coordinates not in the patch to $0$. To generate images that contain both patch and a CIFAR example, we simply add $z \pm \zeta_i$. In all, the hyperparameters we tune are $\sigma_z, \beta, \alpha$.

We must choose $\sigma, \beta, \alpha$ on the correct scale so that large and small learning rates don't both ignore the patch or overfit to the patch. For the experiment shown, $\sigma_z = 1.25, \beta = 0.1, \alpha = 1.75$.

Our large initial learning rate model trains with learning rate 0.1, annealing to 0.004 at the 30th epoch. and the small LR model trains with fixed learning rate 0.004. Our small LR with noise model trains with fixed learning rate 0.004, initial noise 0.4, and decays the noise to 4e-6 after the 30th epoch. We train all models for 60 epochs total, starting from the same dataset and choice of patches. Table 2 demonstrates the final validation accuracy numbers on patch-augmented and clean data.

Now we provide additional evidence that the generalization disparity is indeed due to the learning order effect and not simply because the large learning rate model can already generalize better on clean CIFAR-10 images. To see this, we consider the generalization error of models trained on 10000 clean CIFAR images: the small LR model achieves 65% validation accuracy, and the large LR model achieves 76% validation accuracy. For comparison, on the full clean dataset the small LR model achieves 83% validation accuracy whereas the large LR model achieves 90% accuracy.

We note that the final number of 69.89% clean image accuracy for the small LR model trained on the patch dataset is much closer to 65% than 83%, suggesting that it is indeed using a fraction of the available CIFAR samples because of learning order. On the other hand, the large LR model achieves final clean validation accuracy of 87.61% when trained on the patch dataset, which is very close to the 90% that is achievable training on the full clean dataset. This indicates that the large LR model is still using the majority of the images to learn CIFAR examples before annealing, as it has not yet memorized the patches.

Table 2: Validation accuracies for CIFAR-10 training dataset modified with patch. The mixed validation set similarly contains patches, but the clean set does not.

| Method | Mixed Val. Acc. | Clean Val. Acc. |
|---|---|---|
| Large LR + anneal | 95.35% | 87.61% |
| Small LR | 92.83% | 69.89% |
| Small LR + noise | 94.43% | 81.36% |