[Reviews · NeurIPS 2019]

Reviewer 1



1. The paper is mostly well-written; the problem well-stated and the ideas/solution very clear. I really liked the fact that authors presented the intuitive understanding before stating the technical results; this helps getting the essence of the result without getting lost in the subtle technicalities. In addition, the technical content is solid, rigorous and well presented. At some points, I found the paper to be heavy on notation but perhaps unavoidable here. 2. The authors introduce the notion of "learning order" - the order in which the learning algorithm learns "memorizable"(perhaps not the most apt word? - confused me for a moment) vs "generalizable" features. This is an interesting concept in its own essense and perhaps not elaborated/used, even on a toy distribution, so precisely before? Another good thing is instead of just stating the learning problem, they distinguish essential nuances with terminology like memorizable and generalizable, which are fairly intuitive. Moreover, this intuition of differences in learning memorizable and generalizable features could could reflect folklore wisdom about learning more finer features later? As an aside, this not only provides conceptual understanding but could potentially be helpful in principally exploiting domain knowledge in real world applications. 3. A basic question is how general is the phenomenon? The related work hardly discusses this - they instead focus on large batch vs small batch papers, or survey (perhaps not-so related content) like implicit regularization and adaptive gradient methods. I don't exactly see if small batch vs large batch captures this phenomenon; if yes (because its just a scaling?), should say explicitly. A small discussion on if the phenomenon has been observed for different datasets/tasks with different optimizers would provide a solid motivation and highlight the importance of understanding the phenomenon. Even if it is observed for a small subset of datasets (like vision tasks), this still helps to isolate that perhaps image-like distributions exhibit these memorizable vs generalizable behaviour. 4. At times, the writing is hand-wavy and confusing; for example the concept of "memorizable and generalizable", though intuitive, is sketchy and not formally explained. I assume that the authors wanted to give the informal essence, however since they are such an important part of the narrative, the authors should attempt to formalize these - perhaps identify based on sample complexity and/or complexity of the classifiers. Such a discussion is indeed attempted in lines 38-43, but could be shelved out better. In particular, phrases like "that is learnable by a low-complexity classifier, but are inherently noisy" could be ambiguous - what is "inherently noisy"? Other instances of hand-wavy language - line 72 "while adding noise before the activations which eventually gets annealed."? what do you mean by "getting annealed" - Is annealed a technical term in optimization/learning? (perhaps this is my ignorance). 5. Although the learning problem is explained well, what could help the presentation is perhaps a figure about the data distribution in 2d? Also memorizable and generalizable can also be discussed with a figure perhaps? 6. How important is the Gaussian noise injected in every step for the analysis? Also in experiments section line 282, "We test this empirically by adding small Gaussian noise during training 283 before every activation layer in a WideResNet1 [37] architecture." I am just wondering why its important to add before activation? Were it fine if I add noise after, or maybe just to the SGD iterates? A small comment on this specification would help. 7. There is hardly any discussion on contribution in proof techniques. The authors remark that even though the analysis is inspired form "kernel" regimes, it is unlike other works since "In our analysis, the underlying kernel is changing over time" (line 100). In that case, what tools are used, and moreover what analysis tools do they contribute so that perhaps they be used more generally? ------------------------ I thank the authors for providing clarification to the questions. In the light of this, I have increased my score by 1 point.

Reviewer 2



This is a very interesting theory paper showing that a neural network trained with a large learning rate and annealing generalizes better than the same network trained with small learning rate. The authors construct a data distribution which contains two types of features (low noise, hard-to-fit features, and high noise, easy-to-fit features). Under such a data distribution, the authors show that for a two-layer ReLU network trained with large learning rate and the same network trained with small learning rate, the order of learning two types of patterns is different, which eventually results in the gap in generalizations. In the experiment, the authors confirm on modified CIFAR-10 data that different learning rate schedule can indeed influence the learning order and generalization performance. The authors propose a fix to the small learning rate (inject noise before activations), which works both theoretically and empirically. In the proof, the authors carefully design a data distribution which contains low noise, hard-to-fit feature (Q-feature) and high noise, easy-to-fit feature (P-feature). In the data distribution, a very small fraction of data only has P-feature, a large fraction of data only has Q-feature and the remaining data has both P-feature and Q-feature. For the large learning rate and annealing, the network first learns P-feature and learns Q-feature after the annealing. On the contrast, the network with small learning rate quickly memorizes Q-feature and can only learn P-feature from the samples with only P-feature. Since the number of samples with only P-feature is small, the network can only learn a small margin, which results in the bad generalization performance on samples with only P-feature. Here are my major comments: 1. In the paper, the authors consider logistic loss with l_2 regularization. I was wondering whether this analysis can be extended to other losses (for example, mean square loss). It would also be good to explain the reason we need regularization here. 2. I feel this result requires the fraction of samples with only P-features to be small, otherwise the network with small learning rate can learn P-feature well just from these samples. So I was wondering whether it’s possible to identify a data distribution with only one type of features in which the large learning rate schedule still generalizes better than small learning rate. Of course, in this case, the order of learning features is the same (only one feature) and the generalization gap must due to some other reason. Although this is beyond the scope of this paper, it’s still good to talk about such possible directions. ----------------------------------------------------------------------- I have read the authors' response, which resolves most of my concerns. I think this is a very interesting theory paper. I will keep my score as it is.

Reviewer 3



Originality: This is the first paper to study the implicit regularization of large learning rate training theoretically and rigorously. Constructing a simple task that can rigorously show that larger learning rate training outperforms small learning rate is highly non-trivial. Quality and clarity: I did not check the math of this paper. The paper is well-written and the authors spend some efforts to help the readers to gain the intuition behind the paper. In my opinion, this is a very novel theory paper. I believe many researchers will explore their ideas in depth in the future. Update: I thank the authors' respond and will keep my score.

[Author Response · NeurIPS 2019]

1. We thank the reviewers for the detailed and insightful reviews. As the reviews noted, our work 1) introduces "novel
2. concepts" such as learning order, 2) provides theoretical and empirical evidence for our claims 3) proposes a mitigation
3. strategy for small learning rate which works both theoretically and empirically. We will address questions below and
4. incorporate feedback into our final revision.

5. **[R1]:** "I don't exactly see if small batch vs large batch captures this phenomenon; if yes ... should say explicitly."
6. • Yes, modulo some minor nuisances, the connection (smaller learning rate corresponds to larger batch) is simply
7. through the relative scale of the noise. Smith et al. [2017] make an explicit connection between small vs. large batch
8. and learning rate – this connection is due to the scaling of SGD noise, which can be increased by either increasing
9. learning rate or decreasing batch size.

10. **[R1]:** "A small discussion on if the phenomenon has been observed for different datasets/tasks with different optimizers"
11. • The existing literature on large/small batch or large/small learning rate largely focuses on many vision tasks (including
12. ImageNet) using SGD. The phenomenon may not be true for other optimizers such as Adam, though.

13. **[R1]:** "concept of "memorizable and generalizable", though intuitive, is sketchy and not formally explained ... authors
14. should attempt to formalize these - perhaps identify based on sample complexity"
15. • We acknowledge that the terms "memorizable" and "generalizable" are potentially confusing. Memorizable refers to
16. patterns that require low sample complexity but complex models to fit. On the other hand, "generalizable" patterns
17. require high sample complexity, but can be fit by linear models. We will revise our terminology to clarify this distinction.

18. **[R1]:** "what is "inherently noisy"?", "what do you mean by "getting annealed""
19. • By "inherently noisy", we refer to the fact that high noise in the datapoints will necessitate larger sample complexity.
20. For example, in our distribution $\mathcal{P}$, the norm of the noise is $\sqrt{d}$ times that of the signal $w^\star$, resulting in a $\Omega(d)$ sample
21. complexity. By "getting annealed", we mean reducing the pre-activation noise by a constant factor at a certain epoch.

22. **[R1]:** "How important is the Gaussian noise injected in every step for the analysis?"
23. • The Gaussian noise is essential for our analysis, as it models noise from SGD. Our analysis relies on the fact that the
24. scale of this Gaussian noise is larger with a larger learning rate.

25. **[R1]:** "Gaussian noise ... why ... add before activation? ... add noise after, or maybe just to the SGD iterates?"
26. • Our theory suggests that SGD affects the training dynamics and generalization through adding the noise to the
27. pre-activations, and therefore we did exactly the same thing as a mitigation strategy to imitate SGD. Adding noise to
28. the gradient would also serve as a mitigation strategy for our particular data distribution that we analyze. However,
29. because the analysis says that the fundamental benefit of the noise is perturbing the pre-activations, we suspect that
30. pre-activation noise will transfer better to large-scale real datasets (which is indeed true).

31. **[R1]:** "There is hardly any discussion on contribution in proof techniques", ""In our analysis, the underlying kernel is
32. changing over time" (line 100) ... what tools are used, and moreover what analysis tools do they contribute"
33. • We will clarify the intuitions and contributions of our proof techniques more in the revision of our paper. A main
34. contribution of our proof techniques is that we can deal with the changing kernel caused by the rapid changes of the
35. activation pattern in contrast to the NTK results that often require stable activation patterns. We extend the neural
36. tangent kernel techniques to the case with a sequence of kernels that share a common optimal classifier (Theorem C.2).

37. **[R2]:** "whether this analysis can be extended to other losses (for example, mean square loss)"
38. • It's unclear whether the analysis can be extended – with squared loss it's empirically unclear whether such a
39. phenomenon still exists. Theoretically, we at least need to modify the construction of the data distribution. We used the
40. property of logistic loss that as long as the sign of the prediction is accurate and the magnitude is sufficiently large, the
41. loss gradient vanishes. (This results in the behavior that the small learning rate ignores the $x_1$ components of examples
42. containing both $x_1$ and $x_2$ patterns.) However, this property does not immediately extend to squared loss.

43. **[R2]:** "It would also be good to explain the reason we need regularization here."
44. • The regularization simplifies our analysis. Recall that in each iteration, we add fresh Gaussian noise $\Xi_t$. Without
45. regularization, the noise part of the iterate $\widetilde{U}_t$ will keep accumulating. Adding mild regularization will balance the noise
46. level at a stable level. Though it is possible to extend our analysis to deal with no regularization, the main message
47. would remain the same. Therefore we considered a technically simpler setting.

48. **[R2]:** "whether it's possible to identify a data distribution with only one type of features in which the large learning rate
49. schedule still generalizes better than small learning rate ..."
50. • This is a great question for future work. We will speculate on an alternative perspective which relates to recent
51. work on the NTK perspective of neural nets [Du et al., 2018, Li and Liang, 2018, Jacot et al., 2018][1]. With sufficient
52. overparameterization and correct initialization, a small learning rate and no weight decay places the neural net in the
53. NTK regime. In this regime, the generalization of the neural net will only be as good as that of a kernel method on
54. random features. On the other hand, a larger learning rate could allow the optimization trajectory to depart the kernel
55. regime and make the neural net features non-random. Recent work [Wei et al., 2018][2] has shown that NTK can have
56. worse generalization than if the neural net features are allowed to be non-random.

## Footnotes

[1] Jacot, A. et. al. "Neural Tangent Kernel: Convergence and Generalization in Neural Networks."

[2] Wei, C. et. al. "Regularization Matters: Generalization and Optimization of Neural Nets v.s. their Induced Kernel."


[Meta-Review · NeurIPS 2019]

This paper theoretically justify the phenomenon that deep learning generalizes if a large learning rate is used in the early stage of training. To do so, this paper considers a rather simple problem setting and shows that 2-layer neural network generalizes better if it is trained by a large learning rate first followed by an annealed learning rate than a small learning rate. This concept is supported by numerical experiments on CIFAR10. This is an interesting paper and gives a rigorous insight to the well known phenomenon. This would open up a new research field on this topic that many researchers would follow.